# SAVANA: reliable analysis of somatic structural variants and copy number aberrations using long-read sequencing

Hillary Elrick [1,9], Carolin M. Sauer [1,9], Jose Espejo Valle-Inclan [1], Katherine Trevers [2,3], Melanie Tanguy [4], Sonia Zumalave [1], Solange De Noon[2,3], Francesc Muyas [1], Rita Cascão[5], Angela Afonso[5], Alistair G. Rust[1], Fernanda Amary[2,3], Roberto Tirabosco[2,3], Adam Giess[4], Timothy Freeman [4], Alona Sosinsky [4], Katherine Piculell[6], David T. Miller [6], Claudia C. Faria[5,7], Greg Elgar [4], Adrienne M. Flanagan [2,3] ✉ & Isidro Cortes-Ciriano [1,8] ✉

Accurate detection of somatic structural variants (SVs) and somatic copy number aberrations (SCNAs) is critical to study the mutational processes underpinning cancer evolution. Here we describe SAVANA, an algorithm designed to detect somatic SVs and SCNAs at single-haplotype resolution and estimate tumor purity and ploidy using long-read sequencing data with or without a germline control sample. We also establish best practices for benchmarking SV detection algorithms across the entire genome in a data-driven manner using replication and read-backed phasing analysis. Through the analysis of matched Illumina and nanopore whole-genome sequencing data for 99 human tumor-normal pairs, we show that SAVANA has significantly higher sensitivity and 13- and 82-times-higher specificity than the second and third-best performing algorithms. Moreover, SVs reported by SAVANA are highly consistent with those detected using short-read sequencing. In summary, SAVANA enables the application of long-read sequencing to detect SVs and SCNAs reliably.

Cancer genomes are shaped by diverse forms of structural variants (SVs), ranging from simple deletions, inversions and amplifications to complex patterns involving hundreds of SVs across multiple chromosomes[1–4]. SVs underpin most cancer driver mutations and underlie the somatic copy number aberrations (SCNAs) frequently observed in cancer cells[1,2,4]. To date, the study of cancer genomes has largely relied on short-read whole-genome sequencing (WGS) on the dominant Illumina platform, which has limited power to detect SVs

because breakpoints falling in repetitive regions cannot be unambiguously aligned to the human genome[5,6]. As a result, the landscape of SVs in cancer genomes remains partially mapped.

Long-read sequencing methods, such as the platforms commercialized by Oxford Nanopore Technologies (ONT) and Pacific Biosciences (PacBio), permit continuous reading of individual DNA molecules up to megabases[7,8]. While initially limited by high per-sample cost, high DNA input requirements, low and nonuniform flow cell yield

[1]European Molecular Biology Laboratory, European Bioinformatics Institute, Hinxton, Cambridge, UK. [2]Department of Histopathology, Royal National Orthopaedic Hospital, Stanmore, UK. [3]Research Department of Pathology, University College London Cancer Institute, London, UK. [4]Genomics England, London, UK. [5]Instituto de Medicina Molecular João Lobo Antunes, Faculdade de Medicina, Universidade de Lisboa, Lisbon, Portugal. [6]Division of Genetics and Genomics, Boston Children's Hospital, Boston, MA, USA. [7]Department of Neurosurgery, Hospital de Santa Maria, Centro Hospitalar Universitário Lisboa Norte, Lisbon, Portugal. [8]Wellcome Sanger Institute, Cambridge, UK. [9]These authors contributed equally: Hillary Elrick, Carolin M. Sauer. ✉e-mail: a.flanagan@ucl.ac.uk; icortes@ebi.ac.uk

and high sequencing error rates, long-read sequencing methods are now mature enough to enable the study of diverse types of genomic variants and perform gapless genome assembly[9]. The potential of long reads to characterize complex somatic SVs, such as viral integration events[10–12] and complex rearrangements[13–19], has catalyzed interest in the application of ONT and PacBio sequencing to study SVs in human cancers. Initial long-read studies of cancer cell lines and small sets of clinical samples reported thousands of somatic SVs only detectable in long-read data[20]. However, recent work revealed that most SVs underlying SCNAs larger than 10 kbp are detected by short-read sequencing[21], raising the question as to whether the higher rates of SVs detected in previous studies using long-read sequencing reflect true biological signal or false-positive calls due to limited algorithmic performance.

Over the past few years, several algorithms specifically designed for the detection of SVs using long reads have been developed[19,22–28]. To date, assessment of algorithmic performance has primarily relied on small truth sets of SVs detected in cancer cell lines[19,22–28]. However, relying on SV truth sets to benchmark SV detection methods is limited in multiple ways. First, SV truth sets are biased toward genomic regions amenable to experimental validation. This results in a bias toward excluding SVs in low-complexity regions, which are precisely the genomic elements intractable to short-read sequencing that could potentially be characterized more accurately using long reads. Second, the selection of candidate SVs for experimental validation is performed using existing SV detection algorithms, which might have their own biases (for example, variable sensitivity for SV detection across genomic regions). Third, SV truth sets are only reliable to evaluate the recall of algorithms, but not their specificity, because it is not possible to determine whether SVs called outside the genomic regions interrogated experimentally are artifacts or true SVs. Finally, experimental validation of SVs is time consuming and not easily scalable. Therefore, due to the lack of both large-scale long-read data sets for human cancers and best practices for benchmarking somatic SV detection algorithms in an unbiased manner, the relative performance of existing algorithms and the extent to which long-read sequencing approaches can improve the characterization of SVs and, by extension, SCNAs, as compared with short-read sequencing remains unclear.

To address these gaps, we present SAVANA, a computationally efficient algorithm specifically designed to detect both somatic SVs and SCNAs and infer tumor purity and ploidy, using long-read sequencing data from tumors with or without a matched germline control. Through benchmarking experiments of PCR-validated rearrangements and sequencing replicates of cancer cell lines and clinical samples, we show that SAVANA significantly outperforms existing algorithms in both sensitivity and specificity across long-read sequencing platforms, genomic regions, clonality levels, SV types and SV sizes. Analysis of matched, high-depth nanopore and Illumina WGS data for a collection of 99 clinical samples spanning diverse cancer types revealed that SAVANA detects most SVs and SCNAs detected by Illumina, while also discovering additional rearrangements, which were not detectable by short-read sequencing. In addition, tumor purity and ploidy estimates computed using SAVANA and a short-read WGS analysis pipeline suitable for delivering clinical tumor sequencing reports are strongly correlated. SAVANA is compliant with variant call format (VCF) specifications to facilitate downstream analyses, is implemented in Python 3 and is available at https://github.com/cortes-ciriano-lab/savana.

## Results

### Overview of SAVANA
We developed SAVANA to detect somatic SVs and SCNAs in long-read sequencing data, including the estimation of tumor purity and ploidy (Fig. 1a and Methods). In brief, SAVANA scans sequencing reads from a tumor and, when available, a matched normal sample to identify clusters of SV-supporting alignments. To facilitate the analysis of clinical samples lacking a matched germline control, SAVANA also permits the

detection of somatic SVs without requiring sequencing data from a matched normal sample (Methods). Given that the length of long reads is heterogeneous and alignment accuracy at breakpoints depends on read length, the same SV might be supported by sequencing alignments either fully or partially spanning the SV, which leads to different types of evidence for the same SV. For example, insertions and deletions might be supported by both gapped and split alignments[5,6]. Therefore, all alignments supporting the same SV type at a given genomic locus are considered to support the same putative SV (Methods). By default, SAVANA identifies and discards fold-back-like inversion artifacts generated by long-read sequencing (Supplementary Fig. 1 and Methods). SAVANA also includes functionalities to detect single breakends, in which only one of the two genomic regions bridged by an SV can be unambiguously mapped to the reference genome, thus enabling the detection of SVs involving low complexity or repetitive regions, such as centromeres or retrotransposons, and insertions of sequences not included in the reference genome, such as viral insertions (Methods). In addition, SAVANA provides functionalities to detect somatic SVs at single-haplotype resolution when the input sequencing reads are phased.

A key innovation of SAVANA is the use of machine learning to distinguish somatic SVs from sequencing and mapping errors. Specifically, SAVANA encodes each candidate somatic breakpoint using a set of covariates related to the location, SV type, number and orientation of alignments at the breakpoint junction and depth of coverage (Methods). These features serve to predict whether a given cluster of alignments is consistent with either a true somatic SV or sequencing errors using a machine learning model trained on a large collection of SVs detected using matched long-read and short-read sequencing data (Methods). In this way, true SVs are distinguished from clusters of alignments arising from both recurrent and non-recurrent artifacts.

To detect SCNAs, SAVANA utilizes somatic breakpoints and circular binary segmentation to partition the genome into regions showing equal read depth (Fig. 1a and Methods). Next, SAVANA infers the tumor purity by considering the mean B-allele frequency (BAF) values of heterozygous single nucleotide polymorphisms (SNPs) at regions with loss of heterozygosity (LOH). The key idea is that in a tumor without infiltration of non-neoplastic cells, the number of sequencing reads supporting a lost allele should be zero, and thus, the BAF values at regions with LOH should be 0 or 1. However, the infiltration of normal cells in the tumor results in a shift of the BAF values toward 0.5 proportional to the degree of normal infiltration, thus allowing inference of tumor purity[29]. For this calculation, SAVANA considers patient-specific heterozygous SNPs or polymorphic population SNPs[30] (Methods). SAVANA uses the copy number information to rescue breakpoints that did not pass the thresholds for being called using read alignments if they support copy number changepoints, thus increasing sensitivity. Finally, SAVANA determines the tumor ploidy and allele-specific copy number profile that best explain the observed sequencing read depth and BAF data given the estimated tumor purity (Fig. 1b and Methods).

### Establishing a training set for somatic SV detection
We performed high-molecular weight DNA extraction on 99 tumor-normal pairs (57 diverse soft-tissue sarcomas, 28 osteosarcomas and 14 glioblastomas) and sequenced the DNA using both nanopore WGS (median sequencing depth of 51× and 34× for tumor and normal samples, respectively) and Illumina WGS (118× and 41×) (Fig. 1b, Supplementary Fig. 2, Supplementary Table 1 and Methods). We achieved a median N50 of 15 and 21 kbp for tumor and normal nanopore sequencing reads, respectively. N50 refers to the shortest read length in base pairs at which the set of reads of that length and longer encompass 50% of the sequenced bases. After stringent quality control of the short and long-read data, 92 tumors were considered for further analysis (Supplementary Table 1).

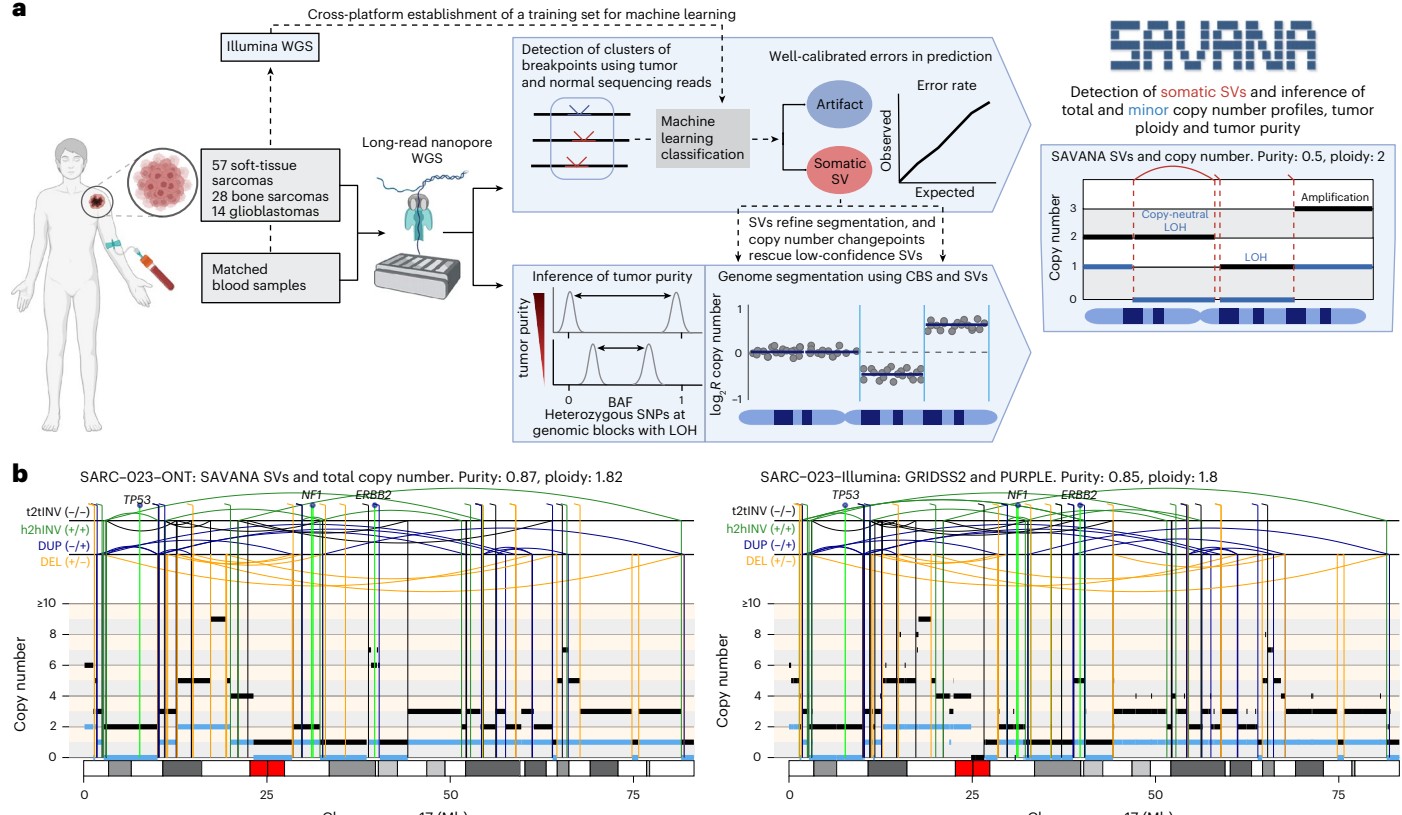

**Fig. 1 | Overview of SAVANA. a**, The methodology for the analysis of somatic SVs, SCNAs, tumor purity and ploidy using SAVANA. Created with BioRender. com. **b**, An example of a complex genomic rearrangement profile detected using SAVANA and matched short-read sequencing data. The total and minor allele copy number data are represented in black and blue, respectively. DEL, deletion-like rearrangement; DUP, duplication-like rearrangement; h2hINV, head-to-head inversion; t2tINV, tail-to-tail inversion. CBS, circular binary segmentation; Mb, megabase pairs.

To assemble a high-quality training set of true somatic SVs (true positives) and artifacts (true negatives), we used a clinical-grade short-read WGS data analysis pipeline[31] to detect somatic SVs in the Illumina data (Methods). Next, we labeled SVs detected by SAVANA in the long-read data as true somatic SVs ($n = 52,464$) if they were also detected in the short-read data and as false positives otherwise ($n = 14,282,014$). To prevent misclassifying true somatic SVs only detectable by long-read sequencing as false positives, we excluded from the training data SVs identified with high-quality support in long-read data but not found in short-read data (Methods).

To assess the generalization of our approach to distinguish true from false-positive SVs, we trained leave-one-tumor-out models using random forest (RF) classification. Specifically, we classified the SVs detected in each tumor using an RF model trained on the SVs detected in all other tumors in the cohort. Overall, the performance of the RF models was high, with a mean area under the receiver operating characteristic curve value of 0.98 (range 0.97–0.98). The most predictive covariates included the number of supporting alignments in the tumor and matched normal sample, SV length, the number of unphased alignments supporting a breakpoint and the count of reads clustered in the normal sample supporting any breakpoint orientation (Supplementary Fig. 3 and Supplementary Table 2).

By default, SAVANA uses RF classification to identify somatic SVs, a type of algorithm previously used to detect germline SVs using sequencing data[32]. However, SAVANA also provides functionalities to assess the reliability of individual predictions and ensure high model performance for the minority class (true somatic SVs in this case) using Mondrian conformal prediction (MCP). While some SV detection algorithms provide quality scores[33], these scores are not formally

guaranteed to be well calibrated, since the fraction of erroneous predictions does not necessarily correlate with class probability scores. By contrast, MCP guarantees mathematically that the fraction of predictions for which the set of predicted labels does not include the true class is not higher than a user-defined tolerated error provided that the training and test examples are independently and identically distributed[34,35]. The maximum error rate is ensured for each category, including the minority class, which is particularly important when modeling highly imbalanced data sets[36,37]. In practice, MCP predicts four classes: true positive (somatic SV), 'noise' (noisy region prone to artifacts or germline SV), 'null' and 'both'. The 'null' category indicates that, at the chosen confidence level, the model considers the SV to be too dissimilar to the training datapoints to make a reliable prediction, indicating that the instance is an outlier (Supplementary Fig. 4). By contrast, the 'both' category is assigned to instances that are similar to both the somatic SVs and 'noise' instances in the training data, and thus, the model cannot decide on a single category. In practice, 'null' and 'both' predictions are considered to be erroneous and correct, respectively. Increasing the confidence level forces the model to reduce the number of 'null' predictions, which might cause an increase in SVs classified as 'both' to exceed the specified error rate[36]. Overall, MCP provides a flexible and mathematically sound approach to assess the reliability of individual SV calls.

## SAVANA outperforms existing SV detection algorithms

Next, we compared the performance of SAVANA against existing SV detection algorithms specifically developed for long-read data, including widely used methods originally designed for germline SV discovery that have been frequently used for the detection of somatic

SVs (Sniffles2 (ref. [28]), cuteSV[24] and SVIM[23]), as well as algorithms developed to detect somatic SVs using matched tumor-normal data (NanomonSV[19], Severus[22] and SVision-pro)[27] (Methods). The number of SVs detected in each tumor varied by up to two orders of magnitude across SV callers (Supplementary Figs. 5 and 6). To assess whether the variable rates of somatic SVs reported by existing algorithms and SAVANA are the result of variable sensitivity or specificity, we first used a truth set of 68 somatic SVs detected in the melanoma cell line COLO829 and validated using either PCR or capture-based technologies[38] (Methods). To this aim, we analyzed long-read PacBio and ONT WGS data from COLO829 and a matched normal cell line, COLO829BL (Methods). Overall, SAVANA showed significantly higher recall and specificity than existing algorithms across sequencing platforms and flow cell versions ($P < 0.0001$, two-sided Wilcoxon test) (Supplementary Fig. 7).

We next sought to establish best practices for benchmarking somatic SV detection algorithms genome-wide in an unbiased manner. Specifically, we propose the use of simulated sequencing replicates. The key idea is that true somatic SVs should be detected in all replicates, whereas false-positive calls resulting from library preparation or sequencing errors should only be detected in one replicate. Replicates have been widely used to benchmark point mutation and indel detection algorithms[39–41]. More broadly, technical and biological replicates represent a cornerstone of experimental design across various research domains to assess the reproducibility and robustness of experimental and computational results, including sequencing data analysis[40,42]. Yet, replication remains a surprisingly underutilized approach for benchmarking somatic SV detection algorithms.

Here, we simulated replicates by randomly splitting the sequencing reads from each tumor sample into two different binary alignment map (BAM) files and applied each SV detection algorithm on each replicate independently (Fig. 2a). This strategy allows us to simulate sequencing replicates with the same sequencing depth, thus overcoming the impact that uneven flow cell yield across sequencing replicates could have on SV detection sensitivity. Next, we compared the sets of somatic SVs detected by each algorithm across replicates. We found that the number of somatic SVs detected in just one replicate varied by up to two orders of magnitude across algorithms (Fig. 2b). Consistent with its higher performance on the COLO829 data set, SAVANA showed a significantly higher concordance across replicates compared with other algorithms across diverse tumor types ($P < 0.001$, two-sided Wilcoxon test) (Fig. 2c and Methods).

A potential limitation of replicates is that tumors often show high levels of intratumor genetic heterogeneity[43,44]. As a result, true somatic SVs present at low allele frequency (AF) in a tumor might

only be detected in one replicate if the number of sequencing reads in other replicates is not enough to meet the threshold for SV calling[41,42]. Thus, we next sought to investigate whether the variable number of SVs detected by different algorithms is the result of variable sensitivity for low-AF SVs. By analyzing the fraction of SVs detected in both replicates as a function of AF (Fig. 2d and Supplementary Fig. 8) or the number of reads supporting each SV (Supplementary Fig. 9a), we found that SAVANA shows a higher and uniform concordance rate across SV clonality levels. In addition, SAVANA uniformly yields the highest fraction of concordant SVs across tumors (Supplementary Fig. 9b), classes of repeats (Supplementary Fig. 9c) and SV types (Fig. 2e). The higher concordance across replicates, especially in the low-AF range, indicates that SAVANA shows both higher specificity and sensitivity than existing methods.

We found that the relative number of SVs of each SV type was highly variable across algorithms, with cuteSV, Severus, Sniffles2 and SVIM detecting thousands of deletions and insertions per sample (Supplementary Fig. 10). In addition, the concordance across replicates was highly variable across SV types, with insertions and deletions showing the lowest concordance rates (Fig. 2f). Because artifactual SVs mapping to genomic regions prone to sequencing or mapping errors, such as homopolymers[45,46], might be detected in both replicates, we next sought to investigate whether the high rates of insertions and deletions reported by existing algorithms represent true biological events or artifacts. Analysis of the genomic distribution of SVs revealed that most of the insertions and deletions detected by these algorithms, especially those smaller than 500 bp, mapped to microsatellites (that is, tandem repeats) (Fig. 2f and Supplementary Fig. 11). By contrast, the rate of insertions and deletions detected by SAVANA at microsatellite loci was significantly lower ($P < 0.0001$, two-sided Wilcoxon test) (Supplementary Fig. 12). In fact, the total number of insertions and deletions at microsatellites detected by the other algorithms was two to three orders of magnitude higher than SAVANA for all tumor types analyzed (Supplementary Fig. 12). High rates of insertions and deletions at microsatellites, a phenotype known as microsatellite instability (MSI), are associated with inactivation of the mismatch repair pathway[47,48]. However, the rates of MSI in the tumors analyzed in this study are very low[47], and we did not detect inactivation of MMR genes or MSI using matched Illumina WGS data for the tumors analyzed here. Taken together, these results indicate that the high rates of insertions and deletions at microsatellite loci reported by existing SV detection algorithms are false-positive calls resulting from sequencing or mapping errors. By contrast, the rates and types of SV detected by SAVANA are low and consistent with tumor biology.

**Fig. 2 | Benchmarking of SAVANA against existing algorithms using replicates. a**, A schematic representation of the replicate analysis strategy implemented to benchmark the performance of somatic SV detection algorithms. Created with BioRender.com. **b**, The distribution of the number of somatic SVs detected by each algorithm stratified based on whether they were detected in one (red) or both (green) replicates. Each point represents a tumor sample ($n = 64$) that has been split into two replicates. **c**, A comparison of the fraction of somatic SVs detected in both replicates by each algorithm. The bars report the result across all samples. The error bars report the 95% confidence interval. Significance with respect to SAVANA was assessed using the two-sided Student's $t$-test (***$P < 0.0001$). The $P$ values for SAVANA compared with all other algorithms were $P < 2.2 \times 10^{-16}$. **d**, The number of somatic SVs detected in both replicates divided by the total number of somatic SVs detected as a function of allele fraction. The results for the entire cohort are shown ($n = 64$). The size of the dots represents the number of somatic SVs in each group. Only algorithms that report the allele fraction or information that can be used to calculate the allele fraction were included in this analysis. **e**, A comparison of the count of somatic SVs detected in one (red) or both (green) replicates stratified by SV type. Note the different $x$-axis scales used to reflect the number of SVs reported

by each algorithm. **c–e** show the aggregated results for the 64 samples with the highest sequencing depth. **f**, The fraction of deletions in replicates mapping to microsatellite regions. Each point represents a tumor sample ($n = 64$) that has been split into two replicates. The significance was assessed using the two-sided Wilcoxon's rank test (****$P < 0.00001$). The $P$ values for the comparison between SAVANA against SVIM, NanomonSV, cuteSV, Sniffles2, SVision-pro and Severus were $P < 2.2 \times 10^{-16}$, $P = 4.9 \times 10^{-10}$, $P < 2.2 \times 10^{-16}$, $P < 2.2 \times 10^{-16}$, $P = 5.1 \times 10^{-13}$ and $P < 2.2 \times 10^{-16}$, respectively. **g**, A haplotype consistency analysis of SV-supporting reads using read-backed phasing across the entire cohort. Each dot represents an SV. The $x$ and $y$ axes report the number of sequencing reads supporting each SV that are assigned to either parental allele (arbitrarily labeled as 'allele 1' and 'allele 2', respectively). **h**, The same data shown in **g** depicted in a stacked barplot format. In **g** and **h**, the SVs supported by sequencing reads assigned to only one parental allele are colored in green. The SVs with significant read support from both parental alleles are shown in red, and those with inconclusive results are shown in blue. The box plots in **b** and **f** show the median, first and third quartiles (boxes) and the whiskers encompass observations within 1.5× the interquartile range from the first and third quartiles.

## SVs detected by SAVANA show consistent read-backed phasing

Another potential limitation of sequencing replicates is that false-positive SVs generated by recurrent artifacts or germline events might be detected in both replicates and, thus, misclassified as bona fide somatic SVs. To assess the impact of this issue on our results, we next performed read-backed phasing analysis of SV-supporting reads, which is a concept that has been extensively used to detect

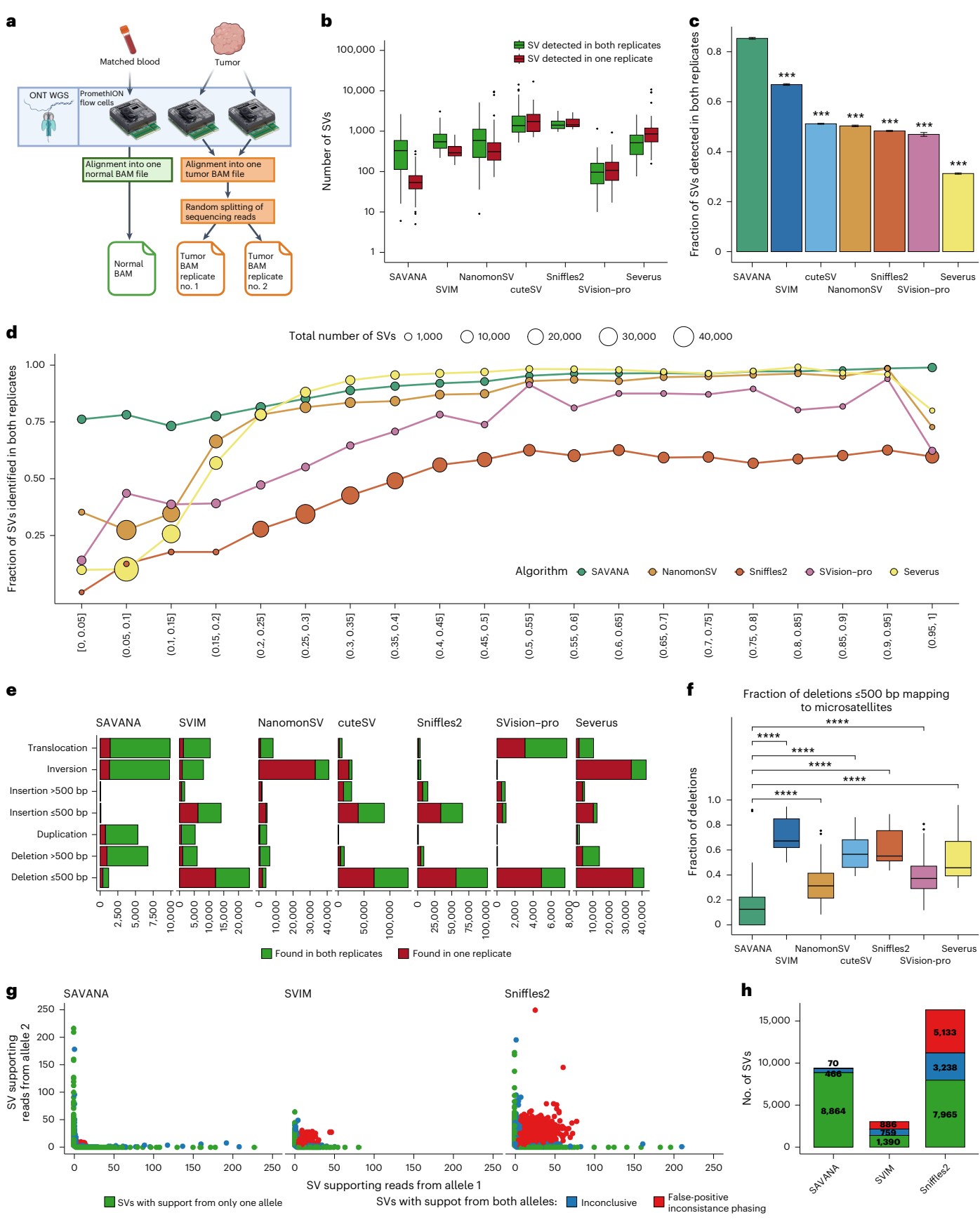

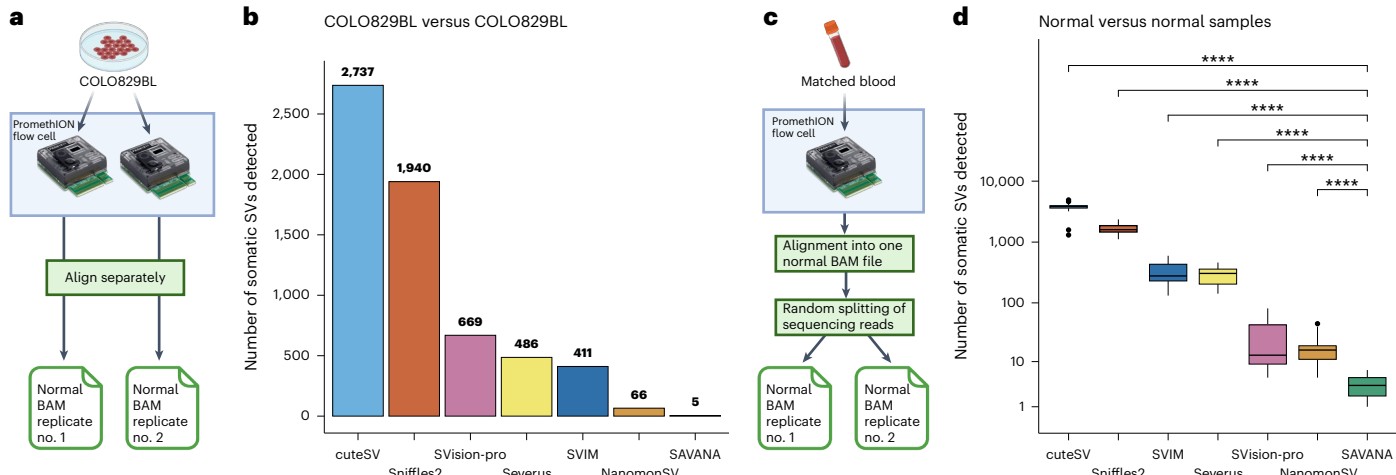

**Fig. 3 | Benchmarking the specificity of SAVANA against existing algorithms using sequencing replicates of matched germline controls. a**, Schematic representation of the COLO829BL normal flow cell replicate analysis strategy implemented to quantify the false-positive rate of somatic SV detection algorithms. Created with BioRender.com. **b**, The number of somatic SVs detected in the COLO829BL cell line when running the algorithms benchmarked using a normal replicate as the tumor sample. The number on top of each bar indicates the number of false-positive calls for each algorithm. **c**, Schematic representation of the replicate analysis strategy implemented to quantify the false-positive rate of somatic SV detection algorithms. Created with BioRender.com. **d**, The

distribution of false-positive SV calls detected when running the SV detection algorithms benchmarked using replicates of 37 whole-blood normal samples with at least 30× coverage, generated in silico by splitting sequencing reads randomly into two BAM files. Each dot represents one blood sample. The significance in **d** was assessed using the two-sided Wilcoxon's rank test (****$P < 0.0001$). The $P$ values for the comparison of SAVANA against cuteSV, Sniffles2, SVIM, Severus, SVision-pro and NanomonSV were $P = 2.5 \times 10^{-12}$, $P = 2.5 \times 10^{-12}$, $P = 2.5 \times 10^{-12}$, $P = 2.5 \times 10^{-12}$, $P = 1.4 \times 10^{-11}$ and $P = 7 \times 10^{-12}$, respectively. The box plots in **d** show the median, first and third quartiles (boxes), and the whiskers encompass observations within 1.5× the interquartile range from the first and third quartiles.

false-positive mutation calls in the context of cancer and somatic mosaicism research[49–53]. Specifically, the reads supporting a somatic SV should be phased to a single haplotype, whereas artifactual SVs are likely to be supported by sequencing reads from both parental alleles (Methods). The vast majority of SVs detected by SAVANA in both replicates are supported by reads phased to a single parental allele, whereas a large number of the sequencing reads supporting the SVs detected in both or a single replicate by other algorithms show inconsistent read-backed phasing (Fig. 2g,h). We obtained comparable results when analyzing the SV calls computed using all sequencing reads from each tumor at genomic regions without LOH (Fig. 2g,h and Methods) and in the analysis of simulated sequencing replicates (Supplementary Fig. 13). Specifically, in the SVs for which we could reliably perform this analysis (Methods), inconsistent read-backed phasing was only detected in 0.15% of the SVs reported by SAVANA (19/12749). By contrast, 25% of the SVs detected by Sniffles2 in both replicates (166 times higher as compared with SAVANA) show significant evidence of inconsistent read-backed phasing (Fig. 2g,h), indicating that SAVANA distinguishes bona fide somatic SVs from artifacts.

**Using replicates of germline controls to quantify specificity**
Next, to quantify specificity, we harnessed both sequencing and simulated replicates of matched germline controls (Fig. 3a). Specifically, we first ran each algorithm on sequencing replicates of COLO829BL using one replicate as the tumor and the other one as the matched germline control. An algorithm with optimal specificity should not detect any somatic SVs in this setting. On the COLO829BL data set, SAVANA showed the lowest false-positive rate, reporting five false-positive SVs (Fig. 3b). In comparison, the other algorithms showed 13× (NanomonSV: 66 false positives), 82× (SVIM: 411), 97× (486 Severus), 134× (669 SVision-pro), 388× (Sniffles2: 1940) and 547× (cuteSV: 2737) higher false-positive rates (Fig. 3b). We next performed the same analysis using simulated replicates for the blood WGS data generated as matched germline controls for the tumor samples sequenced in this study. To this end, we randomly split the sequencing reads from the

matched normal sample into two BAM files (Fig. 3c). Next, each algorithm was run using one replicate as the tumor sample and the other as the matched normal. Consistent with the results on COLO829BL, SAVANA showed a significantly lower false-positive rate as compared with other algorithms, which showed tens to thousands of false positives per sample ($P < 0.0001$, two-sided Wilcoxon test) (Fig. 3d). Notably, the order of magnitude difference between existing methods and SAVANA is comparable for both the number of false positives (Fig. 3d) and the number of somatic SVs detected in tumors (Supplementary Figs. 5 and 6), thus suggesting that the variable number of somatic SVs detected by different methods is driven by the low specificity of existing algorithms. Investigation of rearrangement profiles revealed that somatic SVs detected by SAVANA largely agree with Illumina SV calls and that breakpoints map to copy number changepoints both detected using short-read and long-read data (Fig. 4a,b and Supplementary Figs. 14–19). By contrast, existing algorithms detect high rates of SVs, especially insertions and deletions, which are not supported by somatic copy number changepoints, while also missing SVs detected by SAVANA and present in Illumina WGS data (Fig. 4c–f and Supplementary Figs. 14–19).

Together, these results demonstrate that SAVANA consistently shows both significantly higher specificity and sensitivity than existing methods across diverse SV types and clonality levels and, further, that the high rate of somatic SVs detected by existing algorithms is explained by their low specificity rather than by higher sensitivity for somatic SV detection. Moreover, SAVANA offers significantly faster runtimes than most existing algorithms ($P < 0.0001$, two-sided Wilcoxon test) (Supplementary Fig. 20).

**Detecting cancer driver SVs and SCNAs in clinical samples**
Next, we investigated the differences between long and short-read sequencing for somatic SV detection. On average, 86% of the SVs detected in the Illumina data were also called in the long-read data (Fig. 5a, Supplementary Fig. 21 and Methods). By contrast, the recall for other algorithms was significantly lower ($P < 0.001$, two-sided Wilcoxon test) (Fig. 5a). Furthermore, using existing algorithms,

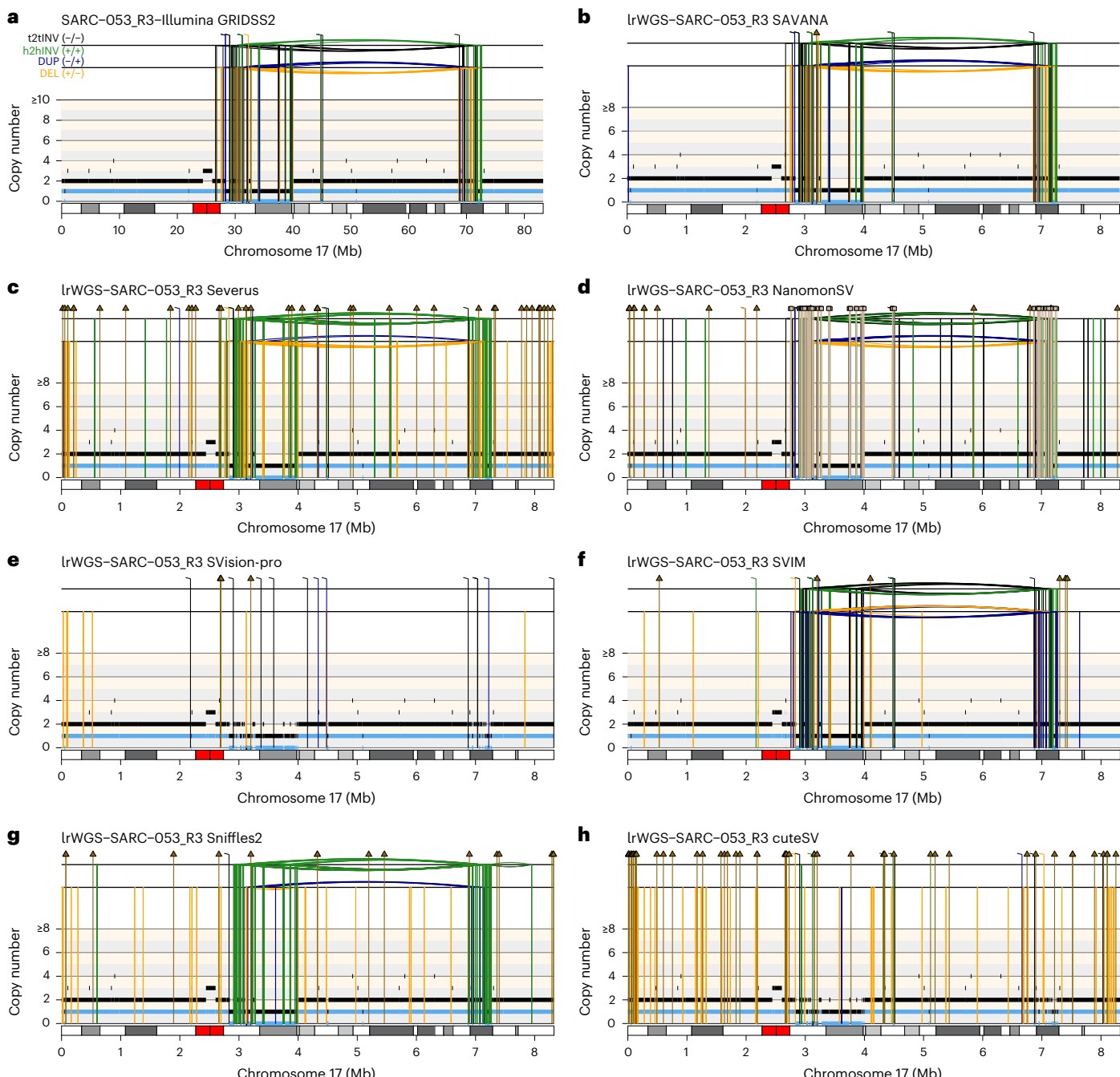

**Fig. 4 | Benchmarking of SV detection algorithms. a**, The somatic SVs and copy number profiles detected using GRIDSS2 and PURPLE in whole-genome short-read sequencing data. **b**–**h**,The somatic SVs were detected in matched long-read nanopore WGS data (lrWGS) using SAVANA (**b**), Severus (**c**), NanomonSV (**d**), SVision-pro (**e**), SVIM (**f**), Sniffles2 (**g**) and cuteSV (**h**). The copy number profiles shown in **a**–**h** were calculated using PURPLE and the short-read sequencing data.

The total and minor allele copy number data in **a**–**h** are represented in black and blue, respectively. DEL, deletion-like rearrangement; DUP, duplication-like rearrangement; h2hINV, head-to-head inversion; t2tINV, tail-to-tail inversion. The lines with a square at the top represent single breakends, and the lines with arrowheads mark insertions.

no more than 50% of the SVs detected in the long-read data were detected in the Illumina data (*P* < 0.0001, two-sided Wilcoxon test) (Fig. 5b,c and Supplementary Fig. 22). These low recall rates are consistent with the lower concordance across replicates observed for existing algorithms (Fig. 2). Moreover, SAVANA identified 92% (71/77) of somatic SVs in cancer driver genes detected in the Illumina data. In addition, 97% (71/73) of somatic SVs in cancer driver genes identified by SAVANA were also detected in Illumina data (Supplementary Fig. 23). The SVs detected in long-read data only were enriched in repetitive regions, including centromeres (Supplementary Table 3). Importantly,

SAVANA and long reads permitted the detection of SVs with breakpoints mapping in low-mappability regions, including driver SVs affecting *NF1* and *COL2A1* (Supplementary Figs. 24 and 25).

SAVANA detected diverse types of complex rearrangement encompassing SVs and SCNAs leading to the loss of tumor suppressor genes, such as chromothripsis events causing the inactivation of *CDKN2A*, *NF1*, *TP53* or *RB1* (Fig. 5d and Supplementary Figs. 26–35) or oncogene amplification across various tumor types, such as amplification of *CDK4*, *MDM2*, *CCNE1* and *MYC* in osteosarcomas and *MYC* and *EGFR* in glioblastomas[1] (Fig. 5e,f).

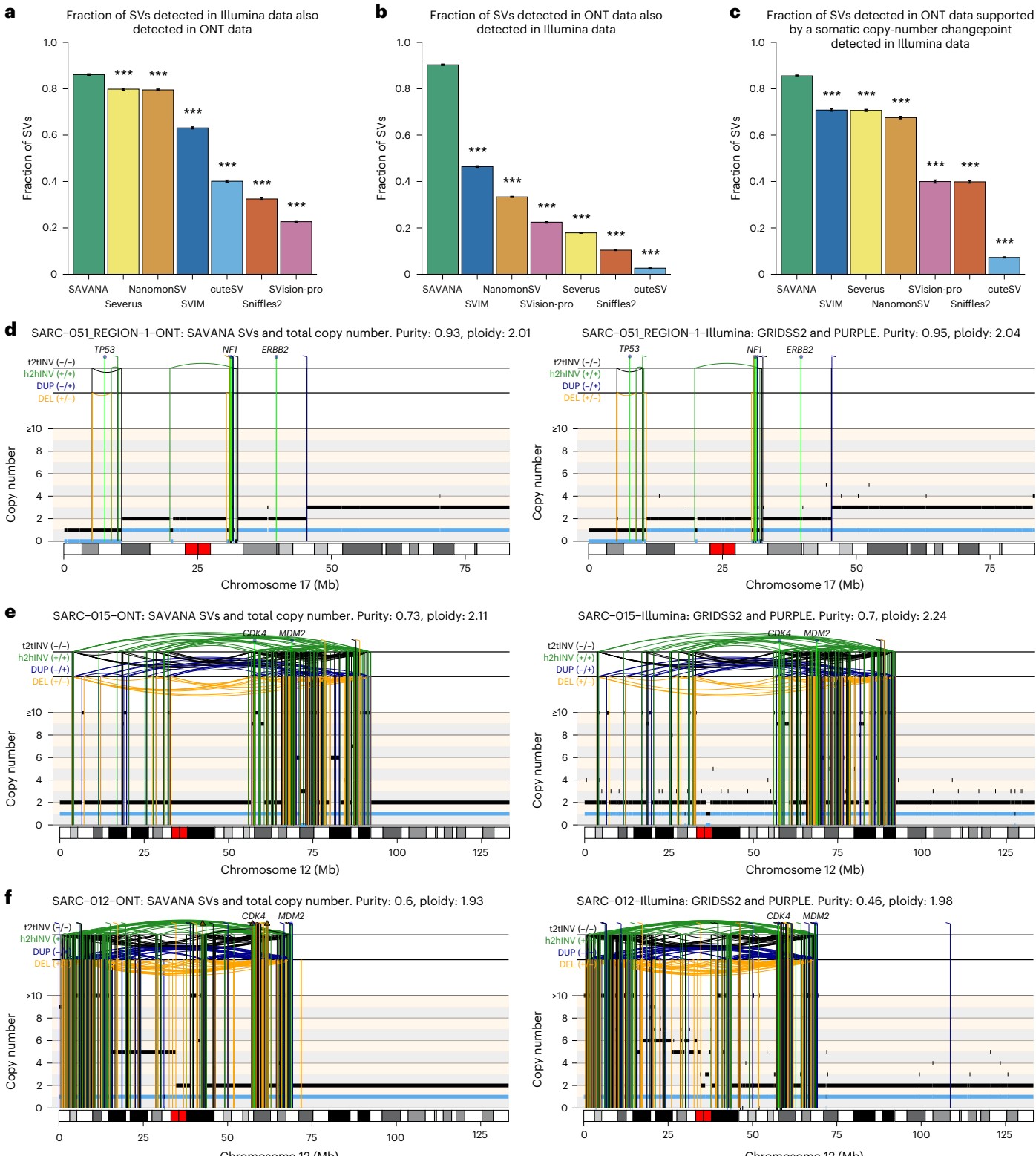

**Fig. 5 | Comparison between short and long-read data for the analysis of SVs and SCNAs. a**, The fraction of high-quality somatic SVs detected in Illumina WGS data using GRIDSS and PURPLE that are also detected in ONT data by the algorithms benchmarked. The fractions shown were computed by aggregating the somatic SV calls detected in all tumors in the cohort. Only samples with at least 30× tumor coverage in ONT WGS data were included (n = 83). **b**, Fraction of somatic SVs detected in ONT WGS data by each of the algorithms benchmarked that were also present in Illumina WGS data. **c**, Fraction of somatic SVs larger than 1,000 bp detected in ONT WGS data by each of the algorithms benchmarked that mapped within 500 bp of a somatic copy number changepoint detected in

Illumina WGS data using PURPLE. The significance in **a**–**c** was assessed using the two-sided Wilcoxon's rank test (***P < 0.001). The P values for the comparison between SAVANA against all other algorithms were P < 2.2 × 10⁻¹⁶. **d**–**f**, Examples of somatic SV and SCNA profiles of increasing complexity detected in long-read nanopore WGS data using SAVANA (left) and in Illumina WGS data using GRIDSS2 and PURPLE (right) for tumors SARC-051 (**d**), SARC-015 (**e**) and SARC-012 (**f**). The total and minor allele copy number data are represented in black and blue, respectively. DEL, deletion-like rearrangement; DUP, duplication-like rearrangement; h2hINV, head-to-head inversion; t2tINV, tail-to-tail inversion. The lines with arrowheads mark insertions.

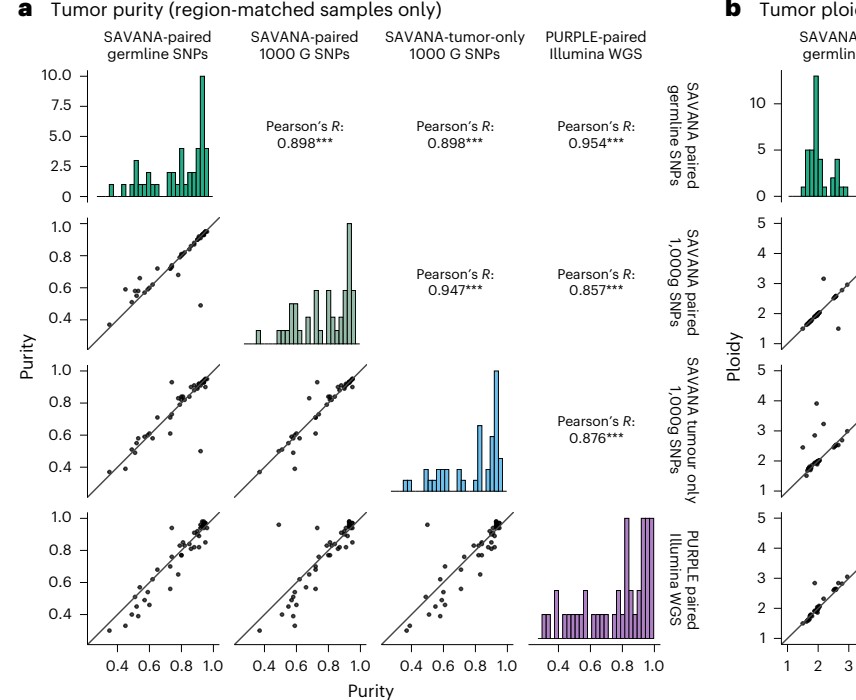

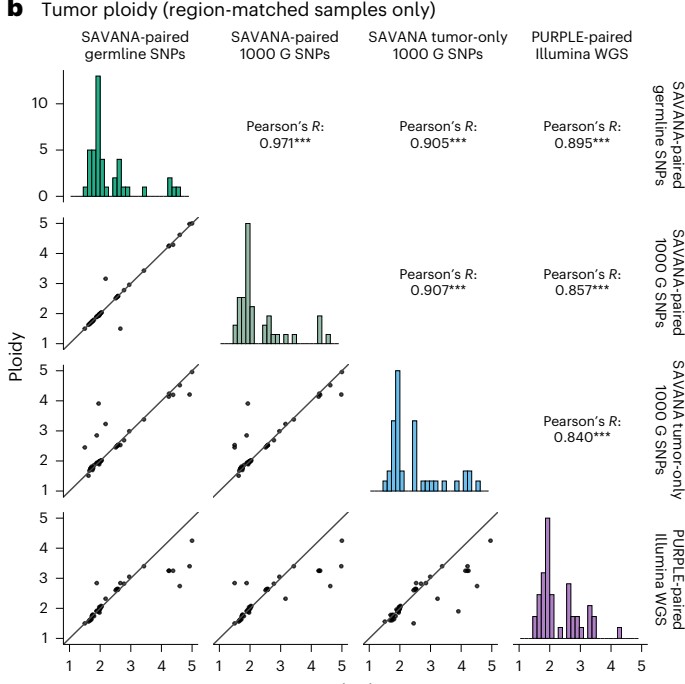

**Fig. 6 | Comparison of the tumor purity and ploidy estimates computed using different SAVANA modes and ONT WGS data against PURPLE and Illumina WGS data. a**, Tumor purity estimates. **b**, Tumor ploidy estimates. Panels labelled as 'SAVANA-paired germline SNPs' show results for matched tumor-normal pair analysis using matched normal germline SNPs for purity estimation. Panels labelled as 'SAVANA-paired 1000 G SNPs' show results for matched tumor-normal pair analysis using the 1000 Genome Project population SNPs with allele fractions >0.25 and allele fractions <0.75. Panels labelled as 'The SAVANA tumor-only 1000 G SNPs' show results for analysis using the

1000 Genome Project population SNPs with allele fractions >0.25 and allele fractions <0.75. For this analysis, we only considered the 44 tumors with region-matched nanopore and Illumina WGS data. ***$P$ < 0.0001. The $P$ values for tumor purity comparisons shown in **a** were $P < 2.2 \times 10^{-16}$, $P = 3 \times 10^{-16}$, $P < 2.2 \times 10^{-16}$, $P < 2.2 \times 10^{-16}$, $P = 1.1 \times 10^{-13}$ and $P = 1.5 \times 10^{-14}$, respectively. The $P$ values for tumor ploidy comparisons depicted in **b** were $P < 2.2 \times 10^{-16}$, $P < 2.2 \times 10^{-16}$, $P = 2.6 \times 10^{-16}$, $P < 2.2 \times 10^{-16}$, $P = 1.1 \times 10^{-13}$ and $P < = 1.9 \times 10^{-12}$, respectively (reading from top to bottom, left to right).

A major advantage of long-read sequencing is the possibility to phase SVs[15]. To enable haplotype-aware analysis of SVs, SAVANA provides functionalities to assign SVs to individual haplotypes if the sequencing reads provided as input are phased. Subsequently, the haplotype-resolved SV calls generated by SAVANA permit the application of genome assembly tools to reconstruct derivative chromosomes resulting from complex SVs (Supplementary Fig. 36).

In addition to somatic SV calling, SAVANA integrates read depth and BAF information to infer tumor purity, ploidy and allele-specific SCNAs. The tumor purity and ploidy values estimated by SAVANA on long-read data and PURPLE/GRIDSS2 (ref. 54) using matched short-read data were strongly correlated (Pearson's correlation coefficient of 0.97 and 0.9, respectively) (Fig. 6). These results are noteworthy given the higher than twofold difference in median sequencing depth between the Illumina and ONT data sets (118× versus 51×, respectively) and that, as opposed to PURPLE, SAVANA does not use the AF values of somatic point mutations to score purity and ploidy combinations.

The availability of a matched normal sample is often limited in clinical settings. To address this challenge, SAVANA permits the detection of somatic SVs and SCNAs without a matched germline control. Specifically, in tumor-only mode, SAVANA can detect somatic SVs using machine learning, which can be further filtered using panels of normals to remove likely germline SVs (Methods). In addition, the tumor-only mode of SAVANA permits the inference of allele-specific copy number profiles, tumor purity and tumor ploidy with comparable accuracy to both short-read data and SAVANA when run on long-read data using a matched germline control (Fig. 6). Notably, SAVANA in tumor-only mode showed comparable recall of SVs detected in Illumina WGS data, including those affecting cancer driver genes (Supplementary Fig. 37).

Overall, most SVs and SCNAs detectable by state-of-the-art short-read sequencing data analysis pipelines can be detected in long-read data using SAVANA, including the inactivation of tumor suppressor genes and the amplification of oncogenes, even in tumor-only mode.

## Discussion

Here, we show that SAVANA enables the integrative analysis of somatic SVs, SCNAs, tumor purity and ploidy using long-read sequencing data. SAVANA shows significantly higher sensitivity and specificity than existing methods across a dynamic range of clonality levels, SV sizes and SV types. This is critical for the analysis of clinical samples given that tumor cellularity is often low in diverse cancer types. In practice, this means that driver mutations might be missed if sensitivity for low-AF SVs is low, even when sequencing is performed to high depth. Consequently, SAVANA enables enhanced detection and analysis of SVs and more robust interpretation of the underlying tumor biology, which will facilitate the reliable application of long-read sequencing to study and detect clinically relevant rearrangements in human tumor samples.

Our genome-wide benchmarking analyses revealed that existing SV detection methods yield hundreds to thousands of false-positive somatic SV calls per tumor. Thus, these results suggest that the high rates of somatic SVs previously reported for cancer genomes using long-read sequencing do not correspond to true biological signal. To facilitate reliable and consistent comparison of algorithms in the future, we establish best practices for benchmarking somatic SV detection methods using replicates that allow quantification of both the sensitivity and specificity of algorithms across the entire genome in an unbiased manner.

Because the diversity of the training data determines the performance of any machine learning model, it is essential to provide well-calibrated confidence metrics for each prediction to increase the reliability and adoption of algorithms based on predictive modeling. To this end, SAVANA relies on MCP[34,55], which permits detection of instances falling outside the applicability domain of the model and provides well-calibrated metrics to quantify prediction confidence, an approach that has proven transformative in other domains[36,56,57]. Here, we have illustrated how MCP can be used to control the error rate for the somatic SV category. However, MCP could be further extended to control the error rate for e.g., different SV types or genomic regions, or integrated with other algorithms, such as neural networks[58]. Finally, the fact that SAVANA uses non-identifiable predictive features for training MCP models opens avenues for improving model performance by integrating genomics data in a federated manner. For example, MCP model training could be distributed without the need for sharing raw sequencing data, thus facilitating data integration while also ensuring compliance with local data privacy regulations and without comprising proprietary data[59].

## Online content

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

## Methods

### SAVANA algorithm

SAVANA consists of the following steps, provided below.

**Discovery of putative breakpoints.** SAVANA uses aligned sequencing reads from tumor and normal BAM files to identify breakpoints. Specifically, SAVANA splits the genome into non-overlapping bins of 100 million base pairs by default, and the alignments from each bin are examined in parallel. Thus, the analysis of large chromosomes is distributed across multiple processes and small chromosomes are analyzed in a single process. The primary and supplementary alignments with a mapping quality over the minimum threshold of 5 are considered for further analysis. To ensure that all potential germline SVs are identified and used to discount germline SVs as somatic, a more lenient mapping quality filter is applied to the normal reads. Specifically, the minimum mapping quality threshold is lowered by a fraction of 0.5 for alignments originating from the normal sample.

For each sequencing read, SAVANA parses supplementary alignments from the supplementary alignments tag and creates breakpoint objects storing the location, orientation, phasing and read name information. To remove recurrent artifactual fold-back-inversion-like events observed in nanopore data, by default, SAVANA does not track putative inversions in which the end of an alignment maps within 200 bp of the end of the next alignment, as this is indicative of an artifactual fold-back-like inversion (Supplementary Fig. 1a,b). If the single breakend option is enabled, supplementary alignments with low mapping quality (default <5) are considered unmapped, and their sequence and location are recorded as potential single breakends.

SAVANA parses the CIGAR string for both primary and supplementary alignments, tracking deletion and insertion operations greater than the minimum SV length (30 bp by default) and inserted bases in the case of insertion operations. Large deletions can be interspersed with short regions of mapped sequence. To account for this phenomenon, adjacent deletions are merged and reported as a single putative event when interspersed by mapped sequences shorter than 30 bp. Soft-clip operations in the CIGAR string greater than 1,000 bp (by default) without a corresponding supplementary alignment are also tracked at this stage, as well as the soft-clipped bases. To more accurately remove germline SVs from the tumor SV set, the minimum SV length to track an insertion or deletion is reduced by a fraction of 0.20 for the normal BAM file. This step prevents reporting as somatic those germline SVs falling below the minimum length cutoff used to analyze the tumor reads.

In addition, while iterating through aligned reads to identify putative breakpoints, a coverage array is built to track the count of overlapping primary and supplementary alignments, separated by their haplotype. For each haplotype (including none for unphased reads), an array of zeros is created for each contig, with an element for each bin across the genome. The value of the array position corresponding to the bin overlapped by an alignment is then incremented by 1 for the corresponding haplotype and contig array. This array is later used to annotate the depth of sequencing at breakpoints by accessing the number of overlapping reads from each haplotype, as well as the sum of these. The bin size is defaulted to five bases, which can be increased by the user to reduce memory usage and run time.

Putative SVs are recorded using breakend notation following the VCF specifications. In addition, SVs are annotated using the notation scheme established by the Pan-Cancer Analysis of Whole Genomes project[1]: (+ −) for deletion-like SVs, (− +) for duplication-like SVs, (+ +) and (− −) for inversion-like SVs and INS for insertions. Putative deletions derived from the CIGAR string are reported as '+ −' to ensure that supplementary and gapped alignments supporting the same deletion event are grouped together. Single breakends are represented as SBND.

**Clustering of putative breakpoints.** Once information about individual putative breakpoints is collected, adjacent breakpoints are clustered together if they fall within a clustering window. By default, this clustering window is 10 bp for all breakpoint types except insertions. We use a clustering window of 250 bp for insertions because we frequently observed that insertions typically have more variance in their mapping location than other breakpoint types. The reads supporting each breakpoint are split into groups based on the contig of mate breakpoints. A final clustering step is performed on the reads supporting each mate breakpoint using a larger clustering window (default of 50 bp) to prevent supporting reads from being incorrectly split into multiple events. Insertion events are clustered by their insert size to ensure that grouped reads support insertions of comparable size. Finally, grouped events are split by their respective breakpoint notation, and their supporting information and location in the genome are stored in a dictionary. Single breakends are reported if they are the only type of event present at a given breakpoint cluster. This is required because the detection of single breakends adjacent to other types of events might indicate that some of the sequencing reads in the cluster were long enough to permit complete reconstruction of the underlying SV, whereas other reads only allowed reliable mapping of one side of the new adjacency, thus resulting in single breakends. For example, an insertion of a repetitive sequence might be fully spanned by some sequencing reads, thus allowing complete reconstruction of the event. However, other reads might span only a small fraction of the inserted sequence, thus hampering unambiguous mapping of the inserted sequence, which would be detected as a single breakend.

**Filtering of false positives using machine learning.** Breakpoints are encoded using 70 features, including the average MAPQ of supporting reads, the mean, standard deviation and median SV length, the standard deviation of the start and end mapping locations of supporting reads (see Supplementary Table 2 for a description of all breakpoint features considered). The sequencing depth in the tumor and normal sample before, at and after each breakpoint is also computed by summing the coverage array from the first base of the contig to the breakpoint start coordinate as described above. If the input BAMs provided are phased, the haplotype of supporting reads and their phase sets are also reported in the VCF output by SAVANA.

To differentiate clusters of noise or sequencing artifacts from true somatic or germline events, we trained a RF classifier using Scikit-Learn[60]. To this aim, we annotated all breakpoints detected by SAVANA in the long-read data sets as true somatic SVs if they were also detected as somatic SVs in matched Illumina data or as noise otherwise. To establish a high-confidence set of SVs for model training, we excluded breakpoints that were not found in Illumina but were, however, supported by at least six tumor reads in the ONT data, had a tumor allele fraction greater than 10%, and mean supporting read MAPQ of at least 50. In addition, for SVs of length greater than 100 bp, we required that the standard deviation of the breakpoint coordinates was less than 15, and for SVs smaller than 100 bp, we required the standard deviation of the breakpoint coordinates to be smaller 1.5 and the standard deviation of the estimated SV sizes to be smaller than 10 bp. In paired mode, we required that SVs had fewer than two reads supporting any size or type of breakpoint in the matched normal sample and at least five normal reads at the location of the breakpoint in the matched normal sample.

The features encoding breakpoints (Supplementary Table 2) are read in from raw VCF files and transformed into a Pandas dataframe[61]. The breakpoints are then split into training and test sets with a 4:1 ratio. We used default parameter values to train the RF models except for the maximum depth, which was set to 20. The optimal value for the depth was determined using cross validation and grid search using a range of 10–25 for the maximum depth. To detect somatic SVs on each tumor, we trained an RF model using the labeled SVs from all other tumors to ensure that no data from the tumor to be analyzed was used during model training.

Once the RF model is trained, the raw SAVANA VCF file is labeled with an additional CLASS variable in the INFO column that denotes the model prediction. The predictions of the model are used to set the FILTER column to 'PASS' if the breakpoint is predicted to be somatic or 'LIKELY_NOISE' otherwise. The model predictions are vetted by additional thresholds on the minimum allele fraction and number of supporting reads, and their filter column is updated to 'LOW_SUPPORT' or 'LOW_AF' if they do not meet minimum thresholds of 0.01 for allele fraction and three supporting reads by default.

**Rescuing of true somatic SVs.** In addition, SAVANA can optionally utilize copy number changepoints to rescue breakpoints not classified as somatic by the machine learning model. To this end, the breakpoints are rescued by SAVANA if they are reported within at least 50 bp of a somatic copy number changepoint, have at least three supporting tumor reads, no supporting reads in the matched normal sample and fewer than three normal reads supporting a breakpoint of any type clustering to the same genomic location.

**Reliability estimation of SV calls using conformal prediction.** Mondrian conformal predictors were implemented as previously described[36,37]. In brief, the training data set was randomly split into two non-overlapping sets: 70% of the datapoints were used to train an RF model, which was applied on the remainder. The fraction of trees voting for each class was used as the nonconformity measure. We generated one list of nonconformity measures for each class, which were then used to compute the $P$ values using a predefined confidence level. The $P$ values were then used to assign breakpoints to one of the following categories: somatic, likely noise, 'both' or 'null'.

**Removal of germline SVs in tumor-only mode using a panel of normals.** SAVANA provides functionalities to detect somatic SVs in the absence of a matched normal sample. However, without a matched germline control, the germline SVs might be incorrectly classified as somatic. Thus, we classified as germline those SVs detected by SAVANA mapping within 100 bp of germline SVs in the gnomadSV v4.1 (ref. [62]) with a population AF >0.10. In addition, we removed SVs found in the Hartwig Medical Foundation SV panel of normals v5.34 (ref. [63]).

### Detection of SCNAs

SAVANA provides functionalities for the detection of total and allele-specific copy number aberrations and the estimation of tumor purity and ploidy.

**Genome-wide read-depth computation.** First, SAVANA generates non-overlapping bins of 10 kbp across the reference genome. In addition, SAVANA further segments the genome using somatic SV breakpoint information when provided. Each bin is annotated with the percentage of non-N bases and its overlap with blacklisted regions when provided. Next, SAVANA counts the number of tumor and normal sequencing reads mapping to each bin across the genome, excluding bins with more than 5% overlap with blacklisted regions and bins encompassing more than 75% non-N bases in the reference genome. Secondary and supplementary read alignments, as well as reads with a mapping quality <5, are not considered for this analysis. The bins that do not have any read counts are filtered and removed. The binned tumor read counts are then normalized using normal counts from the matched normal sample while accounting for differences in read depth between tumor and matched normal BAM files. Alternatively, if no matched normal BAM is provided, the binned tumor read counts are self-normalized using the median of read counts across all bins. Subsequently, the binned read counts are $\log_2$ transformed providing relative $\log_2$ copy number ratio values ($\log_2 R$). Next, single point outliers in the $\log_2 R$ read counts are smoothened, as previously described[64], to reduce noise before segmentation of the $\log_2 R$ data. Next, $\log_2 R$ read counts are Anscombe transformed to stabilize variance

and subsequently segmented using circular binary segmentation, an algorithm for finding changepoints in sequential data[65], as follows. First, copy number changepoints are identified using 1,000 permutations setting the threshold for statistical significance to 0.05. The changepoints are then validated using 1,000 permutations and a threshold for statistical significance of 0.01. To handle potential oversegmentation, the resulting genomic segments are subsequently merged if their $\log_2 R$ difference is smaller than the 20th percentile of the distribution defined by the $\log_2 R$ difference between all pairs of segments.

**Inference of tumor purity.** To infer tumor purity, SAVANA computes the BAF for all germline heterozygous SNPs in the genome. Alternatively, if germline heterozygous SNPs are not available (for example tumor-only mode), a curated set of SNPs from the 1000 Genomes Project is used[30]. Next, SAVANA divides the genome in blocks of equal size to improve computational speed and searches for genomic blocks showing evidence of LOH. To this end, SAVANA computes the bimodality coefficient of the BAF distribution for each genomic block with at least ten heterozygous SNPs. The genomic blocks with read depths larger than twice the mean read depth are excluded, as potentially amplified or gained regions would bias the BAF distribution. By default, SAVANA uses a block size of 1.2 Mbp, which corresponds to the median size of phased blocks obtained for the set of matched normal samples reported in this study. However, the block size can be modified by the user. Subsequently, the median BAF values are computed for the top ten genomic blocks with the highest bimodality coefficient value for each of the parental alleles ($BAF_A$ and $BAF_B$), and the tumor purity is estimated using the following equation:

$$\text{Purity}_{\text{tumour}}$$

$$= \frac{1}{10} \sum_{i=1}^{10} \frac{\left(1 - 2 \times \left(1 - \text{median BAF}_{A_i}\right)\right) + \left(1 - 2 \times \left(\text{median BAF}_{B_i}\right)\right)}{2}.$$

**Inference of tumor ploidy.** Once the tumor purity ($\text{Purity}_{\text{tumor}}$) is estimated, SAVANA infers tumor ploidy using grid search, as previously described[66], encompassing a purity search space of purity $\in \{\text{Purity}_{\text{tumor}} - 0.1, ..., \text{Purity}_{\text{tumor}} + 0.1\}$ with a step size of 0.01, and a ploidy search space of ploidy $\in \{1.50, 1.51, 1.52, ..., 5.00\}$. Each purity–ploidy combination is subsequently assessed for its goodness-of-fit by computing the root mean squared deviation or mean absolute deviation weighted by the segment size between the inferred absolute copy number and the nearest integer values. In addition, for each purity–ploidy combination, SAVANA evaluates whether the proportion of copy number states fitted to zero is <0.1, the proportion of copy number segments that are considered close to the next nearest integer is >0.5, and the copy number change step size between the two most frequent copy number states is <2. Purity–ploidy combinations that do not pass these criteria are discarded. The purity–ploidy combination with the best goodness-of-fit (that is, the lowest root mean squared deviation or mean absolute deviation value) is finally used to compute the absolute copy number for each segment using the observed log2R data as follows[66]

$$\text{Absolute}_{\text{copy number}} = \text{ploidy} + (2^{\log_2 R} - 1) \times \left(\text{ploidy} + \left(\frac{2}{\text{Purity}_{\text{tumor}}}\right) - 2\right).$$

**Allele-specific copy number fitting.** Finally, the optional tumor purity and ploidy values estimated in the previous step are used to compute the allele-specific copy number profile for the tumor as previously described[67].

### Selection of human sarcoma samples and matched germline controls

Fresh-frozen bone and soft-tissue sarcoma samples were obtained from patients consented and enrolled in both the Genomics England

100,000 Genomes Project as well as the Royal National Orthopaedic Hospital (RNOH) Biobank, satellite of the UCL/UCLH Biobank for Health and Disease (REC reference 20/YH/0088). Patients did not receive financial compensation for donating samples. Surplus tumor tissue from resection and/or biopsy samples were collected and frozen as part of routine clinical practice. The matched blood samples were used for germline sequencing. The processing of tissue samples for pathology review and molecular analyses was performed at the RNOH for the 100,000 Genomes Project, founded by England's National Health Service in 2012, as previously described[68]. Fresh-frozen tissue sections (hematoxylin and eosin, 5 μm) were used to guide the selection of the most viable areas for each tumor specimen in terms of lack of necrosis and tumor cellularity. The DNA was extracted from matched tumor and blood samples using established protocols and in accordance with the 100,000 Genomes Project guidelines. The DNA was sent for centralized library preparation and sequencing at the Illumina Laboratory Services in Cambridge, UK[69]. Sequencing was performed as part of the 100,000 Genomes Project. The DNA from tumor and normal samples was sequenced to an average depth of 116x (median 118x) and 42x (median 36x), respectively. The dual consented somatic and germline genomic data were shared with RNOH and EMBL-EBI. The data linked to local clinical data (no National Health Service (NHS) Digital or NHS England data) were shared by Genomics England with RNOH and EMBL-EBI. No analysis was undertaken in the Genomics England Research Environment or National Genomic Research Library.

## Selection of glioblastoma tumor samples and matched germline controls

Glioblastoma and matched blood samples were collected in the Neurosurgery Department at Centro Hospitalar Universitário Lisboa Norte (CHULN) and stored less than 1 h after surgery at Biobanco-iMM CAML (Lisbon Academic Medical Center). Ethical approval was obtained from the Ethics Committee of CHULN (ref. no. 367/18). Written informed consent was obtained from all patients before study participation in accordance with the European and National Ethical Regulation (law 12/2005). The patients did not receive financial compensation for donating samples.

## DNA extraction and processing for nanopore sequencing

Tumor genomic DNA (gDNA) was extracted from tissue samples using the Nanobind tissue kit (PacBio SKU 102-302-100) and Germline DNA from blood samples using the Nanobind CBB kit (PacBio, SKU 102-301-900). After extraction, the gDNA was homogenized by three to ten passes of needle shearing (26G) and 1 h of incubation at 50 °C. All the samples were quantified on a Qubit fluorometer (Invitrogen, Q33226) with the Qubit BR dsDNA assay (Thermo Fisher Scientific, Q32853), and manual volume checks were performed. The DNA size distributions were assessed at each relevant step by capillary pulse-field electrophoresis with the FemtoPulse system (Agilent, M5330AA and FP-1002-0275). A total of 4–10 μg of gDNA were either fragmented to a size of 10–50 kb with the Megaruptor 3 (Diagenode, E07010001 and E07010003) or fragmented to a size of 15–25 kb with a gTUBE (Covaris, 520079) centrifuged at 1,500g. All the samples were depleted of short DNA fragments (less than 10 kb long) with the SRE kit (Circulomics, SS-100-101-01, now PacBio) or the SRE XS kit (Circulomics SS-100-121-01, now PacBio) when sample availability or concentration was limiting. To generate matched Illumina and nanopore tumor WGS data, the DNA aliquots were tumor matched. The cases for which multiregion nanopore and Illumina tumor sequencing was performed, DNA aliquots were tumor and region matched.

## Library preparation and nanopore WGS

The sequencing libraries were generated from 600 ng up to 1.8 μg of gDNA, with one library prepared for germline samples and two libraries prepared for tumor samples with the SQK-LSK110 or SQK-LSK114 kits (Oxford Nanopore Technologies, ONT), according to manufacturer's recommendations with minor modifications. Briefly, the samples were end-repaired by adding 2 μl NEBNext FFPE DNA Repair Mix (NEB, M6630) and 3 μl NEBNext Ultra II End Prep Enzyme Mix (NEB, E7546), incubated for 10 min at room temperature followed by 10 min at 65 °C, then cleaned up with 1× AMPure XP beads (Beckman Coulter, A63880) and eluted in 60 μl of Elution Buffer. The end-repaired DNA was ligated with 5 μl Adapter Mix (ONT, SQK-LSK110) using 8 μl NEBNext Quick T4 DNA ligase (NEB, E6056) at 21 °C for up to 1 h. The adapter-ligated DNA was cleaned up by adding a 0.4× volume of AMPure XP beads. The sequencing libraries were quantified using the average peak size of the samples determined after sample preparation on a Femto Pulse system. A total of 20 fmol of the obtained sequencing libraries were loaded onto R9.4 flow cells (ONT), or 10 fmol of libraries were loaded onto R10 flow cells and sequenced on a PromethION48 (ONT). The libraries were stored overnight in the fridge. After 24 h, the sequencing runs were paused and a DNAse treatment or nuclease flush (ONT, WSH-003) was performed, and 20 fmol of the libraries were reloaded on the flow cells.

## Nanopore sequencing data analysis

Base-calling was performed using the high accuracy model of Guppy-4.0.11 and a qscore filter of 7. The sequencing reads were aligned to GRCh38 and chm13v2.0 (T2T) using minimap2 (v2.24)(68) with parameters '-ax map-ont-MD'. Quality control statistics and plots were generated using cramino[70] v0.14.5.

## Analysis of artifactual fold-back-like inversions

Fold-back-like inversion artifacts were detected by the presence of reads characterized by having two nearly identical alignments in forward and reverse orientations. As a result, the first alignment starts where the second alignment ends, and both alignments are of equal length (Supplementary Fig. 1). To estimate the number of fold-back-like inversion artifacts per sample, we analyzed all the aligned reads per sample and classified them as a fold-back-like inversion artifact when the following criteria were met: (1) the read had exactly one primary and one supplementary alignment, both with a minimum mapping quality of 20; (2) the primary and supplementary alignments overlapped but were mapped in opposite orientations; and (3) the distance between the alignment positions corresponding to the start and end of the read were less than 150 bp apart from each other. To calculate the rate of fold-back-like inversion artifacts per sample, we determined the total number of reads by counting those reads with a primary alignment and a minimum mapping quality of 20. The code used for this analysis can be accessed at https://github.com/cortes-ciriano-lab/ont_fb-inv_artefacts. By default, SAVANA discards fold-back-like inversion artifacts for SV calling.

## Detection of SVs using existing algorithms

Sniffles2 (v2.2.0) was run with '--output-rnames'. cuteSV (v2.1.0) was run with recommended parameters: '--max_cluster_bias_INS 100', '--diff_ratio_merging_INS 0.3', '--max_cluster_bias_DEL 100' and '--diff_ratio_merging_DEL 0.3'. SVIM (v1.4.2) was run in 'alignment' mode as prealigned BAM files were used. Sniffles2, cuteSV and SVIM were run separately on tumor and normal BAM files, all with a minimum SV length of 32 bp. To filter out germline SVs, the coordinates of germline SVs for each tool were extracted from the corresponding germline VCF. The start and end coordinates were extended by 1,000 bp and saved to a file in browser extensible data (BED) format. A bedtools[71] subtraction of this file from the tumor VCF was performed. To this aim, we required a fraction of at least 0.01 of the tumor SV to be overlapped by the extended germline SV coordinates for it to be removed. All germline SVs detected in each matched normal sample by the SV caller being evaluated were considered for this filtering step, and only tumor SVs with three or more supporting reads were kept. NanomonSV (v0.5.0) was run with the recommended, '--use_racon' flag and '--single_bnd', as well as a control panel provided by the tool developers to reduce false

# Article

positives (available at https://zenodo.org/records/7017953). Severus (v1.0) was run using haplotyped BAM files, as recommended by the tool. SVision-pro (v2.1) was run in somatic mode with the default image size of 256 and the corresponding model. In all cases, the minimum SV length was set to 32 bp and SVs mapping to alternative contigs, chromosome M, unplaced contigs and the Epstein–Barr virus were removed. SVs detected in Illumina and ONT data were considered to support the same event if the underlying breakpoints mapped within 100 bp of each other.

## Benchmarking SV detection algorithms

The breakpoint coordinates for the SVs in the COLO829 SV truth set from ref. 38 were lifted over to GRCh38. The SV truth set was downloaded from zenodo.org/records/4716169#.YL4yTJozYUE (truthset_somaticSVs_COLO829_hg38lifted.vcf), which was filtered to contain only breakpoints that were detected by the authors in ONT WGS data. The VCF file was curated to report insertions in a single entry.

SAVANA includes a module, termed evaluate, to compare VCF files. Specifically, the evaluate module compares breakends and reports them as matched when they are within an overlap buffer (100 bp by default). The VCF files generated by the germline SV detection algorithms benchmarked were compared with the COLO829 somatic SV truth set VCF. Specifically, we assessed whether breakends in the truth set mapped within 100 bp of the breakends reported by the algorithms benchmarked. These results were then used to compute the precision, recall and $F$-measure for each algorithm. The evaluate module in SAVANA accounts for both breakend format VCFs, where each breakend of an SV has a line in the VCF, and for VCF formats, where both breakends are reported in the same line. We used the same criteria to benchmark the performance of SV detection methods using replicates.

## Detection of false-positive SVs using read-backed phasing

We performed read-backed phasing of reads supporting somatic SV calls to assess whether the SVs are supported by reads assigned to a single haplotype (as it would be expected for true positive SVs) or by reads from both alleles (which would be consistent with false-positive SVs). Specifically, we computed the fraction of SV calls generated by each algorithm that are supported by sequencing reads assigned to only a single parental allele. The sequencing reads were assigned to either parental allele based on read-level phasing using WhatsHap[72]. For this analysis, we excluded genomic regions with LOH, as even false-positive SV calls mapping to genomic regions with LOH would be supported by reads from a single parental allele. Given that phasing is not always accurate due to, for example, sequencing errors at SNP loci or limited read length, it is possible that a small subset of reads might be assigned to the incorrect parental allele. As a result, some of the reads supporting a true SV might show discordant phasing. Therefore, we performed a Binomial test (using a probability of success of 0.95 and with a one-tailed alternative hypothesis) on each SV with supporting reads from both alleles and applied an FDR-corrected $P$ value of 0.05 as the threshold for statistical significance. The SVs with a significant number of discordant reads were considered to have discordant read support. The SVs for which we do not have enough statistical power to reject the null hypothesis (that is, that a given SV is supported by reads from one allele only) were considered inconclusive.

## Assembly of derivative chromosomes at single-haplotype resolution using SAVANA SVs

For each phase set defined by WhatsHap[72], SV-supporting reads identified by SAVANA were used as input to generate an assembly using wtdbg2 (ref. 73) and subsequently polished using Racon[74].

## Processing of short-read WGS data

The sequencing reads were mapped to the GRCh38 build of the human reference genome using BWA-MEM[75] v0.7.17-r1188. The aligned reads were processed following the Genome Analysis Toolkit (GATK, v4.1.8.0) Best Practices workflow to remove duplicates and recalibrate base quality scores[76]. NGSCheckMate (v1.0.1) was utilized with default options to verify that sequencing data from tumor-normal pairs were properly matched[77]. The somatic SVs were detected using GRIDSS[33] (v2.12.0, https://github.com/PapenfussLab/gridss), which we ran using default parameter values. The BAF information for heterozygous SNP sites was collected using AMBER (v3.5), and the read-depth ratios for the reference and alternate alleles were computed using COBALT (v1.11). The BAF and read-depth ratios for heterozygous SNPs were used as input to PURPLE[54] (v2.54) to estimate the purity, ploidy and SCNAs of the tumor samples. The MSI was assessed using PURPLE. AMBER, COBALT and PURPLE are developed by the Hartwig Medical Foundation and are freely available via GitHub at https://github.com/hartwigmedical/hmftools. The implementation of the WGS data analysis pipeline used in this study is available at https://github.com/cortes-ciriano-lab/osteosarcoma_evolution.

## Statistical analysis and visualization

All statistical analyses were performed using R v4.2.2. The level of significance for all statistical analyses was set at 0.05 unless otherwise stated. No statistical method was used to predetermine sample size. The rearrangement and copy number profiles were visualized using the R package ReConPlot[78] v0.1. The structural variants were read into R data frames using the R package StructuralVariantAnnotation[79] v1.14.1.

## Reporting summary

Further information on research design is available in the Nature Portfolio Reporting Summary linked to this article.

# Data availability

WGS data from the participants enrolled in the 100,000 Genomes Project can be accessed via Genomics England Limited following the procedure described at https://www.genomicsengland.co.uk/about-gecip/joining-research-community/. In brief, the applicants from registered institutions can apply to join one of the Genomics England Research Networks and then register a project. Access to the Genomics England Research Environment is then granted after completing online training. The long-read nanopore WGS data generated in this study for sarcoma samples and matched blood samples are available under controlled access at EGA under the accession number EGAS50000000651. The short and long-read sequencing data from the glioblastoma samples are available under controlled access at EGA under the accession number EGAD00001012101. Data access can be granted via the EGA for a defined time period after successful completion of a data access agreement provided by the WTSI CGP Data access committee (datasharing@sanger.ac.uk). ONT sequencing data for the COLO829 and COLO829BL cell lines[38] using R9.4 MinION/GridION flow cells were downloaded from the European Nucleotide Archive (project ID PRJEB27698). The nanopore WGS data generated by ONT (Oxford, UK) for the cell lines COLO829 and COLO829BL using the Ligation Sequencing Kit v14 were downloaded from the Amazon Web Services S3 bucket s3://ont-open-data/colo_2023.04/. Finally, PacBio HiFi sequencing data for COLO829 and COLO829BL generated using the Revio system were downloaded from https://downloads.pacbcloud.com/public/revio/2023Q2/COLO829/. The reference genome build GRCh38 was downloaded from https://hgdownload.soe.ucsc.edu/downloads.html. The genome assembly T2T-CHM13v2.0 was downloaded from https://www.ncbi.nlm.nih.gov/datasets/genome/GCF_009914755.1/.

# Code availability

The code for SAVANA is available via GitHub at https://github.com/cortes-ciriano-lab/savana.

The code for the benchmarking analysis is available via GitHub at https://github.com/cortes-ciriano-lab/savana_manuscript_benchmarking_code/.

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

## Acknowledgements

H.E., C.M.S., J.E.V.-I., S.Z., F.M. and I.C.-C. thank EMBL for funding. I.C.-C. thanks The Wellcome Trust for funding. C.M.S. acknowledges support and funding from the Marie Skłodowska-Curie grant (no. 101106070). This project was supported by research grants from the Sarcoma Foundation of America (grant no. SFA 20-05), the Connective Tissue Oncology Society (Basic Science Sarcoma Research Award) and NF Research Initiative at Boston Children's Hospital awarded to A.M.F. and I.C.-C. Provision of patients' samples from the RNOH was made possible through the RNOH Pathology Department and the Research and Development Department, The Tom Prince Trust, The Rosetrees Trust, Skeletal Cancer Trust, Sarcoma UK, The Bone Cancer Research Trust and The Pathological Society of Great Britain and Ireland, over the last two decades. We also acknowledge support to A.M.F. from the National Institute for Health Research, UCLH Biomedical Research Centre and the UCL Experimental Cancer Centre. SDN is funded jointly by the Pathological Society of Great Britain and Ireland and the Jean Shanks Foundation. This research was made possible through access to data in the National Genomic Research Library, which is managed by Genomics England Limited (a wholly owned company of the Department of Health and Social Care). The National Genomic Research Library holds data provided by patients and collected by the NHS as part of their care and data collected as part of their participation in research. The National Genomic Research Library is funded by the National Institute for Health Research and NHS England. The Wellcome Trust, Cancer Research UK and the Medical Research Council have also funded research infrastructure. R.C., A.A. and C.C.F. acknowledge the support of Associação David Vaz, Bolsa João Lobo Antunes—GAPIC (iMM/FMUL) and Millennium bcp. All authors thank the computational resources provided by the European Bioinformatics Institute (EMBL-EBI). The authors acknowledge the Biobanco-iMM CAML, which enabled the collection, processing and storage of tumor and blood samples from glioblastoma patients. We thank E. McCargow and H. Louis dit Picard for technical and administrative support. We thank the patients and their families for their participation in this study. We also thank all members of the Flanagan and Cortes-Ciriano laboratories for numerous discussions and feedback. Figures 1a, 2a and 3a,c were generated using BioRender.com.

## Author contributions

A.M.F. and I.C.-C. designed, supervised, administered and obtained funding for the study. H.E. implemented the SAVANA functionalities for the detection of structural variants with input from C.M.S., J.E.V.-I., S.Z., F.M. and I.C.-C. C.M.S. and I.C.-C. developed the SAVANA functionalities for somatic copy number analysis with input from H.E. H.E., C.M.S., J.E.V.-I., S.Z. and I.C.-C. performed analyses and generated the figures, with input from A.M.F. H.E., J.E.V.-I., S.Z., A.G. and I.C.-C. performed nanopore sequencing data analysis. S.D.N, F.A., R.T. and A.M.F. performed pathology review of sarcoma samples. C.C.F., R.C. and A.A. collected and processed the glioblastoma and matched blood samples. M.T. and G.E. performed nanopore sequencing. K.T., M.T., A.G.R., A.G., T.F., K.P., D.T.M., A.S., G.E., A.M.F. and I.C.-C. provided technical support. H.E., C.M.S. and I.C.-C. wrote the manuscript with input from all authors. All authors read and approved the final version of the manuscript.

## Funding

## Competing interests

H.E. and C.M.S. have received travel bursaries from ONT. The other authors declare no competing interests.

## Additional information

**Correspondence and requests for materials** should be addressed to Adrienne M. Flanagan or Isidro Cortes-Ciriano.

Dr. Isidro Cortes-Ciriano

# Reporting Summary

## Statistics

For all statistical analyses, confirm that the following items are present in the figure legend, table legend, main text, or Methods section.

| n/a | Confirmed | |
|---|---|---|
| ☐ | ☒ | The exact sample size (*n*) for each experimental group/condition, given as a discrete number and unit of measurement |
| ☐ | ☒ | A statement on whether measurements were taken from distinct samples or whether the same sample was measured repeatedly |
| ☐ | ☒ | The statistical test(s) used AND whether they are one- or two-sided<br>*Only common tests should be described solely by name; describe more complex techniques in the Methods section.* |
| ☐ | ☒ | A description of all covariates tested |
| ☐ | ☒ | A description of any assumptions or corrections, such as tests of normality and adjustment for multiple comparisons |
| ☐ | ☒ | A full description of the statistical parameters including central tendency (e.g. means) or other basic estimates (e.g. regression coefficient) AND variation (e.g. standard deviation) or associated estimates of uncertainty (e.g. confidence intervals) |
| ☐ | ☒ | For null hypothesis testing, the test statistic (e.g. *F*, *t*, *r*) with confidence intervals, effect sizes, degrees of freedom and *P* value noted<br>*Give P values as exact values whenever suitable.* |
| ☒ | ☐ | For Bayesian analysis, information on the choice of priors and Markov chain Monte Carlo settings |
| ☒ | ☐ | For hierarchical and complex designs, identification of the appropriate level for tests and full reporting of outcomes |
| ☒ | ☐ | Estimates of effect sizes (e.g. Cohen's *d*, Pearson's *r*), indicating how they were calculated |

*Our web collection on statistics for biologists contains articles on many of the points above.*

## Software and code

Policy information about availability of computer code

| | |
|---|---|
| Data collection | No specific software was used for data collection. |
| Data analysis | Structural variants were detected in long-read sequencing data sets using the following algorithms: Sniffles2 v2.2,  cuteSV v2.1.0, SVIM v1.4.2, Severus v1.0, and SVision-pro v2.1. All ONT data were aligned to GRCh38 and T2T-CHM13v2.0 reference genomes with minimap2 (v2.24) with parameters "-ax map-ont-MD". ONT reads were phased using WhatsHap v2.3 and germline heterozygous SNPs detected using strelka-2.9.10.centos6_x86_64.<br><br>Illumina WGS data sets were aligned to the reference genome using BWA-MEM v0.7.17-r1188, and processed with the Genome Analysis Toolkit (GATK, v4.1.8.0) to remove duplicates and recalibrate base quality scores. We used NGSCheckMate v1.0.1 to verify that sequencing data from tumour-normal pairs were properly matched. SVs were identified in Illumina data using GRIDSS (v2.12.0). B-allele frequency (BAF) values were calculated using AMBER (v3.5), read depth ratios for heterozygous SNPs were computed using COBALT (v1.11). The outputs of AMBER and COBALT were used as input to PURPLE (v2.54) to estimate the purity, ploidy, and copy number aberrations in tumour samples. Microsatellite instability was assessed using PURPLE. QC statistics and plots were generated using cramino v0.14.5.<br><br>All statistical analyses were performed using R version 4.2.2. The level of significance for all statistical analyses was set at 0.05. No statistical method was used to predetermine sample size. Rearrangement and copy number profiles were visualised using the R package ReConPlot68 v0.1. Structural variants were read into R data frames for visualisation purposes using the R package StructuralVariantAnnotation n v1.14. |

For manuscripts utilizing custom algorithms or software that are central to the research but not yet described in published literature, software must be made available to editors and reviewers. We strongly encourage code deposition in a community repository (e.g. GitHub). See the Nature Portfolio guidelines for submitting code & software for further information.

## Data

Policy information about [availability of data](availability of data)

All manuscripts must include a [data availability statement](data availability statement). This statement should provide the following information, where applicable:
- Accession codes, unique identifiers, or web links for publicly available datasets
- A description of any restrictions on data availability
- For clinical datasets or third party data, please ensure that the statement adheres to our [policy](policy)

WGS data from the participants enrolled in the 100,000 Genomes Project can be accessed via Genomics England Limited following the procedure described at: https://www.genomicsengland.co.uk/about-gecip/joining-research-community/. In brief, applicants from registered institutions can apply to join one of the Genomics England Research Networks, and then register a project. Access to the Genomics England Research Environment is then granted after completing online training. The short and long-read sequencing data from the glioblastoma samples are available under controlled access at EGA under the accession number EGAS50000000651. Data access can be granted via the EGA for a defined time period after successful completion of a data access agreement provided by the WTSI CGP Data access committee (datasharing@sanger.ac.uk). ONT sequencing data for the COLO829 and COLO829BL cell lines35 using R9.4 MinION/GridION flow cells were downloaded from the European Nucleotide Archive (ENA; project ID PRJEB27698). The nanopore WGS data generated by Oxford Nanopore Technologies (Oxford, UK) for the cell lines COLO829 and COLO829BL using the Ligation Sequencing Kit v14 were downloaded from the Amazon Web Services S3 bucket s3://ont-open-data/colo_2023.04/. Finally, PacBio HiFi sequencing data for COLO829 and COLO829BL generated using the Revio system were downloaded from https://downloads.pacbcloud.com/public/revio/2023Q2/COLO829/. The reference genome build GRCh38 was downloaded from https://hgdownload.soe.ucsc.edu/downloads.html. The genome assembly T2T-CHM13v2.0 was downloaded from https://www.ncbi.nlm.nih.gov/datasets/genome/GCF_009914755.1/.

## Human research participants

Policy information about [studies involving human research participants and Sex and Gender in Research.](studies involving human research participants and Sex and Gender in Research.)

| | |
|---|---|
| Reporting on sex and gender | We did not collect gender data in this study. We provide biological sex information for all samples analysed in Supplementary Table 1. Biological sex information was not used to define experimental groups or stratify samples for any of the analyses reported. |
| Population characteristics | The selection of tumour samples and data sets analysed was not guided by population information. Patient demographic information, including biological sex, age and tumour diagnosis, is provided in Supplementary Table 1. |
| Recruitment | No specific recruitment criteria were applied for the selection of tumour samples to be sequenced except for tumour quality. Specifically, fresh-frozen tissue sections [haematoxylin and eosin (H&E); 5 μm] were used to guide the selection of the most viable areas for each tumour specimen in terms of lack of necrosis and tumour cellularity. DNA was extracted from matched tumour and blood samples using established protocols and in accordance with the 100,000 Genomes Project guidelines. |
| Ethics oversight | Fresh-frozen bone and soft-tissue sarcoma samples were obtained from patients consented and enrolled in both the Genomics England 100,000 Genomes Project (G100k) as well as the Royal National Orthopaedic Hospital (RNOH) Biobank, satellite of the UCL/UCLH Biobank for Health and Disease (REC reference 20/YH/0088). Patients did not receive financial compensation for donating samples. <br><br> Glioblastoma and matched blood samples were collected in the Neurosurgery Department at Centro Hospitalar Universitário Lisboa Norte (CHULN) and stored less than 1h after surgery at Biobanco-iMM CAML (Lisbon Academic Medical Center, Lisbon, Portugal). Ethical approval was obtained from the Ethics Committee of CHULN (Ref. № 367/18). Written informed consent was obtained from all patients prior to study participation in accordance with the European and National Ethical Regulation (law 12/2005). Patients did not receive financial compensation for donating samples. |

Note that full information on the approval of the study protocol must also be provided in the manuscript.

# Field-specific reporting

Please select the one below that is the best fit for your research. If you are not sure, read the appropriate sections before making your selection.

☒ Life sciences          ☐ Behavioural & social sciences          ☐ Ecological, evolutionary & environmental sciences

For a reference copy of the document with all sections, see [nature.com/documents/nr-reporting-summary-flat.pdf](nature.com/documents/nr-reporting-summary-flat.pdf)

# Life sciences study design

All studies must disclose on these points even when the disclosure is negative.

| | |
|---|---|
| Sample size | No sample size calculations were performed. |
| Data exclusions | QC statistics and plots were generated using cramino version 0.14.5. <br> We did not exclude any data points from the published data sets analysed. |

| Replication | To assess the performance of SV detected algorithms we used replication experiments. Specifically, we used simulated sequencing replicates of human tumour nanopore whole-genome sequencing data sets, and sequencing replicates of the melanoma cell line COLO829BL. All replication analyses were successful. |
|---|---|
| Randomization | No specific allocation of samples was performed. Randomization was applied when we generated simulated seqeuncing replicates. Sepecifically, sequencing reads were assigned to each simulated sequencing replicate file randomly. |
| Blinding | The researchers involved in this study were not blinded to sample allocation across groups during data analysis as the selection of samples for sequencing was largely driven by sample availability and quality based on histophatological examination of tumour samples. |

# Reporting for specific materials, systems and methods

We require information from authors about some types of materials, experimental systems and methods used in many studies. Here, indicate whether each material, system or method listed is relevant to your study. If you are not sure if a list item applies to your research, read the appropriate section before selecting a response.

## Materials & experimental systems

| n/a | Involved in the study |
|---|---|
| ☒ | ☐ Antibodies |
| ☒ | ☐ Eukaryotic cell lines |
| ☒ | ☐ Palaeontology and archaeology |
| ☒ | ☐ Animals and other organisms |
| ☒ | ☐ Clinical data |
| ☒ | ☐ Dual use research of concern |

## Methods

| n/a | Involved in the study |
|---|---|
| ☒ | ☐ ChIP-seq |
| ☒ | ☐ Flow cytometry |
| ☒ | ☐ MRI-based neuroimaging |

