## [Peer Review File · Nature Methods]

SAVANA: reliable analysis of somatic structural variants and copy number aberrations using long-read sequencing

Corresponding Author: Dr Isidro Cortés-Ciriano

Version 0:

Decision Letter:

17th Oct 2024

Dear Dr Cortés-Ciriano,

Your Article, "SAVANA: reliable analysis of somatic structural variants and copy number aberrations in clinical samples using long-read sequencing", has now been seen by 3 reviewers. As you will see from their comments below, although the reviewers find your work of considerable potential interest, they have raised a number of concerns. We are interested in the possibility of publishing your paper in Nature Methods, but would like to consider your response to these concerns before we reach a final decision on publication.

We therefore invite you to revise your manuscript to address these concerns that should include data demonstrating the advantages of using long-read sequencing for SVs and CNAs analyses. We also suggest you to remove "clinical samples" in the title and abstract. We are committed to providing a fair and constructive peer-review process. Do not hesitate to contact us if there are specific requests from the reviewers that you believe are technically impossible or unlikely to yield a meaningful outcome.

Link Redacted

We hope to receive your revised paper within 12 weeks. If you cannot send it within this time, please let us know. In this event, we will still be happy to reconsider your paper at a later date so long as nothing similar has been accepted for publication at Nature Methods or published elsewhere.

OPEN SCIENCE REQUIREMENTS

REPORTING SUMMARY AND EDITORIAL POLICY CHECKLISTS

IMAGE INTEGRITY

DATA AVAILABILITY

All novel DNA and RNA sequencing data, protein sequences, genetic polymorphisms, linked genotype and phenotype data, gene expression data, macromolecular structures, and proteomics data must be deposited in a publicly accessible database, and accession codes and associated hyperlinks must be provided in the "Data Availability" section.

CODE AVAILABILITY

Please include a "Code Availability" subsection in the Online Methods which details how your custom code is made available. Only in rare cases (where code is not central to the main conclusions of the paper) is the statement "available upon request" allowed (and reasons should be specified).

For more information on our code sharing policy and requirements, please see:
<https://www.nature.com/nature-research/editorial-policies/reporting-standards#availability-of-computer-code>

ORCID

Sincerely,
Lei

Lei Tang, Ph.D.
Senior Editor
Nature Methods

Reviewers' Comments:

Reviewer #1 (Remarks to the Author):

The manuscript "SAVANA: reliable analysis of somatic structural variants and copy number aberrations in clinical samples using long-read sequencing" by Elrick et al. is presenting a novel algorithm for SV and SCNA detection in paired tumor-normal sequencing data.

The work is welcomed advancement in the field of structural variation calling for cancer diagnostics. Despite the availability of tools from the major sequencing platforms, there is no clearly established state-of-the-art software. The manuscript provides evidence that SAVANA outperforms a battery of software packages, designed for the same type of analysis, both in terms of sensitivity and specificity. It is a clear benefit that the algorithms of SAVANA are designed with the specifics of tumor biology in mind, for the somatic SV analysis, which is considerably harder problem than germline SV calling. The solution works with aligned sequencing data from all three major sequencing providers (Illumina, ONT and PacBio) and the source code is freely accessible under the Apache 2.0 license.

Although SAVANA interprets data from several sequencing platforms, it is reliant on their stock aligners or third-party tools, which is a potential drawback. The SAVANA package has a good potential for future use in scientific research, however the applicability in the clinical setting is likely going to be limited. The availability of a sequenced paired-normal sample in the clinics is usually very limited, albeit needed for more accurate analysis. On top, the regulatory requirements may prevent certification and accreditation of non-deterministic model, based on machine learning. Nevertheless, SAVANA is an advance in the effort to characterize the structural variation in cancer genomes, and as such deserves attention.

Methodology and statistics:

The construction of somatic reference based on the use of both short and long reads in 99 T/N pairs is an asset not only for the paper, but for the cancer field. I am hopeful that the access to such data can be granted as described in the Data Availability section of the Methods.

The use of established somatic reference of COLO829/COLO829-BL as a test set is well accepted, but it is far from real-world tumor data. That is because the sequencing data are coming from two relatively pure cell lines and do not represent the tumor heterogeneity in solid tumors. Using tumor samples (from the TCGA or similar) would have been more convincing for the real use case.

The choice of benchmarking software is adequate, although some packages are intended as use for germline SV/SCNA callers. There is no established ground-truth, which is an illustration of the level of maturity of the field. The visual concordance between Illumina's GRIDSS2 and SAVANA is evident. Most statements of enhanced sensitivity and specificity of SAVANA are well grounded in the analysed datasets.

Data are treated with statistical tests, although I wonder what the interpretation of the statistical significance between two event count bins is (SAVANA vs. another software). Counts differ significantly, but that does not mean that any software count is more correct/wrong. Therefore, the statistical tests are mostly a measure of consistent performance.

Additional questions:

1. Since SAVANA traverses the whole genome, is there a metrics for the global tumor purity? I see that between lines 139 and 144 there is model for purity, which will report local purity estimates. Any output for the global value, would benefit the research and the clinics. Do you plan to implement that?
2. The rationale around the MSI influence (lines 271 to 275) is sound and I generally agree that probably it is not the main factor for the discrepancies between the benchmarked software. However, what evidence authors provide for the absence of inactivation of MMR genes? There are potentially inactivating point-mutations (small SNVs), which may cause gene disruptions, not to mention the epigenetic methylation events. Did you try to assess the mutation status or the epigenetic state of MMR genes? The wording in "we did not detect inactivation of MMR genes or MSI using matched Illumina WGS data for the tumours analysed here" sounds to me as very strong statement, without evidence in the paper.
3. I understand that SAVANA is dependent of machine learning and hence of a large training set. However, for any clinical use, there is requirement for reproducibility of the analysis. If the model is updated (more trained) it may not show the same results with a simple sample re-analysis. How does the authors address the absence of determinism in the further model training?

Minor issues:

1. Line 92 and 93: "as well as infer tumour purity and ploidy, using long read sequencing data, including the estimation of tumour purity and ploidy." This sounds as tautology, please decide on use of "infer" or "estimate".
2. There is slight discrepancy in the nomenclature of true positive, false positive, Null, Both as described in lines 193 to 202 and the corresponding Suppl. Figure 4. Please, stick to the MCP model categories or the commonly established definitions, but not both. E.g. "Noise" or "false positive"?
3. In my version of the PDF (on two operational systems) it was not possible to see the equations on line 556 and 572, making the assessment of the final tumour purity or ploidy model very hard. Please, correct the font or the symbols.
4. It was not clear to me how Supplementary Figure 19 was produced and in what computing environment/configuration. No explanation of the hardware or methodology is given at the present. Is there accountability for single/multi core steps in the various packages? Is it in real or system time? Perhaps, a small section in the Methods and a longer figure legend can address this issue.

Reviewer #1 (Remarks on code availability):

I can verify that the code is easily accessible and available with few mouse clicks. The software installation is possible, and the tool can be set and run without any advanced knowledge through the Conda environment (or can be built from the source with a relatively small effort). The project seems very well managed, and the work looks solid. Although, I have the technical expertise to review all source lines and the logic, going through everything will consume too much of my time, so I haven't performed an extensive review.

Reviewer #2 (Remarks to the Author):

The author presented a machine learning based tool to detected somatic SVs and SCNA from long-read sequencing data. Long-reads have been widely used in analysis healthy genomes, such as the human pagenome reference project, but limited to cancer studies. From the methodology aspect, this study tried to solve two major issues in detecting somatic SVs: 1) distinguish real somatic breakpoint from mapping artifact; and 2) difficulties in assessing SV detection specificity. Besides, the 99 tumor-normal matched long-read sequencing data will be a valuable resource to the community for future methods development. Overall, the benchmark and analysis are comprehensive, but the advantages of using long-reads are not well described and should be further improved.

Questions and comments:

1. As for the machine learning model, both the SV features and random forest model have been used for SV classification with short-reads, such as the tool forestSV published in 2012. Though the model is not novel, the author provides the option to train personalized models, which is a contribution to the community. I have the following questions regarding to this model:
 - a) The machine learning model used to distinguish real breakpoint from artifact was trained by SVs outside of repetitive regions, blacklist regions and amplified regions (>200 reads). Since repeat regions are know hotspot for SVs, I am wondering how SAVANA achieves somatic breakpoint detection at repeat regions (such as centromere) with a model that trained on selected regions.
 - b) The model parameters, performance and training details should be provided, such as the creation of validation and test set. The top important features related to tumor support (Supplementary Figure 3) are interpretable, but not for features like "CLUSTERED_READS_NORMAL", "SVLEN", "TUMOR_ALT_HP_NA_COUNT" and "CIGAR".
 - c) The training set was created by overlapping short-read calls and SVs detected by SAVANA with long-reads data from the 99 samples, which is the same as the samples used for prediction (detection SVs). How dose the model perform if other long-read tools are used to create the true SVs? Or train the model on well-established benchmark, such as HG002, because some of the important features also applied to germline SVs.

2. The author evaluates the breakpoint concordance between each caller and the true SVs from COLO829, here are my questions:

- a) The author used SVIM and cutesv (they are not designed for somatic SV detection) to call tumor and normal separately and subtract the coordinates. While tools like SAVANA, nanomosv use matched normal-tumor samples, it is a biased comparison for SVIM and cutesv.
- b) Is this evaluation distinguish different types of SVs or just compare every detected breakpoints. For example, a somatic deletion contains two breakpoints, how does the author calculate the precision and recall for this deletion? It is also critical to evaluate the whole event, especially for a complex event that containing more than two breakpoints. The breakpoint shift evaluation is not enough to show the performance.
- c) Looking at the absolute value shown in the barplot of Supplementary Figure 7c, the recall of SAVANA, Nanomosv and Severus are very close, why SAVANA is significantly higher than those callers.
- d) SVision-pro also uses the matched tumor-normal samples as input, the author should also assess this deep learning based tool.
- e) Looking at the boxplot in Figure 2b, SAVANA shows a big variation in terms of the number of SVs detected in all replicates. Would this be a problem?

3. The usage of replicates to benchmark could reflect the performance to some extent. While the statement "true somatic SVs should be detected in all replicates ..." is suspicious because a method could have systematic errors for detecting SVs at certain regions due to alignment bias. For example, ONT reads usually aligned poorly at regions of high VNTR copies, where you could find several continuous variant signatures (such as 'D' in CIGAR string) from the alignments. Here are my questions and suggestions regarding this replicates based evaluation:

- a) The false discovery rate of SAVANA should be assessed on normal tissue (i.e., germline SVs) to understand its potential bias or at least show how many SVs detected in all replicates might be false, which could better support the results shown in Figure 2.
- b) All results in Figure 2 are based on identical SVs detected in replicates. How SVs are considered as the same event among all replicates? Does it only consider breakpoint?
- c) Why Figure 2d does not have SVIM and cuteSV? What is the coverage of each replicate?
- d) The observation of deletion enrichment at microsatellite is strange. Microsatellite sequence is essentially tandem repeats, I suggest to add background distribution of SVs on microsatellite.

4. It is great to see the comparison of short-read and long-read data on clinical samples. Phasing tumor genome and detecting complex SVs are the advantage of using long-read. Here are my questions:

- a) SAVANA shows the highest concordance with short-reads (Supplementary Figure 20-21), then why we should sequence tumor with long-reads.
- b) How SVs are determined to be detected by both short-read and long-read data. 85% of SVs are detected by both technologies, what are those only detected by long-read data?
- c) In Figure 5d-f, copy number variation can be direct cause of suppressor gene loss, how to decide if this complex rearrangements are the causes of suppressor gene copy number loss? In other words, with long-read data and phased information, is it possible to show the copy number loss is associated with rearrangement and they are not independent events?
- d) Similar figures like Figure 5d-f are too complex and hard to capture the main point that they author want to show, is there a way to simplify this? And what is the blue line at the bottom, is that copy number loss?
- e) What is the total percentage of somatic SVs could be phased and phased blocks N50 ?
- f) The assembly used all SV supporting reads as described in the methods. Why the assembled contig does not span some of the breakpoint shown by the vertical lines. Moreover, are you able to identify the same breakpoint when you align it back to the reference?

5. I value the efforts have been made to compare every events and provide them as examples in the supplementary.

- a) Figure 4 is strange to me. It looks like that SAVANA only detected breakpoints roughly in range 75Mb-90Mb, and only a duplication at 150Mb that is also detected by GRIDSS, Severus and NanomonSV. What are those deletions detected by callers like Severus and NanomonSVs, since most of them looks concordant between these two callers.
- b) In supplementary Figure 15b, SAVANA, GRIDSS and NanomonSV detected lots of interspersed duplication (blue lines), but these duplication breakpoints on the left side seems to be highly concordant with deletions (orange lines). If the deletion are flanked by large identical repeats, it is possible to detect both deletion and duplication? I suggest to show the raw alignments of ONT reads around some breakpoints.

Reviewer #2 (Remarks on code availability):

The software is easy to install and I could run without error on some test data. Here are my comments and questions for running the SAVANA:

- a) There are several different evaluation in this study, but it is unclear how SAVANA produces those results. Please provide every steps in different evaluation either on Github or in the paper.
- b) Following the experimental setting shown in Figure 3a, I tested on one of the HGVC sample HG00733 (HiFi reads from https://ftp.1000genomes.ebi.ac.uk/vol1/ftp/data_collections/HGVC3/working/20231126_UW_HiFi/HG00733/). The tumor and normal input BAM files for SAVANA are exactly the same and aligned with minimap2. I only used the chr1 alignment as a test,

while it gives me around ~10,000 SVs (records in the *.bedpe file). I also tried to classify with 'savana classify --vcf HG00733.sv_breakpoints.vcf --pb --output HG00733.sv_breakpoints.cly.vcf --somatic_output', but the results are the same. I might miss some steps, could the author explain how they did for the results in Figure 3a.

Reviewer #3 (Remarks to the Author):

In this study, the authors described a novel algorithm called SAVANA for the detection of somatic SVs and SCNAs and the estimation of tumour ploidy and purity in long-read sequencing data. The main focus of this study was to compare the performance of SAVANA against existing SV callers developed for long-read sequencing data. They generated matched high-depth nanopore (51X) and Illumina WGS (118X) data from different tumour types and corresponding germline samples. They trained a random forest classifier with a set of SVs detected by SAVANA in long-read data that were labelled as true somatic SVs if they were also detected in short-read data (by a different pipeline). In addition SAVANA uses Mondrian Conformal Prediction to assess the reliability of individual SV calls.

When comparing SAVANA against existing SV detection algorithms (Sniffles2, CuteSV, SVIM, NanomonSV and Severus), the number of SVs detected was highly variable. In particular, when they used a truth set of 68 somatic SVs detected in a melanoma cell line (also experimentally validated) SAVANA showed higher recall and specificity compared to the other algorithms. In order to assess the performance of these SV callers in an unbiased manner and across the entire genome, they simulated technical sequencing replicates by randomly splitting the reads in two BAM files and called SVs on each replicate. SAVANA not only showed higher concordance across replicates but also across different clonality levels, SV sizes and SV types.

Compared to SAVANA, the other algorithms detected much more insertions and deletions. The authors found that most of these insertions and deletions detected by those algorithms mapped to microsatellites. Considering that the samples used in this study are not MSI, the authors concluded that those insertions and deletions were most likely false positive calls derived from sequencing or mapping errors.

In order to assess the specificity of these algorithms, the authors generated simulated technical sequencing replicates also for the matched germline controls and used one replicate as tumour and the other as matched germline control. Notably, SAVANA showed a significant lower false positive rate compared to the others (which showed much higher detection of false positive SVs (9X to 391X higher). The authors concluded that the higher SV detection rate of the other algorithms was most likely due to lower specificity compared to SAVANA.

In addition 85% of SVs detected in the illumina data by SAVANA were also present in the long read data while the recall rate of the other algorithms was much lower. At the end, the authors showed very briefly that SAVANA can also reliably call SCNA, estimate tumour purity and ploidy and identify SVs involved in the inactivation of tumour suppressor genes or activation of oncogenes.

Comments:

The method is novel and timely. Overall, the benchmarking of the method is sound, with evidence of SAVANA outperforming existing methods. However additional evidence would further support the robustness of the method:

1. In order to train their model, the authors used a set of true SVs which was derived from SVs detected by a short-read WGS data pipeline in the illumina data that were also detected by SAVANA in the long-read sequencing data. Then they classified the SVs detected in each tumour using a random forest model trained on the SVs detected in all other tumours in the cohort (leave-one-tumour-out models). This approach ensures that no data from the tumour investigated was used to train the model. Although this approach is valid as an "internal validation" of the model, a completely independent long-read sequencing dataset, exploring also other cancer types, would further strengthen the validity of their approach.

2. The authors showed that existing algorithms have lower specificity and higher detection rate of false positive SVs compared to SAVANA (measured in simulated sequencing replicates of blood WGS data). Could the authors show the fraction of insertion and deletions among those false positive SVs? And if there are any recurrent ones or overlapping with microsatellites? This could be a further evidence that the higher number of insertions and deletions detected by the existing algorithms are most likely false positive as stated in lines 275-277.

3. Figure 2d shows that SAVANA has a consistent detection rate of somatic SVs in both simulated sequencing replicates when evaluated in different allele fractions compared to existing algorithms, thus allowing this type of analysis in low tumour purity samples. Have the authors tested SAVANA in such samples, i.e. FFPE or ctDNA samples?

4. Overall this study shows that SVs detected by SAVANA are highly consistent to those detected by Illumina short read sequencing. It would be nice if authors could show if any new SVs, potentially biologically relevant, were detected by SAVANA in long read sequencing data that were missed by short read sequencing data.

Minor comments:

1. In line 91 "To address these needs, we present SAVANA, a computationally efficient algorithm specifically designed to

detect both somatic SVs and SCNAs, as well as infer tumour purity and ploidy, using long-read sequencing data, including the estimation of tumour purity and ploidy". Please revise this sentence.

2. In line 397, authors might wrongly refer to figure 3b instead of 3c.

Version 1:

Decision Letter:

Our ref: NMETH-A57445A

5th Feb 2025

Dear Dr. Cortés-Ciriano,

Thank you for submitting your revised manuscript "SAVANA: reliable analysis of somatic structural variants and copy number aberrations using long-read sequencing" (NMETH-A57445A). It has now been seen by the original referees and their comments are below. The reviewers find that the paper has improved in revision, and therefore we'll be happy in principle to publish it in Nature Methods, pending minor revisions to satisfy the referees' final requests and to comply with our editorial and formatting guidelines.

TRANSPARENT PEER REVIEW

ORCID

Sincerely,
Lei

Lei Tang, Ph.D.
Senior Editor
Nature Methods

Reviewer #1 (Remarks to the Author):

I appreciate the authors' dedication in enhancing the quality of the manuscript and reworking the code to incorporate new functionalities. This work represents a valuable contribution to the community, and I would be delighted to see it published in Nature Methods.

One of the most notable improvements is the inclusion of the Tumor-only mode, which is thoroughly discussed on pages 9 to 12 of the rebuttal letter. I recognize the complexity of this undertaking and was pleasantly surprised by the authors' decision to pursue this approach. While it is expected that the Tumor-only mode introduces more noise, as demonstrated in Figures R10 to R13, the reporting of real event remains valuable. Certain SVs and SCNAs may be

anticipated and actionable in clinical contexts. However, I do not advocate for clinical diagnostics based solely on the Tumor-only mode if a normal sample is obtainable. I align with the authors' conclusion:

"...given all the reasons discussed above about the potentially low specificity of tumour only analysis, we still recommend the use of a matched normal sample as germline control whenever possible".

I certainly noted the efforts to improve speed and accuracy, as highlighted on page 13, and I particularly acknowledge the enhanced tumor purity estimates derived from the matched normal, rather than relying solely on the aligner (phased BAM). Additionally, employing a panel of SNPs as a fallback strategy is an excellent approach.

Regarding the point-by-point discussion, I will address it page by page without directly copying the relevant paragraphs.

p.15:

Reliance on third-party aligners: I agree that incorporating and using SAVANA with commonly used aligners is straightforward. My only concern is that errors from the aligner could potentially limit the performance of downstream software, possibly leading to missed or over-reported events. However, I believe this issue will largely be mitigated with the future use of long reads, which should provide more accurate genomic locus mapping (while alignment may be less critical in the context of SV detection). There is little that can be done to address this issue at present, and it should not impede the publication of the rest of the work.

I appreciate the deposition of the 99 T/N reference data at the EGA.

COLO829 reference has its limitations, and I am grateful to see that the authors also acknowledge that.

p.16:

Germline callers: I agree with the authors that this is a common practice and a 'necessary evil' in benchmarking studies. It is far better to include comparisons to this 'ground truth' than to be left wondering how the somatic SV caller relate to the germline counterparts.

p.17:

Thank you for providing additional clarification on the statistical assessment. I am pleased to see such a substantial number of replicates!

p.18:

I welcome the tumor purity improvements and the authors' clarification regarding their estimation method.

Thank you for clarifying the MSI status protocol. While I was not expecting many SNVs in the MMR genes, it is reassuring that it is now clearly stated that SNV/InDel calling was performed.

p.19:

The application of machine learning in clinical practice is a complex topic that should not hinder the publication of the current work. I recognize the value of SAVANA in scientific research and investigative clinical practice. While the argument that model-state equates to versionmodel is theoretically sound, its practical implementation across multiple clinical sites—especially without data sharing—could pose significant challenges. Ultimately, this is a matter for regulatory authorities to address. Thank you for the clarification.

Fig. 19 and related: Thank you for the additional comments and the revised figure caption.

Personally, I am not concerned about the RAM consumption, as long as the software is able to fit in the most computer configurations. Speed is always a positive feature—often the first question raised, but typically one of the last priorities in clinical settings. That said, great job!

In conclusion, I commend the authors for their additional efforts, their dedication to improving both the manuscript and the software, and their rational in addressing my concerns. Therefore, I strongly recommend this work for publication.

Reviewer #1 (Remarks on code availability):

I can verify that the code is easily accessible and available with few mouse clicks. The software installation is possible, and the tool can be set and run without any advanced knowledge through the Conda environment (or can be built from the source with a relatively small effort). The project seems very well managed, and the work looks solid. Although, I have the technical expertise to review all source lines and the logic, going through everything will consume too much of my time, so I haven't performed an extensive review.

Reviewer #2 (Remarks to the Author):

I appreciate the extra analysis and method improvement that had been done by the author. I think most of my previous questions are addressed. The revision is well organized but the manuscript lacks description for some important results,

especially in Figure 2.

Questions and comments.

1. The green box in Figure R2 and others lack proper legend. It is hard to match the IGV view with its top panel. For example, in Figure R2 IGV view, the DEL's left breakpoint locates inside COL2A1, which seems discordant with the top panel.
2. "Tumor-only" mode is definitely an important function for clinical applications. The author says "SAVANA can now detect somatic SVs by using a machine learning model trained for this purpose and using a large panel of normals to filter out likely germline SVs". While I am not able to find other details regarding to this part, what is the difference of this "tumor-only" model. Looking at the examples called from "tumor-only" mode, it seems that this function actually depends on filtering likely germline SVs in large normal panel.
3. How is the copy number estimated for centromere? For example, in Figure1, the total and minor allele copy number is 1 and 0. The example in Figure 4 shows total copy number 3 for the middle part of the centromere. Since centromere is one of the most divergent regions among humans, could those just be mapping artifact? I am aware this might out of the scope while it would be great to clarify this.
4. The authors showed important evaluation for somatic SV detection in Figure 2b-d but need further details in the method section:
 - a. It seems that the author created two replicates. Dose "all replicates" in Figure 2b indicate both replicate?
 - b. For the y-axis labels in Figure 2d "Fraction of validated SVs using replicates", dose it indicate SVs that are detected in both replicates?
 - c. Figure 2d decompose the result in 2c by allele frequency (x-axis in 2d) estimated by supporting reads. Looking at the circle size in 2d, it seems that savana detects similar number of SVs at different AF while Severus and NanomonSV decrease. It is also odd that Sniffles2 have fewer SVs of AF smaller than 0.2 and there is significant increase at (0.2,0.25].
 - d. In 2d, the number of validated SVs using replicates can be calculated by multiply the value on y-axis and the circle size (total number of SVs). For example, the savana y-axis value ~0.75 at (0.15,0.2] with a circle size of 10000, so the number of validated SV is ~7500. Is this number should be concordant with the number shown in Figure 2b?
5. I am confused about the phasing based evaluation shown in Figure 2g and 2h.
 - a. How those SVs shown in 2h are selected for this evaluation.
 - b. It says in line 288-289 "The key idea is that ...". It is true for a heterozygous SV but not for homozygous events.
 - c. According to the method section, the author tested if some reads are incorrect assigned to an allele with the null hypothesis, a give SVs is supported by reads from one allele only. In 2g, inconclusive are SVs rejected the null hypothesis but the definition of inconsistent phasing is unclear. d. It seems that the test is only applied to SVs supported by reads from single allele, but why the inconclusive legend is used for "SVs with support from both alleles" in 2g.
6. In Figure 1, the step "SVs refine segmentation and copy-number change points rescue ..." is not described in the method section.
7. The method section "Detection of SVs using existing algorithms" miss the description for SVision-pro.

Reviewer #2 (Remarks on code availability):

The code is well organized with clear instructions.

Reviewer #3 (Remarks to the Author):

The authors answered all my comments.

Version 2:

Decision Letter:

18th Apr 2025

Dear Dr Cortés-Ciriano,

I am pleased to inform you that your Article, "SAVANA: reliable analysis of somatic structural variants and copy number aberrations using long-read sequencing", has now been accepted for publication in Nature Methods. The received and accepted dates will be 6th Aug 2024 and 18th Apr 2025. This note is intended to let you know what to expect from us over the

next month or so, and to let you know where to address any further questions.

Over the next few weeks, your paper will be copyedited to ensure that it conforms to Nature Methods style. Once your paper is typeset, you will receive an email with a link to choose the appropriate publishing options for your paper and our Author Services team will be in touch regarding any additional information that may be required. It is extremely important that you let us know now whether you will be difficult to contact over the next month. If this is the case, we ask that you send us the contact information (email, phone and fax) of someone who will be able to check the proofs and deal with any last-minute problems.

If you are active on Twitter/X or Bluesky, please e-mail me your and your coauthors' handles so that we may tag you when the paper is published.

Best regards,
Lei

Lei Tang, Ph.D.
Senior Editor
Nature Methods

** Visit the Springer Nature Editorial and Publishing website at http://editorial-jobs.springernature.com?utm_source=ejP_NMeth_email&utm_medium=ejP_NMeth_email&utm_campaign=ejp_Nmeth www.springernature.com/editorial-and-publishing-jobs for more information about our career opportunities. If you have any questions please click [here](mailto:editorial.publishing.jobs@springernature.com).

Response to Reviewers

We thank the Reviewers for their careful reading of our manuscript and their constructive comments. We think that the quality of the manuscript has been improved significantly as we modified the paper based on their comments. Importantly, we have made substantial changes to SAVANA, including the implementation of novel functionalities to enable tumour-only detection of somatic structural variants (SVs) and copy number aberrations (SCNAs). We summarise below the key changes and analyses that we have incorporated in response to the Reviewer's comments, followed by more detailed point-by-point responses.

General comments

Reviewer 1 highlighted the challenges associated with requiring a matched germline control for SV discovery given that the availability of matched normal samples (e.g. blood) in clinical settings is often limited. Therefore, the Reviewer highlighted that the broader applicability of SAVANA in clinical settings would be contingent on its ability to detect SVs and SCNAs without requiring a paired normal sample. In addition, Reviewer 1 asked for more explanation on the technicalities of the benchmarking analyses and data availability. **Reviewer 2** asked for more discussion on the advantages of long-read sequencing for SV and SCNA analysis, and more clarification on technical aspects related to model training. **Reviewer 3** asked primarily for additional analyses on the enrichment of false positives at microsatellite regions, and for examples showcasing the value of long-read sequencing to uncover novel cancer biology aspects through the detection of SVs and SCNAs missed by short-read sequencing.

We agree that these are important issues. Indeed, we have invested a great deal of effort in performing additional analyses to address the main concerns raised by the Reviewers (for example, we have now included SVision-pro in all the benchmarking analyses). In addition, we have further extended the functionality and performance of SAVANA as explained below.

1. Advantages of long-read sequencing for the analysis of somatic SVs and SCNAs using SAVANA.

Given that short and long reads (when analysed using SAVANA) perform on par for the detection of SVs and SCNAs, Reviewers 2 and 3 questioned why it would be beneficial to analyse cancer genomes using long-read sequencing. Overall, the analysis of SVs and SCNAs using long-read sequencing and SAVANA has several advantages over short-read sequencing that can be transformative to harness the unique features of long-read sequencing in both research and clinical settings. We describe below what we consider are the key advantages of long-read sequencing for cancer genome analysis. For a broader discussion on the advantages of long reads, we refer the Reviewers to the following perspective published in *Nature Methods* in 2022 to celebrate long reads as the method of the year, in which our work on cancer genomics was already highlighted(1).

- Improved detection of driver SVs and SCNAs using SAVANA

Overall, most SVs mapping to driver genes in the cohort detected using short-read sequencing can also be detected using long reads (**Figure R1**).

Figure R1. Comparison of short-read WGS (srWGS; blue) and long-read WGS (lrWGS; green) for the detection of SVs in cancer driver genes. Cancer driver genes in each tumour were identified through the analysis of the Illumina data using GRIDSS and PURPLE. SVs in long-read data were detected using SAVANA. Overall, srWGS and lrWGS perform on par for the detection of SVs in cancer driver genes.

However, to illustrate the added value of long reads, we have now included additional analyses and specific examples of cases where only long reads identified cancer driver mutations. For example, **Figure R2** below shows the example of a structural variant disrupting a driver gene, namely *COL2A1*, in a chondrosarcoma. The breakpoints cannot be reliably reconstructed using short reads. However, as can be seen in the IGV screenshots below, the alignment quality of the long reads is significantly higher as compared to the Illumina data, which permits improved reconstruction of breakpoints.

Figure R2. Example of an SV disrupting *COL2A1* in a central conventional chondrosarcoma. The top and bottom panels show the sequencing reads from nanopore and Illumina sequencing of the same tumour, respectively. The quality of the alignment of the Illumina reads is poor, which results in mapping qualities of 0 (light grey sequencing reads in the top panel). As a result, the breakpoints cannot be reliably detected.

Figure R3. Somatic SVs and SCNAs mapping to the *NF1* locus in malignant peripheral nerve sheath tumours (MPNSTs) SARC-054 and SARC-057. The SVs and SCNAs detected using SAVANA and long reads are shown on the left, and the SVs and SCNAs detected using Illumina WGS and GRIDSS/PURPLE are shown on the right. These two examples illustrate two cases in which the rearrangements that lead to the loss of *NF1* (the main driver of MPNST development) are only detected by SAVANA using the long-read data.

Investigation of the raw sequencing data clearly shows that short reads struggle to reliably map SVs in low-complexity regions. The clinical relevance and functional consequences of SVs in low-complexity regions, such as centromeres, are still poorly understood. However, SVs in low-complexity regions can also be directly involved in the inactivation of driver genes.

For example, in the tumour below (**Figure R4**) the SV leading to the loss of *NF1* in an MPNST can only be fully reconstructed using long-read sequencing. *NF1* is the pathognomonic driver gene in this tumour type(2), making its detection essential for correct diagnosis and patient management.

Figure R4. Top panel: Somatic SVs and SCNAs mapping to the *NF1* locus in malignant peripheral nerve sheath tumour (MPNST) SARC-058. The SVs and SCNAs detected using SAVANA and long reads are shown on the left, and the SVs and SCNAs detected using Illumina WGS and GRIDSS/PURPLE are shown on the right.

Bottom panel: IGV screenshots showing the Illumina and nanopore read alignments for the SV shown on the top panel. The rightmost breakpoint maps to *TBC1D3G*, one of the 8 paralogues that encode for oncoprotein *TBC1D3*(3). The breakpoint and change in copy number at both breakpoints can be reliably reconstructed using the long reads, whereas the short reads show low mapping quality, as indicated by the light colour of sequencing reads.

In another example (**Figure R5**), a deletion mapping to an Alu element in *ATRX* could only be reliably reconstructed using long reads. Specifically, the mapping quality of the short reads at that regions is very low due to the repetitive nature of the Alu element, whereas the long reads can span the region to achieve high mapping quality and accurate breakpoint reconstruction.

Figure R5. Example of a breakpoint mapping to a repetitive region in an intronic region of ATRX. Long reads can be reliably mapped to the locus permitting accurate breakpoint detection, whereas short reads cannot be unambiguously mapped and the breakpoint is not detected.

In addition to SVs, we note that SAVANA is the only somatic copy number caller for long-read sequencing data. This functionality is critical to identify e.g. the inactivation of tumour suppressor genes through copy neutral loss of heterozygosity (LOH; see **Figure R6** for an example). Overall, these examples show the value of long-read sequencing to improve the detection of breakpoints in cancer driver genes mapping to genomic regions to which short reads cannot be reliably aligned, thereby highlighting the advantages of long-read sequencing for SV detection in cancer genomes. We have now included these results in the manuscript.

Figure R6. Example of a complex genomic rearrangement profile for osteosarcoma SARC-023 showing the detection of copy-neutral LOH at the *TP53* locus and the amplification of *ERBB2*.

- **Phasing of SVs and SCNAs**

As Reviewer 2 indicates, the potential for phasing of SVs is a major advantage of long-read sequencing. However, existing methods do not harness or report phasing information for SV or SCNA analysis. To address this need, SAVANA provides functionalities for the detection of SCNAs and SVs at single haplotype resolution. In practice, this is a critical innovation as haplotype-resolved SV and SCNA calls are essential to determine whether a given genomic rearrangement (simple or complex) affects one or two alleles. For example, this is important to determine whether in a given tumour both copies of a tumour suppressor gene are inactivated, indicating biallelic inactivation, or only one. This can be particularly important to assess whether a somatic aberration in a patient with a cancer predisposition syndrome affects the wild type allele, thus causing biallelic inactivation, or the same parental copy harbouring a pathogenic germline variant.

In addition, phased SVs enable cancer genome assembly at single-haplotype resolution, thereby helping determine the rearrangement mechanisms and evolutionary trajectories of simple and complex SVs. For example, we recently used SAVANA for the analysis of nanopore WGS data from human osteosarcomas, which allowed us to discover a novel rearrangement mechanism, termed Loss-Translocation-Amplification chromothripsis (*Cell*, In press; preprint available at: <https://www.biorxiv.org/content/10.1101/2023.12.29.573403v1>). Notably, using the phased SV calls generated by SAVANA we could perform genome assembly at single-haplotype resolution, which allowed us to determine that the rearrangement process only affects one allele of *TP53* (see **Figure R7**).

Figure R7. Haplotype-resolved SV and SCNA calls generated using SAVANA permit the reconstruction and assembly of multi-chromosomal LTA chromothripsis events in high-grade osteosarcomas. Somatic SV and somatic copy number profiles for chromosomes 6 and 17 involved in LTA chromothripsis events reconstructed using Illumina WGS data from high-grade osteosarcomas G100-31 (**A**) and G100-66 (**B**). Haplotype-resolved assembly of the LTA chromothripsis events detected in G100-31 (**C**) and G100-66 (**D**) using nanopore whole-genome sequencing data. The plots show the alignment of the assembled contig for each parental allele (arbitrarily labelled as “allele 1” and “allele 2”) to chromosomes 6 and 17 of the human reference genome. Allele-specific genome assembly was performed using sequencing reads from a single allele of a haplotype block identified using WhatsHap. Overall, the assemblies revealed that LTA chromothripsis affects one parental allele only, and that the derivative chromosomes resulting from LTA chromothripsis involve genomic regions from chromosomes 6 and 17. Adapted from Espejo Valle-Inclán, De Noon, et al. *Cell*. *In press*.

Beyond helping us decode the mechanisms underpinning rearrangement processes, phasing SVs can be harnessed to improve algorithmic performance and benchmark SV detection

algorithms. In the new version of our manuscript, we have performed read-backed phasing of SVs to assess whether the reads supporting SVs detected by the algorithms we have benchmarked are supported by reads assigned to a single haplotype (as it would be expected for true positive SVs) or by reads from both alleles (which would be consistent with false positive SVs). The strategy of read-backed phasing has been extensively used in other applications, such as the detection of somatic mutations in single-cell WGS data affected by high rates of allelic imbalances and dropouts caused by imperfect DNA amplification(4). However, SAVANA is the first method that harnesses haplotype information for SV detection and applies the concept of read-backed phasing for benchmarking somatic SV callers. Specifically, we have computed the fraction of SV calls generated by each algorithm that are supported by sequencing reads assigned to only a single parental allele. For this analysis, we have excluded genomic regions with LOH, as even false positive SV calls mapping to genomic regions with LOH would be supported by reads from a single parental allele.

Importantly, this analysis (**Figure R8**) has revealed that, as opposed to other methods, the reads supporting SVs detected by SAVANA are assigned to a single haplotype. Overall, these results, which agree with the other analyses we present in our manuscript, further show the much lower false positive rate of SAVANA as compared to existing SV detection methods. Moreover, the total number of SVs in green (i.e. with consistent read-backed phasing) is highest for SAVANA. Importantly, this indicates that SAVANA has both higher specificity and higher sensitivity to detect true somatic SVs.

Figure R8. Somatic SVs called by SAVANA are supported by sequencing reads from a single parental allele, whereas SV detected by other methods are supported by sequencing reads from both parental alleles, consistent with them being false positive SV calls.

Left. Each dot represents an SV. The x and y axes report the number of sequencing reads assigned to either parental allele (arbitrarily labelled as allele 1 and allele 2, respectively) supporting each SV.

Right. Same data shown in the left panel depicted in stacked Barplot format.

In both panels, SVs supported by sequencing reads assigned to the same parental allele are coloured in green. SVs supported by sequencing reads assigned to allele 1 AND also by sequencing reads assigned to allele 2 are shown in blue and red. Given that phasing is not always accurate (due to e.g. sequencing errors at SNP loci or limited read length, among other reasons) it is possible that a small subset of reads might be assigned to the incorrect parental allele. As a result, some of the reads supporting a true SV might show discordant phasing. To account for this, we performed a Binomial test (using a probability of success of 0.95 and with a one-tailed alternative hypothesis) on each SV with supporting reads from both alleles and applied an FDR-corrected *P* value of 0.05 as the threshold for statistical significance. SVs with a significant number of discordant reads are shown in red. Those SVs for which we do not have enough statistical power to reject the null hypothesis (i.e.. that the SV is supported by reads from one allele only) are shown in blue. Thus, our analysis of read-backed phasing is inconclusive for the SVs labelled in blue. Overall, these results are consistent with the notion that SAVANA

distinguishes true somatic SVs from false positive calls, thus showing a significantly lower false positive rate than other SV detection algorithms.

- **Rapid turnaround, scalability and multi-modal readouts**

Finally, we note that long-read sequencing technologies, especially nanopore sequencing, have elicited great interest in the clinical community due to the rapid turnaround of results. For example, Illumina WGS of tumours via the NHS in the UK requires centralized sequencing and analysis, which results in turnaround times in the order of weeks. By contrast, nanopore WGS can be performed in a decentralized manner at local genomic testing laboratories and has the potential to deliver genomic results (even WGS) in a few days. This is critical as for some cancer types, e.g., solid tumours in children, rapid turnaround is critical to enable effective patient management. In addition to rapid turnaround, long-read sequencing methods provide DNA methylation information in the same assay, which has already shown great potential to e.g. assist surgeons even during surgery(5).

However, a major challenge to harness the full potential of long-read sequencing technologies for clinical applications is the availability of robust and comprehensively validated algorithms for data analysis. This is why we are confident that the high performance and reliability afforded by SAVANA can be transformative for harnessing the practical advantages of long-read sequencing in clinical settings, thereby helping bridge the gap between genomic sequencing technology development and the lack of sound algorithms for analysis. In fact, the development of SAVANA is part of a broader effort in the UK focused on testing the potential of long-read sequencing methods to improve the turnaround of genomic testing results. The authors of this article have played a major role in that effort. Through an iterative process of algorithm development and validation throughout >4 years, the team behind SAVANA has managed to develop a tool that is compatible with clinical timelines in that SAVANA runs much faster than existing methods for SV detection while also providing functionalities for SCNA, purity and ploidy analysis. Therefore, we consider that SAVANA is an exemplar of how to integrate clinical and academic research to harness novel genomic technologies to help advance genomic medicine and consider that its functionalities and validation framework will be of great interest to the community.

2. Novel functionalities to expand the potential of SAVANA for the analysis of clinical samples.

As Reviewer 1 indicates, the availability of a matched normal sample is often limited in clinical settings. Motivated by the importance of this point, we have now greatly expanded the functionalities of SAVANA to facilitate the analysis of samples without a matched germline control - what the community refers to as “tumour-only” mode or analysis. Specifically, in tumour-only mode, SAVANA can now detect somatic SVs by using a machine learning model trained for this purpose and using a large panel of normals (available in the GitHub repository of SAVANA) to filter out likely germline SVs.

In addition, the tumour-only mode of SAVANA permits the inference of allele-specific copy number profiles and tumour purity with comparable accuracy to Illumina sequencing and SAVANA when run using a matched germline control (see **Figure R9** below, which we have now included as Figure 6 in our manuscript). SAVANA is the only algorithm with these functionalities, which we are confident will help increase the applicability of SAVANA to clinical samples.

Figure R9. Comparison of the tumour purity and ploidy estimates computed using different SAVANA modes and ONT WGS data against PURPLE and Illumina WGS data. (a) Tumour purity estimates. (b) Tumour ploidy estimates. SAVANA paired germline SNPs shows results for matched tumour-normal pair analysis using matched normal germline SNPs for purity estimation. SAVANA paired 1000g SNPs shows results for matched tumour-normal pair analysis using the 1000 Genome Project population SNPs with AFs > 0.25 and AFs < 0.75. SAVANA tumour only 1000g SNPs shows results for analysis using the 1000 Genome Project population SNPs with AFs > 0.25 and AFs < 0.75. For this analysis, we only considered the 44 tumours with region-matched nanopore and Illumina WGS data.

Finally, we have evaluated the performance of SAVANA to detect SVs in cancer driver genes in tumour-only mode. This analysis has revealed that SAVANA - even in tumour-only mode - reliably detected somatic SVs and SCNAs disrupting tumour suppressor genes or amplifying oncogenes on par with SAVANA run using a matched normal sample. See **Figures R10-13** for examples (which we have now included in the manuscript).

Figure R10. Somatic SVs and SCNAs for chromosome 12 for tumour SARC-028. The SVs and SCNAs detected using SAVANA in tumour-only and paired mode are shown on the left and right, respectively. Overall, it can be seen that the SCNAs and SVs detected in either mode are highly correlated. More importantly, the amplification of *CDK4* and *MDM2* are detected at the SV and SCNA levels in tumour-only mode.

Figure R11. Somatic SVs and SCNAs for chromosomes 13 (top) and 17 (bottom) for tumour SARC-023. The SVs and SCNAs detected using SAVANA in tumour-only and paired mode are shown on the left and right, respectively. Overall, it can be seen that the SCNAs and SVs detected in either mode are highly correlated. More importantly, the SVs affecting *RB1*, *TP53*, *ERBB2* and *NF1* and the associated changes in copy number (including LOH) are detected reliably in tumour-only mode.

Figure R12. Somatic SVs and SCNAs for chromosomes 11 and 13 (top and bottom, respectively) for tumour GBM-001. The SVs and SCNAs detected using SAVANA in tumour-only and paired mode are shown on the left and right, respectively. Overall, it can be seen that the SCNAs and SVs detected in either mode are highly correlated. More importantly, the SVs affecting *RB1* and the associated copy number aberrations are detected reliably in tumour-only mode.

Figure R13. Somatic SVs and SCNAs for chromosome 9 for tumour SARC-001. The SVs and SCNAs detected using SAVANA in tumour-only and paired mode are shown on the left and right, respectively. Overall, it can be seen that the SCNAs and SVs detected in either mode are highly correlated, and that the disruption and LOH at the *CDKN2A* locus is detected in tumour-only mode reliably.

However, we note that SV calling using the tumour-only mode might be inevitably increase the false positive rate in some instances. This is expected given that population databases do not encompass all possible germline SVs. As a result, in tumour-only mode it is not possible to fully distinguish somatic SVs from germline SVs not reported in population databases. We note that the impact of this issue is not even across populations as some genetic ancestries are underrepresented in genomic databases. Therefore, while the recall of somatic SVs detected by SAVANA is comparable when run in either paired or tumour-only mode (**Figure R14**), given all the reasons discussed above about the potentially low specificity of tumour-only analysis, we still recommend the use of a matched normal sample as germline control whenever possible.

Figure R14. Fraction of high-quality SVs detected in Illumina data using GRIDSS (SV quality score ≥ 1000) that are detected in nanopore data using SAVANA in paired (dark green) and tumour-only (light green) mode.

3. Additional improvements on SAVANA

In addition to the implementation of a tumour-only mode, we have greatly improved the performance of SAVANA and added novel functionalities that have increased its performance further. Many of these improvements have been guided by the feedback we have received from worldwide users of SAVANA via GitHub and direct correspondence with the team. Specifically, the major improvements we have implemented are:

- Increased speed and accuracy of the copy number analysis

Based on feedback from the community, we have redesigned the copy number calling component algorithm of SAVANA to improve both speed and accuracy. Specifically, we have reimplemented the parallelisation design so that the computing requirements requested can be different for the copy number and SV calling components. In practice, this improves the use of computational resources and has resulted in a 3-5 fold increase in speed. In addition, we have expanded the output of SAVANA to report information requested by diverse users, such as the mean BAF value and number of heterozygous SNPs used to compute copy number values for each genomic segment. All these changes have resulted in more robust inference of SCNAs, including at challenging regions. For example, the new version of SAVANA delivers improved copy number results for the minor copy number allele in highly amplified regions, allowing improved interpretation of rearrangement processes (**Figure R15**).

Figure R15. Rearrangement profile for chromosome 6 of tumour SARC-006. The top ReConPlot shows the SVs and SCNAs detected SAVANA v1.1.29 (old version; top panel). The minor copy number profile at the highly amplified region around 50Mb shows oscillations that are not consistent with the complex genomic rearrangement affecting a single allele, as indicated by the phasing data of SVs, genome assembly and the Illumina data (see also **Figure R4**). However, with the new version of SAVANA (v1.2.7), the noise in the segmentation of the minor copy number is not present (bottom panel), enabling the correct biological interpretation that only one allele is affected by the complex genomic rearrangement.

Finally, previous releases of SAVANA required a phased BAM file to perform copy number analysis. We have now modified the algorithm to rely on the estimation of tumour purity and allele specific copy number values using either (1) the list of germline heterozygous SNP calls detected in a matched normal sample, or (2) a panel of polymorphic SNPs, as established by initiatives such as the 1000g project. Notably, this novel functionality increases the versatility of the tool and its computational

footprint by making the phasing of germline heterozygous SNPs an optional step without compromising accuracy (**Figure R9**).

- Filter to remove artefactual foldback-like inversions

In our manuscript, we reported the identification of a nanopore-specific artifact that produces sequencing reads with a fold-back-like pattern. In our first submission, we used the number of such artefacts to establish a threshold to remove low-quality samples but did not consider such artefactual fold-back-like inversions during the SV calling process itself. We have now implemented a filter in SAVANA to prevent the use of artefactual reads during SV calling. This has led to improvements in the quality of the SAVANA calls, especially for inversions. We note that SAVANA is the only algorithm that provides this functionality.

Point-by-Point Responses

Reviewer #1

The manuscript "SAVANA: reliable analysis of somatic structural variants and copy number aberrations in clinical samples using long-read sequencing" by Elrick et al. is presenting a novel algorithm for SV and SCNA detection in paired tumor-normal sequencing data. The work is welcomed advancement in the field of structural variation calling for cancer diagnostics. Despite the availability of tools from the major sequencing platforms, there is no clearly established state-of-the-art software. The manuscript provides evidence that SAVANA outperforms a battery of software packages, designed for the same type of analysis, both in terms of sensitivity and specificity. It is a clear benefit that the algorithms of SAVANA are designed with the specifics of tumor biology in mind, for the somatic SV analysis, which is considerably harder problem than germline SV calling. The solution works with aligned sequencing data from all three major sequencing providers (Illumina, ONT and PacBio) and the source code is freely accessible under the Apache 2.0 license.

We thank the Reviewer for these encouraging comments, especially for appreciating the importance of considering tumour biology for accurate somatic SV and SCNA detection and interpretation.

Although SAVANA interprets data from several sequencing platforms, it is reliant on their stock aligners or third-party tools, which is a potential drawback. The SAVANA package has a good potential for future use in scientific research, however the applicability in the clinical setting is likely going to be limited. The availability of a sequenced paired-normal sample in the clinics is usually very limited, albeit needed for more accurate analysis. On top, the regulatory requirements may prevent certification and accreditation of non-deterministic model, based on machine learning. Nevertheless, SAVANA is an advance in the effort to characterize the structural variation in cancer genomes, and as such deserves attention.

We agree with the Reviewer that the availability of matched germline controls might be sometimes limited in certain clinical contexts. Motivated by the importance of this issue, we

have now incorporated functionalities in SAVANA to detect SVs in tumour-only mode. In addition, SAVANA now includes functionalities for the detection of SCNAs and estimate tumour purity (a.k.a. tumour cellularity) using tumour-only WGS data. That is, SAVANA now permits the detection of both somatic SVs and SCNAs even when a matched normal sample is not available. Please see our Response in the “General Comments” section for further details on the tumour-only model of SAVANA. Given these improvements, we are confident that these new functionalities will greatly enhance the applicability of SAVANA to clinical samples.

As for the dependency of SAVANA on third-party resources, we note that SAVANA only depends on a reference genome and an alignment algorithm. These are requirements that are common to the vast majority of mutation detection methods for cancer genome analysis, including those used to deliver genomics analysis results to tumour molecular boards in the EU (e.g. Hartwig Medical Foundation), UK (e.g. the Genomic Medicine Service of the NHS and Genomics England) and US (e.g. providers like Foundation Medicine or Tempus). Importantly, we have shown that the performance of SAVANA is uniformly high across aligners, reference genome versions and long-read sequencing technologies (ONT and PacBio). In practice, this means that SAVANA does not require the use of a specific aligner-reference genome version pair, increasing its versatility. Moreover, we consider that the fact that SAVANA can be used with commonly used aligners and reference genome versions is an advantage in that it will make it easy to integrate SAVANA in cancer genome analysis frameworks that also include algorithms for the detection of other type of mutations (e.g. SNVs) without requiring additional computational overhead for e.g. realignment.

Methodology and statistics:

The construction of somatic reference based on the use of both short and long reads in 99 T/N pairs is an asset not only for the paper, but for the cancer field. I am hopeful that the access to such data can be granted as described in the Data Availability section of the Methods.

We thank the Reviewer for recognizing the value of the long-read sequencing data set we present. Yes, all the data have been deposited at the EGA under accession number: EGA50000000614 and will be released upon publication.

The use of established somatic reference of COLO829/COLO829-BL as a test set is well accepted, but it is far from real-world tumor data. That is because the sequencing data are coming from two relatively pure cell lines and do not represent the tumor heterogeneity in solid tumors. Using tumor samples (from the TCGA or similar) would have been more convincing for the real use case.

We thank the Reviewer for the opportunity to discuss this point further. We very much agree that the COLO829/COLO829-BL cell lines are far from clinical samples. In fact, we consider that a major contribution of our study is the use of a large collection of clinical samples for the benchmarking and development of SV detection algorithms. So far, the articles presenting existing SV detection algorithms (e.g., NanomonSV(6); Severus(7)) have solely relied on cell lines to quantify performance and establish truth sets of somatic SVs, thus making it not possible to quantitatively assess performance in clinical samples. As we

extensively show and discuss in our study, using cell lines for developing and benchmarking SV detection methods is limited in many ways (see the third paragraph of the Introduction section, also copied below):

“To date, assessment of algorithmic performance has primarily relied on SV truth sets encompassing small sets of tens SVs detected in cancer cell lines and validated experimentally(6–13). However, using SV truth sets to benchmark SV detection methods is limited in multiple ways. First, SV truth sets are biased towards genomic regions amenable to experimental validation. This results in a bias towards excluding SVs in low-complexity regions, which are precisely the genomic elements intractable to short-read sequencing that could potentially be characterised more accurately using long reads. Second, the selection of candidate SVs for experimental validation is performed using existing SV detection algorithms, which might have their own biases (e.g., sensitivity for SV detection might be variable across SV classes or genomic regions). Third, SV truth sets are only reliable to evaluate the recall of algorithms, but not their specificity because, without additional experimental validation, it is not possible to determine whether SVs called outside the genomic regions interrogated experimentally are artefacts or true SVs. Finally, experimental validation of SVs is time-consuming and not easily scalable, which limits the size of SV truth sets, and thus, the statistical power for benchmarking algorithms. Therefore, due to the lack of both large-scale long-read data sets for human cancers and best practices for benchmarking somatic SV detection algorithms in an unbiased manner, the relative performance of existing algorithms and the extent to which long-read sequencing approaches can improve the characterization of SVs, and by extension SCNAs, as compared to short-read sequencing remains unclear.”

However, despite these considerations, we considered that keeping the benchmarking results for COLO829 and COLO829-BL in our study is useful to (1) show the much higher performance of SAVANA also on cell lines widely considered in previous studies, and to (2) motivate the discussion on why using cell lines is very limited for assessing algorithmic performance reliably, which is the reason why we established the benchmarking framework based on replicate analysis of clinical samples and sequenced 99 samples.

Therefore, the benchmarking framework we propose in our study will have a major impact in the field as it will enable the data-driven benchmarking of SV detection methods using clinical samples instead of cell lines.

The choice of benchmarking software is adequate, although some packages are intended as use for germline SV/SCNA callers. There is no established ground-truth, which is an illustration of the level of maturity of the field. The visual concordance between Illumina’s GRIDSS2 and SAVANA is evident. Most statements of enhanced sensitivity and specificity of SAVANA are well grounded in the analysed datasets.

We thank the Reviewer for these positive and encouraging comments. Indeed, the lack of ground truth data is a major reason underpinning our efforts to establish a framework for the data-driven benchmarking of SV callers.

Similar to the analysis of data from the COLO829 cell line, we very much agree with the Reviewer that germline callers, such as SVIM, cuteSV and Sniffles, were not designed, and should not be used, for cancer genome analysis. However, the critical point here is that such

algorithms have been used in the publications that reported that long reads detected thousands of SVs that short-read sequencing missed, a claim that has been echoed by reviews and sequencing companies during the last 5+ years(14). As a result of the strong amplification and dissemination of this notion, the research and clinical communities have considered it to be true and driven by true biological signals. In fact, we are frequently contacted by clinicians willing to implement long-read sequencing primarily because the expectation that long reads will uncover thousands of novel SVs that cannot be detected by short-read sequencing has widely percolated the clinical and research communities.

Importantly, by including germline SV callers in our benchmarking analysis and running them as performed in previous studies, we have been able to explain why existing algorithms show very high false positive rates, which in turn, has allowed us to understand and report why the claims on the much higher rates of SVs detected using long reads are the result of poor algorithmic performance rather than true biological signals missed by short reads.

Overall, we consider that the methodological advances and best practices for benchmarking that we present in this manuscript make it particularly suitable for publication in *Nature Methods*, as our study clearly shows the importance of developing sound data analysis strategies and methods to harness novel technologies to derive correct biological conclusions.

Data are treated with statistical tests, although I wonder what the interpretation of the statistical significance between two event count bins is (SVANA vs. another software). Counts differ significantly, but that does not mean that any software count is more correct/wrong. Therefore, the statistical tests are mostly a measure of consistent performance.

We thank the Reviewer for the opportunity to clarify this point.

For each algorithm, we computed the replication rate, which is the fraction of SVs detected in both replicates divided by the total number of SVs detected in one or both replicates. As this is a point estimate the accuracy of which depends on the total number of SVs called by each algorithm, we use bootstrapping, which allows us to compute distributions and confidence intervals suitable for hypothesis testing. Specifically, we sample with replacement from the set of SVs detected by a given algorithm, and then compute the replication rate for each resample. This process is repeated 100 times. In this way, we compute a distribution where each datapoint in the distribution is the replication rate for a resample. This allows us to quantify the confidence interval for the replication rate distribution without assuming any underlying distribution of the data. In addition, we can use statistical tests to assess whether the distributions computed for two callers are significantly different, which would indicate that the performance of one algorithm is higher than the other. This strategy to compare the performance of algorithms based on bootstrapping has been widely used in the past for the comparison of mutation detection methods, see e.g. (15). We have now clarified these points in the manuscript.

Additional questions:

1. Since SAVANA traverses the whole genome, is there a metric for the global tumor purity? I see that between lines 139 and 144 there is model for purity, which will report local purity

estimates. Any output for the global value, would benefit the research and the clinics. Do you plan to implement that?

We thank the Reviewer for indicating that this point was not sufficiently clear. SAVANA computes the global (not local) tumour purity. That is, it estimates the tumour cellularity for each tumour, in a similar way as previous cancer genome studies(16). We have now revised the main text and Methods section in the manuscript to clarify this point. The comparison between global tumour purity estimates computed using SAVANA on long read WGS data and PURPLE applied on matched Illumina WGS data is presented in Figure 6 (**Figure R8** in this document). This analysis shows that the purity estimates computed by SAVANA on long read data are highly correlated with PURPLE, which is a state-of-the-art Illumina WGS data analysis tool used for the analysis of clinical tumour samples at the Hartwig Medical Foundation in the Netherlands(17, 18). Importantly, SAVANA yields results highly concordant with the Illumina estimates when run using a matched germline control sample as well as in tumour-only mode, thus showing the potential of SAVANA to deliver meaningful results for clinical samples.

2. The rationale around the MSI influence (lines 271 to 275) is sound and I generally agree that probably it is not the main factor for the discrepancies between the benchmarked software. However, what evidence authors provide for the absence of inactivation of MMR genes? There are potentially inactivating point-mutations (small SNVs), which may cause gene disruptions, not to mention the epigenetic methylation events. Did you try to assess the mutation status or the epigenetic state of MMR genes? The wording in “we did not detect inactivation of MMR genes or MSI using matched Illumina WGS data for the tumours analysed here” sounds to me as very strong statement, without evidence in the paper.

We investigated the presence of germline and somatic mutations in MMR genes in all cases using the matched Illumina WGS data, including point mutations, indels, SCNAs and SVs. In addition, we analysed all Illumina samples for the presence of microsatellite instability following best practices that have been widely used by us and others to detect MSI in tumour sequencing data sets(19, 20). Moreover, we did not detect evidence of *MLH1* promoter hypermethylation in any of the samples using the long-read data, which is the most common epigenetic aberration leading to MMR gene inactivation in MSI tumours(20). Consistent with previous reports on the very low rates of MSI in soft-tissue and bone sarcomas, as well as in glioblastomas, we did not detect evidence of MSI in any of the samples(19, 20). As a result, the very low rates of insertions and deletions at microsatellites (that is, expansions and contractions at microsatellites) detected using SAVANA are consistent with the underlying tumour biology. By contrast, the high rates of deletion and insertions at microsatellites reported by existing methods are consistent with poor algorithmic performance, as our other analyses (e.g. the read-backed phasing of SVs, **Figure R8** above) also suggest.

3. I understand that SAVANA is dependent on machine learning and hence of a large training set. However, for any clinical use, there is a requirement for reproducibility of the analysis. If the model is updated (more trained) it may not show the same results with a simple sample re-analysis. How does the authors address the absence of determinism in the further model training?

We thank the Reviewer for bringing up this important point. We note that the predictive model itself is deterministic in that, once trained, the model will always generate the same output for a given input. That is, a model already trained will always classify a given SV as somatic or not every time the model is run.

Predictions might indeed change if a model is updated. This is in essence equivalent to the situation when a given algorithm (even when not relying on machine learning) is updated or improved. This is precisely a major motivation to comply with best coding practices and version control software, which we have followed throughout the project using GitHub, where we provide the trained models for each version of SAVANA and functionalities for users to train their own models if desired.

Finally, we note that SAVANA provides functionalities to process the output VCF files using hard filters to identify somatic SVs. While we recommend using the machine learning model due to increased performance, we have shown that hard filters also deliver good results (see e.g. the results for PacBio data we report in the manuscript).

Minor issues:

1. Line 92 and 93: “as well as infer tumour purity and ploidy, using long read sequencing data, including the estimation of tumour purity and ploidy.” This sounds as tautology, please decide on use of “infer” or “estimate”.

We have now amended this error.

2. There is slight discrepancy in the nomenclature of true positive, false positive, Null, Both as described in lines 193 to 202 and the corresponding Suppl. Figure 4. Please, stick to the MCP model categories or the commonly established definitions, but not both. E.g. “Noise” or “false positive”?

We have now revised this point.

3. In my version of the PDF (on two operational systems) it was not possible to see the equations on line 556 and 572, making the assessment of the final tumour purity or ploidy model very hard. Please, correct the font or the symbols.

We have now amended this error.

4. It was not clear to me how Supplementary Figure 19 was produced and in what computing environment/configuration. No explanation of the hardware or methodology is given at the present. Is there accountability for single/multi core steps in the various packages? Is it in real or system time? Perhaps, a small section in the Methods and a longer figure legend can address this issue.

To assess the computational performance of algorithms, each caller was run on the cohort of split tumour BAMs on a SLURM scheduler, where 16 CPUs were made available for multiprocessing (if applicable). Callers that allowed threads to be set via command-line

argument (SAVANA, Severus, NanomonSV, Sniffles2, SVision-pro, and cuteSV) were set using the relevant thread argument. Each caller was assessed on total execution time and maximum memory. While SAVANA used the second-highest amount of memory, it was significantly faster than all other algorithms. We have now revised the caption to clarify this point.

Reviewer #1 (Remarks on code availability):

I can verify that the code is easily accessible and available with few mouse clicks. The software installation is possible, and the tool can be set and run without any advanced knowledge through the Conda environment (or can be built from the source with a relatively small effort). The project seems very well managed, and the work looks solid. Although, I have the technical expertise to review all source lines and the logic, going through everything will consume too much of my time, so I haven't performed an extensive review.

We thank the Reviewer for these positive comments on the accessibility and ease of use of SAVANA.

Reviewer #2

The author presented a machine learning based tool to detected somatic SVs and SCNA from long-read sequencing data. Long-reads have been widely used in analysis healthy genomes, such as the human pagenome reference project, but limited to cancer studies. From the methodology aspect, this study tried to solve two major issues in detecting somatic SVs: 1) distinguish real somatic breakpoint from mapping artifact; and 2) difficulties in assessing SV detection specificity. Besides, the 99 tumor-normal matched long-read sequencing data will be a valuable resource to the community for future methods development. Overall, the benchmark and analysis are comprehensive, but the advantages of using long-reads are not well described and should be further improved.

We thank the Reviewer for these positive comments. Regarding the advantages of long-read sequencing, please see our response above in the section "General Comments".

Questions and comments:

1. As for the machine learning model, both the SV features and random forest model have been used for SV classification with short-reads, such as the tool forestSV published in 2012. Though the model is not novel, the author provides the option to train personalized models, which is a contribution to the community.

We thank the Reviewer for pointing us to forestSV, which we now cite in our manuscript.

I have the following questions regarding to this model:

a) The machine learning model used to distinguish real breakpoint from artifact was trained by SVs outside of repetitive regions, blacklist regions and amplified regions (>200 reads). Since repeat regions are know hotspot for SVs, I am wondering how SAVANA achieves somatic breakpoint detection at repeat regions (such as centromere) with a model that trained on selected regions.

We thank the Reviewer for the opportunity to clarify this important point. As the Reviewer correctly points out, we initially decided to only include non-repetitive regions in the training data. The underlying rationale was that the features we use for distinguishing somatic SVs from mapping/sequencing errors would be comparable across repetitive and non-repetitive regions. Although the high performance of SAVANA and our manual inspection of hundreds of SVs suggest that this assumption is correct, we have now included SVs in both repetitive and non-repetitive regions in the training data set. We have now clarified this point in the Methods section.

b) The model parameters, performance and training details should be provided, such as the creation of validation and test set. The top important features related to tumor support (Supplementary Figure 3) are interpretable, but not for features like "CLUSTERED_READS_NORMAL", "SVLEN", "TUMOR_ALT_HP_NA_COUNT" and "CIGAR".

The description of all the features used by the model is given in Supplementary Table 2. We have now revised the text to make this more explicit and included a reference to Supplementary Table 2 in the caption of Supplementary Figure 3. We have also expanded the discussion of the model training details in the Methods section. Finally, we provide the code used for model training and the benchmarking of algorithms in the GitHub repository of SAVANA: https://github.com/cortes-ciriano-lab/savana_manuscript_benchmarking_code/.

c) The training set was created by overlapping short-read calls and SVs detected by SAVANA with long-reads data from the 99 samples, which is the same as the samples used for prediction (detection SVs). How dose the model perform if other long-read tools are used to create the true SVs? Or train the model on well-established benchmark, such as HG002, because some of the important features also applied to germline SVs.

We thank the Reviewer for indicating that this critical point was not clear enough. We did not train the model using data from all samples and then used the trained model to detect SVs in the same samples used for training. We did not do that because, as the Reviewer correctly points out, using the same samples for testing and training would be circular reasoning and not a robust way for testing the generalization capability of the model (that is, how well the model performs on unseen samples).

In all the results we presented in the manuscript, we implemented a leave-one-tumour-out approach. In brief, we train a model using the data from all samples except for one. Then, we use the trained model to detect SVs in the left-out sample. This means that we trained 99 models, each time holding out a different sample from the training set. In this way, we ensure that data from a given sample is only used in either the training or test set. We have now revised the Main Text and Methods to clarify this important point.

The tumour types we selected for this study span diverse tumour types, lineages, anatomical locations (brain, bone, etc) and a wide dynamic range of karyotypic complexity and SV rates. Moreover, the samples come from two biobanks from two different countries (Royal National Orthopaedic Hospital in London and the Lisbon Academic Medical Center in Portugal). Therefore, this diverse collection of tumour samples has allowed us to show the uniformly high performance of SAVANA across tumour types, data sets and institutions. We also note that there are still no large, high-coverage data sets with matched normal controls for clinical samples available, hence the high value of our data set as well.

2. The author evaluates the breakpoint concordance between each caller and the true SVs from COLO829, here are my questions:
 a) The author used SVIM and cutesv (they are not designed for somatic SV detection) to call tumor and normal separately and subtract the coordinates. While tools like SAVANA, nanomosv use matched normal-tumor samples, it is a biased comparison for SVIM and cutesv.

This point has also been raised by Reviewer 1 – please see our response to Reviewer 1 on pages 16-17 of this document about the use of germline callers in our benchmarking analysis.

b) Is this evaluation distinguish different types of SVs or just compare every detected breakpoints. For example, a somatic deletion contains two breakpoints, how dose the author calculate the precision and recall for this deletion? It is also critical to evaluate the whole event, especially for a complex events that containing more than two breakpoints. The breakpoint shift evaluation is not enough to show the performance.

SAVANA evaluates each breakpoint separately. This is necessary for detection purposes as supporting reads have primary alignments for only one breakpoint and clipped alignments for the other. In practice, this means that primary alignments supporting a given SV are assigned to one of the two breakpoints that make up the SV. Next, SAVANA identifies which two breakpoints make up each SV (in the case of single breakends only one breakpoint can be reliably identified, which is indicated in the output VCF file following the VCF 4.2 specifications).

Deletions can be detected as gapped alignments, and thus reflected in the cigar string as “D”, or through the detection of clipped alignments. In this latter case, each SV is defined by a pair of breakpoints. A major challenge for benchmarking SV callers is the use of arbitrary output formats and terminology that does not conform with the VCF 4.2 specifications. In fact, we have spent a great deal of effort to bring the output format of some of the SV callers benchmarked to a common SV representation scheme following best practices specified by PCAWG (some of us were deeply involved in the SV working group of PCAWG, where we also faced significant challenges to harmonize the output formats of VCF callers). This effort has allowed us to perform a fair comparison across callers– we are proud to note that SAVANA fully complies with the VCF 4.2 specifications to facilitate the unambiguous interpretation and reference to SV calls in the future.

As for the Reviewer's question about how we evaluate recall and precision, we have considered breakpoints in all the benchmarking analyses, and group breakpoints based on the SV types they support when performing comparisons. We have now revised the Methods and Main text to clarify these points.

We note that the comparison of SV calls between Illumina and long-read data sets is also complicated by additional notation issues. For example, the same duplication may appear as an insertion in long-read data due to differences in the reads alignment to the reference genome. While we have invested substantial effort to address these issues and fix inconsistencies in SV notation across algorithms, we considered that comparing the SV calls at the breakpoint level would be more reliable.

The SVs involved in complex events are considered separately. That is, if e.g. a chromothripsis event is made up of ten SVs, we consider them as ten datapoints for benchmarking purposes rather than as a single unit. The reason is that complex events often span large regions of the genome, cannot be fully reconstructed, involve tens to hundreds of SVs, and multiple mutational processes might have rearranged the same genomic region, all of which makes it very difficult to determine which SVs exactly make up each complex event(21–23). For example, only complex SVs involving less than six SVs were considered for analysis in the flagship SV paper of PCAWG and those involving more than six SVs were considered too complex for representing them as a single event(21).

c) Looking at the absolute value shown in the barplot of Supplementary Figure 7c, the recall of SAVANA, Nanomonsv and Severus are very close, why SAVANA is significantly higher than those callers.

We performed bootstrapping to compare the differences in performance across callers. The difference in recall between SAVANA, and Nanomonsv and Severus is statistically significant even if the effect size is not as marked as in other analyses. We note that this is likely the case because the COLO829 truth set only contains 68 somatic SVs (as discussed above, we have only included the COLO829 data set because it has been the most commonly used truth set for benchmarking SV detection methods so far). Finally, we would like to emphasize that the effect size is small for recall only - the effect size for precision is large.

d) SVision-pro also uses the matched tumor-normal samples as input, the author should also assess this deep learning based tool.

We have now included SVision-pro in all the benchmarking analyses we have performed. Overall, these additional analyses have shown that SAVANA shows significantly higher performance than SVision-pro across SV types, clonality levels and data sets.

e) Looking at the boxplot in Figure 2b, SAVANA shows a big variation in terms of the number of SVs detected in all replicates. Would this be a problem?

We thank the Reviewer for the opportunity to clarify this point and for noticing this important result. The variation in the number of SVs detected by SAVANA in one or both replicates is definitely not a problem. On the contrary, Figure 2b shows that SAVANA only detects a very small number of SVs in a single replicate (green boxplot in Figure 2b), which correspond to false positive calls. Importantly, SAVANA detected one order of magnitude less SVs in just a single replicate than the other algorithms. These results agree with the very low number of false positives detected by SAVANA in the replicate analysis of normal (i.e. non-neoplastic blood) samples reported in Figure 3. We have now revised the main text and Figure 2 to more clearly illustrate these points.

3. The usage of replicates to benchmark could reflect the performance to some extent. While the statement "true somatic SVs should be detected in all replicates ..." is suspicious because a method could have systematic errors for detecting SVs at certain regions due to alignment bias. For example, ONT reads usually aligned poorly at regions of high VNTR copies, where you could find several continuous variant signatures (such as 'D' in CIGAR string) from the alignments. Here are my questions and suggestions regarding this replicates based evaluation:

a) The false discovery rate of SAVANA should be assessed on normal tissue (i.e., germline SVs) to understand its potential bias or at least show how many SVs detected in all replicates might be false, which could better support the results shown in Figure 2.

We agree with the Reviewer that some of the SVs that are called in both replicates in the tumour analysis (Figure 2) might be false positives that are called in both flow cells independently due to e.g. alignment bias, as the Reviewer points out. Analysis of the SVs detected in both replicates did show an enrichment in microsatellites (tandem repeats) that are inconsistent with the underlying tumour biology, as high rates of mutations at microsatellite regions are driven by mismatch repair deficiency, which is uncommon in the tumour types studied here (sarcomas and glioblastomas) and we did not find evidence of mismatch repair inactivation in any tumour (see also our response to Reviewer 1 on this topic above).

We have now also analysed whether the SVs detected in just one or both replicates show consistent read-backed phasing patterns (similar analysis to **Figure R8; now included in Figure 2g-h**). As can be seen in **Figure R16** below, the SVs detected by SAVANA in both replicates are supported by reads phased to a single parental allele (shown in green), whereas a large number of the sequencing reads supporting the SVs detected in both or a single replicate by other algorithms show inconsistent read-backed phasing (red). **While SAVANA only detects inconsistent read-backed phasing in 0.15% of the SVs detected in both replicates across the cohort on which we can reliably perform this analysis (19/12749), 25% of the SVs detected by Sniffles (166 times higher as compared to SAVANA) in both replicates show significant evidence of inconsistent read-backed phasing.**

Overall, these results illustrate the important point raised by the Reviewer. Moreover, these new results reinforce the robust and higher performance of SAVANA and illustrate the value of using sequencing replicates for benchmarking SV detection methods.

Figure R16. Analysis of the phasing consistency using of SV support reads using read-backed phasing for the SVs detected in sequencing replicates. **a:** results SVs detected in one replicate only. **b:** results SVs detected in both replicates. Similar analysis to the right panel in **Figure R8**.

In addition, to further study alignment biases, we performed the replicate analysis on matched normal samples (**Figure R17**; results also reported in Figure 3 in the manuscript). In this analysis, we split the sequencing reads from the matched non-neoplastic samples used as germline controls (blood for all the cases in our cohort) into two non-overlapping sets of reads. Next, we consider one of the sets as the “tumour” sample and the other as the “normal” sample. The key idea is that we should not call any somatic SVs because both the “tumour” and “normal” samples in this scenario are essentially the same. This analysis allowed us to quantify the false positive rate for each caller in a data-driven manner, and to show that SAVANA has the lowest false positive across all algorithms benchmarked.

Figure R17. Number of somatic SVs detected in the normal vs normal replicate analysis (also shown in Figure 3 in the manuscript). Each dot represents a tumour.

b) All results in Figure 2 are based on identical SVs detected in replicates. How SVs are considered as the same event among all replicates? Dose it only consider breakpoint?

The identity of SVs is based on the detection of breakpoints in both replicates. We consider that the same breakpoint is detected in both replicates within 100bp of each other. We have now revised the manuscript to clarify this point. We decided to use this approach due to the strong inconsistency in how SVs are reported by different algorithms discussed above in response to **point b** raised by this Reviewer.

c) Why Figure 2d dose not have SVIM and cuteSV? What is the coverage of each replicate?

We could now include SVIM and cuteSV in the analysis reported in Figure 2d because these two algorithms do not report the ID or number of reads supporting each SV, which prevents computation of allele fraction values for breakpoints.

As for the sequencing coverage of each replicate, we split the sequencing reads in each tumour BAM file into two non-overlapping sets of reads, which resulted in equal coverage in each replicate. The sequencing depth for all tumours is provided in Supplementary Table 1.

d) The observation of deletion enrichment at microsatellite is strange. Microsatellite sequence is essentially tandem repeats, I suggest to add background distribution of SVs on microsatellite.

We thank the Reviewer for indicating that this point was not sufficiently clear. We indeed use the term microsatellite to refer to tandem repeats. Our bias toward the term microsatellite likely originates from its popular use in the cancer research community due to the importance of microsatellite instability as a predictive biomarker for immune checkpoint blockade therapy. To prevent confusions about this term, we have now revised the text to explicitly explain that we refer to tandem repeats when using the term microsatellite. Somatic SVs at tandem repeats (that is, expansions or contractions of tandem repeats) are only frequent in tumours with mismatch repair deficiency. We and others have studied the rates and types of microsatellite mutations across diverse cancer types using genome sequencing data(19, 20). Notably, in tumours with mismatch repair *proficiency*, the rates of SVs at microsatellites are very low, usually zero(19, 20). Therefore, in mismatch repair proficient tumours the expected rate of SVs at microsatellites (especially with the read lengths that can be achieved currently with long-read sequencing methods) should be very low or zero, as reported by SAVANA. We note that tandem repeats can be involved in SVs other than repeat expansions and contractions. However, the high rates of SVs at tandem repeats reported by some algorithms in our benchmarking analysis are indeed small deletions and insertions (i.e. expansions and contractions of microsatellites), which, as indicated above, are inconsistent with the biology of mismatch repair proficient tumours. Finally, we note that microsatellites are polymorphic across individuals (that is, the number of repetitive units varies across individuals). However, we note that the polymorphic nature of microsatellites does not affect somatic detection of SVs at microsatellites when using a matched normal reference as germline control, as we and others have shown(19, 20).

Together, all these results and considerations, in combination with the other analyses in our study (e.g. the replicate analysis on matched normal samples shown in Figure 3 and the read-backed phasing analysis discussed in the “General Comments” section above) indicate that the SVs at tandem repeats identified by existing algorithms are false positive SV calls.

4. It is great to see the comparison of short-read and long-read data on clinical samples. Phasing tumor genome and detecting complex SVs are the advantage of using long-read. Here are my questions:
a) SAVANA shows the highest concordance with short-reads (Supplementary Figure 20-21), then why we should sequence tumor with long-reads.

Please see our response in “General Comments” above on the advantages of long-read sequencing over short reads, and the new tumour-only mode functionality of SAVANA.

b) How SVs are determined to be detected by both short-read and long-read data. 85% of SVs are detected by both technologies, what are those only detected by long-read data?

Please see our response in “General Comments” above on the analysis of SVs detected by long reads only.

c) In Figure 5d-f, copy number variation can be direct cause of suppressor gene loss, how to decide if this complex rearrangements are the causes of suppressor gene copy number loss? In other words, with long-read data and phased information, is that possible to show the copy number loss is associated with rearrangement and they are not independent events?

We thank the Reviewer for this excellent question. In fact, one of the key applications of long-read sequencing that we are exploiting with our clinical colleagues is to determine whether SVs affecting tumour suppressor genes affect one or both alleles. Because SAVANA reports which sequencing reads support each SV at the haplotype level, it is possible to evaluate whether the reads supporting a given (complex) SV affect one or two alleles. This analysis can be simply performed by investigating whether all reads originate from the same allele, which would be consistent with the SV affecting one parental allele, or through allele-specific genome assembly. See also **Figures R8 and R16** in this document.

d) Similar figures like Figure 5d-f are too complex and hard to capture the main point that the author wants to show, is there a way to simplify this? And what is the blue line at the bottom, is that copy number loss?

The panels in Figure 5d-g and similar ones represent somatic copy number and rearrangement profiles. We used the package ReConPlot(24) that we recently developed to generate this type of plots, which have been popular in the cancer genomics community to study complex genomic rearrangements; see e.g. reference (25) or the PCAWG flagship paper(22).

The blue line in the plot represents the copy number of the least amplified allele, usually referred to as minor copy number. For example, in a region with loss of heterozygosity, the blue line would be at zero. This is explained in the caption of all figures showing ReConPlots.

The point we want to highlight in Figure 5d-f is that SAVANA permits the reconstruction of complex rearrangements spanning a wide dynamic range of complexity, ranging from events involving a few SVs (Figure 5d) to events involving hundreds to thousands of rearrangements (Figure 5f).

e) What is the total percentage of somatic SVs could be phased and phased blocks N50 ?

In brief, we implemented a phasing pipeline that consists of detecting germline heterozygous SNPs in Illumina WGS data from the matched normal sample (blood) followed by phasing using WhatsHap(26) and the normal long-read WGS data. Using this approach, we identified a median of 4,271 phasing blocks per sample, with an average of 561 heterozygous SNPs per block. The median N50 of the phased blocks was 1.18 Mb, ranging from 0.22 Mb to 22.7Mb. In total, 45.3% of the alignments were successfully assigned to a haplotype, enabling the confident phasing of 60.7% of the SVs (27,702 out of 45896) detected by SAVANA across the cohort. SV phasing was determined using the criterion that at least three SV supporting reads were assigned to a single haplotype, with no supporting reads assigned to the alternative haplotype.

See also our response in the “General Comments” section above on the use of phasing information to benchmark SV detection algorithms.

f) The assembly used all SV supporting reads as described in the methods. Why the assembled contig dose not span some of the breakpoint shown by the vertical lines. Moreover, are you able to identify the same breakpoint when you align it back to the reference?

The initial plot included all the SVs. However, the assembly was restricted to a phasing block, which typically spans a median size of 1.18 Mb. The pipeline was initially designed to assemble regions within the same phasing block to ensure allele resolution (that is, to perform genome assembly using sequencing reads assigned to the sample parental allele), but this approach limited assembly contiguity as SVs can link regions of the genome from different phasing blocks in the same or even different chromosomes. In such situations it is not possible to identify phase switching errors with the contiguity that can be achieved with the read lengths generated by long-read sequencing.

We have now reviewed cases where complex rearrangements affected contiguous phasing blocks and ensured that only one allele was involved. In these instances, specifically in *CDK4-MDM2* co-amplification (**Figure R18**) or chromothripsis event (**Figure R19**), we incorporated all relevant phasing blocks to improve contiguity.

Regarding the alignment back to the reference genome, we verified the correspondence between the breakpoints identified by SAVANA and the assembled contigs. The median

distance between SAVANA breakpoints and those derived from the assembly was 0 bp (range: 0–861 bp), confirming strong agreement between the methods.

Figure R18. (a) Somatic SV and somatic copy number profiles for chromosomes 3 and 12 for SARC-018. **(b)** The plot shows the alignment of the assembled contig for each parental allele (arbitrarily labelled as “allele 1” and “allele 2”) to chromosomes 3 and 12 of the human reference genome. Allele-specific genome assembly was performed using sequencing reads from a single allele of a haplotype block identified using WhatsHap.

Figure R19. (a) Somatic SV and somatic copy number profiles for chromosome 5 for GBM-009.

(b) The plot shows the alignment of the assembled contig for each parental allele (arbitrarily labelled as “allele 1” and “allele 2”) to chromosome 5 of the human reference genome. Allele-specific genome assembly was performed using sequencing reads from a single allele of a haplotype block identified using WhatsHap.

5. I value the efforts have been made to compare every events and provide them as examples in the supplementary.

a) Figure 4 is strange to me. It looks like that SAVANA only detected breakpoints roughly in range 75Mb-90Mb, and only a duplication at 150Mbp that is also detected by GRIDSS, Severus and NanomonSV. What are those deletions detected by callers like Severus and NanomonSVs, since most of them looks condordant between these two callers.

We thank the Reviewer for pointing out that the example shown in Figure 4 was confusing. The point we wanted to make with this example is that both GRIDSS and SAVANA correctly identify the SVs mapping to the amplified region, and thus supported by SCNAs, but not in genomic regions where no SCNAs are detected. The many deletions detected by Severus

and NanomonSV are not supported by SCNAs and are not identical. They only seem to align because of the resolution of the representation we chose. We recognize that this example might not be ideal to highlight the high false positive rate of callers like Severus and NanomonSV. Therefore, we have included now a different case for clarity:

b) In supplementary Figure 15b, SAVANA, GRIDSS and NanomonSV detected lots of interspersed duplication (blue lines), but these duplication breakpoints on the left side seems to be highly concordant with deletions (orange lines). If the deletion are flanked by large identical repeats, it is possible to detect both deletion and duplication? I suggest to show the raw alignments of ONT reads around some breakpoints.

The event shown in Supplementary Figure 15b corresponds to a breakage-fusion-bridge (BFB) cycle with chromothripsis. Mechanistically, such events are characterised by interleaved patterns of SVs with an even distribution of SV types that reflect the stitching of DNA shards arising from chromosome fragmentation caused by chromothripsis in random order and orientation(27). In practice, this means that in a chromothripsis event there is a comparable number of deletion-like (orange lines), duplication-like (blue) and inversions (green and black) mapping to the region affected by chromothripsis. This is precisely what is shown in Figure 15b.

Very important from a mechanistic standpoint, only GRIDSS and SAVANA (and to a lesser extent NanomonSV) show the even distribution of SV types, thereby allowing accurate interpretation of the underlying biology and inference of the rearrangement mechanisms that affected this tumour. The inaccuracy and high false positive rate of the other callers obscure the rearrangement pattern, thus hampering a biologically sound interpretation of the rearrangement profile and the identification of the biological mechanisms that shaped the cancer genome in this tumour.

Reviewer #2 (Remarks on code availability):

The software is easy to install and I could run without error on some test data. Here are my comments and questions for running the SAVANA:

a) There are several different evaluation in this study, but it is unclear how SAVANA produces those results. Please provide every steps in different evaluation either on Github or in the paper.

We have now included the code used for benchmarking algorithms in the GitHub repository of SAVANA: https://github.com/cortes-ciriano-lab/savana_manuscript_benchmarking_code/

b) Following the experimental setting shown in Figure 3a, I tested on one of the HGSC sample HG00733 (HiFi reads from https://ftp.1000genomes.ebi.ac.uk/vol1/ftp/data_collections/HGSC3/working/20231126_UW_HiFi/HG00733/). The tumor and normal input BAM files for SAVANA are exactly the same and aligned with minimap2. I only used the chr1 alignment as a test, while it gives me around ~10,000 SVs (records in the *.bedpe file). I also tried to classify with 'savana classify

```
--vcf HG00733.sv_breakpoints.vcf --pb --output HG00733.sv_breakpoints.cly.vcf --somatic_output', but the results are the same. I might miss some steps, could the author explain how they did for the results in Figure 3a.
```

We thank the Reviewer for their question about running SAVANA. In the second command specified, the `somatic_output` flag should have an argument to a new VCF file. This is the file that will contain somatic SVs. The `bedpe` file is generated as a first step of SAVANA and contains all breakpoints without any classification applied.

To run an example of an experiment we ran for Figure 3a (Figure 3b in the current version of the manuscript) on PacBio data, we used the publicly available COLO829 PacBio data which we use in the manuscript and is aligned to hg38 using the following command:

```
savana --tumour COLO829-BL.pacbio.hg38.bam --normal COLO829-BL.pacbio.hg38.bam --length 32 --outdir <outdir> --ref <ref> --pb --contigs <example/contigs.chr.hg38.txt> --threads 16
```

This generates the bedpe file mentioned by the Reviewer, indeed containing many lines. However, somatic SVs are found in the file ending with "classified.somatic.vcf" which, in our test, contains four lines (removing the header, "grep -v "^#" | wc -l"), representing four breakpoints which correspond to two SVs. In the analysis for Figure 3a (Figure 3b in the current version of the manuscript), the counts show the number of SVs.

If it was preferred to view somatic SVs in bedpe format, this file could be filtered with the somatic callset using a bash command such as:

```
"vgrep <sample>.classified.somatic.vcf | awk '{print $3}' | cut -d '_' -f1,2 | xargs -l{} grep "{}" <sample>.sv_breakpoints.bedpe"
```

We thank the Reviewer for raising this point of potential confusion and have added functionality so that a somatic bedpe file is also generated alongside the somatic VCF. This functionality will be available in the next release of SAVANA.

Reviewer #3

Reviewer #3 (Remarks to the Author):

In this study, the authors described a novel algorithm called SAVANA for the detection of somatic SVs and SCNAs and the estimation of tumour ploidy and purity in long-read sequencing data. The main focus of this study was to compare the performance of SAVANA against existing SV callers developed for long-read sequencing data. They generated matched high-depth nanopore (51X) and Illumina WGS (118X) data from different tumour types and corresponding germline samples. They trained a random forest classifier with a set of SVs detected by SAVANA in long-read data that were labelled as true somatic SVs if they were also detected in short-read data (by a different pipeline). In addition SAVANA uses Mondrian Conformal Prediction to assess the reliability of individual SV calls.

When comparing SAVANA against existing SV detection algorithms (Sniffles2, CuteSV, SVIM, NanomonSV and Severus), the number of SVs detected was highly variable. In particular, when they used a truth set of 68 somatic SVs detected in a melanoma cell line (also experimentally validated) SAVANA showed higher recall and specificity compared to the other algorithms. In order to assess the performance of these SV callers in an unbiased manner and across the entire genome, they simulated technical sequencing replicates by randomly splitting the reads in two BAM files and called SVs on each replicate. SAVANA not only showed higher concordance across replicates but also across different clonality levels, SV sizes and SV types.

Compared to SAVANA, the other algorithms detected much more insertions and deletions. The authors found that most of these insertions and deletions detected by those algorithms mapped to microsatellites. Considering that the samples used in this study are not MSI, the authors concluded that those insertions and deletions were most likely false positive calls derived from sequencing or mapping errors.

In order to assess the specificity of these algorithms, the authors generated simulated technical sequencing replicates also for the matched germline controls and used one replicate as tumour and the other as matched germline control. Notably, SAVANA showed a significant lower false positive rate compared to the others (which showed much higher detection of false positive SVs (9X to 391X higher). The authors concluded that the higher SV detection rate of the other algorithms was most likely due to lower specificity compared to SAVANA.

In addition 85% of SVs detected in the illumina data by SAVANA were also present in the long read data while the recall rate of the other algorithms was much lower. At the end, the authors showed very briefly that SAVANA can also reliably call SCNA, estimate tumour purity and ploidy and identify SVs involved in the inactivation of tumour suppressor genes or activation of oncogenes.

Comments:

The method is novel and timely. Overall, the benchmarking of the method is sound, with evidence of SAVANA outperforming existing methods. However additional evidence would further support the robustness of the method:

We thank the Reviewer for these positive comments and for recognizing the value and thoroughness of our benchmarking analyses.

1. In order to train their model, the authors used a set of true SVs which was derived from SVs detected by a short-read WGS data pipeline in the illumina data that were also detected by SAVANA in the long-read sequencing data. Then they classified the SVs detected in each tumour using a random forest model trained on the SVs detected in all other tumours in the cohort (leave-one-tumour-out models). This approach ensures that no data from the tumour investigated was used to train the model. Although this approach is valid as an “internal validation” of the

model, a completely independent long-read sequencing dataset, exploring also other cancer types, would further strengthen the validity of their approach.

Please see our response to Point c raised by Reviewer 2.

2. The authors showed that existing algorithms have lower specificity and higher detection rate of false positive SVs compared to SAVANA (measured in simulated sequencing replicates of blood WGS data). Could the authors show the fraction of insertion and deletions among those false positive SVs? And if there are any recurrent ones or overlapping with microsatellites? This could be a further evidence that the higher number of insertions and deletions detected by the existing algorithms are most likely false positive as stated in lines 275-277.

We thank the Reviewer for suggesting this analysis, which we have now included in our manuscript. As we report below (**Figure R20**), A large fraction and absolute number of the SVs detected in the replicate analysis of normal samples map to microsatellites. See also our response to point 2 raised by Reviewer 1 and point 3 raised by Reviewer 2 in relation to the further support showing that these are false positive calls.

Figure R20. Fraction of SVs detected in the replicate analysis of blood samples that map to microsatellites (shown in green).

3. Figure 2d shows that SAVANA has a consistent detection rate of somatic SVs in both simulated sequencing replicates when evaluated in different allele fractions compared to existing algorithms, thus allowing this type of analysis in low tumour purity samples. Have the authors tested SAVANA in such samples, i.e. FFPE or ctDNA samples?

We thank the Reviewer for pointing out these important applications. We are indeed actively working on the application of nanopore sequencing to cfDNA and FFPE samples.

In the case of cfDNA samples, detection of SVs is very limited for two major reasons. First, cfDNA fragments are short (around 167bp). Although nanopore sequencing allows the analysis of the small fraction of longer fragments found in plasma, the vast majority of fragments are short, and even shorter in cancer samples. In addition, the coverage that can be obtained per cfDNA sample is usually below 1x, which in practice means that for most of the genome we either do not have sequencing reads or just one. With such low coverage and short fragments it is not possible to reliably detect somatic breakpoints in the same way as we do for tumour samples. We anticipate that as protocols for cfDNA analysis using nanopore sequencing improve we might be in a better position to detect SVs. However, at present it is only possible to detect high level amplifications and large copy number aberrations based on read depth. We have recently reported unpublished data from our team in various conferences, see e.g. the presentation by our team at the Nanopore London calling conference here: <https://nanoporetech.com/blog/news-blog-liquid-biopsies-multi-modal-cell-free-dna-assays-using-nanopore-sequencing-0>

As for the analysis of FFPE samples, this is really the next frontier and real-world test for long-read sequencing methods. Currently, the DNA adducts induced by sample fixation prevent reliable sequencing of FFPE samples using either Nanopore or PacBio sequencing. One study(28) earlier this year showed that nanopore sequencing of FFPE samples is possible, but still the sequencing depth obtained was below 1x, which, as discussed above for cfDNA, is not high enough to enable reliable detection of SVs.

Despite these challenges, we are very actively working on the application of long-read sequencing to FFPE samples in the context of the Cancer Grand Challenge SAMBAI (Social, Ancestry, Molecular and Biological Analysis of Inequalities)

<https://www.cancergrandchallenges.org/sambai>. Specifically, we are working on the development of protocols for the application of long-read sequencing to real-world clinical samples collected and processed in Africa. As the Reviewers correctly pointed out, these samples usually lack matched germline controls, for which the tumour-only mode of SAVANA will be essential.

In sum, although these are important areas in which we are trying to make an impact, there are no datasets available yet of sufficient quality and scale for us to currently show the performance of SAVANA in these areas. Thus, we consider that, although important, the application of SAVANA to cfDNA and FFPE samples falls beyond the scope of this manuscript.

4. Overall this study shows that SVs detected by SAVANA are highly consistent to those detected by Illumina short read sequencing. It would be nice if authors could show if any new SVs, potentially biologically relevant, were detected by SAVANA in long read sequencing data that were missed by short read sequencing data.

Please see our response in “General Comments” above on the advantages of long-read sequencing over short reads.

Minor comments:

1. In line 91 “To address these needs, we present SAVANA, a computationally efficient algorithm specifically designed to detect both somatic SVs and SCNAs, as well as infer

tumour purity and ploidy, using long-read sequencing data, including the estimation of tumour purity and ploidy”. Please revise this sentence.

We have now revised this sentence.

2. In line 397, authors might wrongly refer to figure 3b instead of 3c.

We have now revised this sentence.

References

1. V. Marx, Method of the year: long-read sequencing. *Nature Methods* **20**, 6–11 (2023).
2. I. Cortes-Ciriano, C. D. Steele, K. Piculell, A. Al-Ibraheemi, V. Eulo, M. M. Bui, A. Chatzipli, B. C. Dickson, D. C. Borchering, A. Feber, A. Galor, J. Hart, K. B. Jones, J. T. Jordan, R. H. Kim, D. Lindsay, C. Miller, Y. Nishida, P. Z. Proszek, J. Serrano, R. T. Sundby, J. J. Szymanski, N. J. Ullrich, D. Viskochil, X. Wang, M. Snuderl, P. J. Park, A. M. Flanagan, A. C. Hirbe, N. Pillay, D. T. Miller, Genomic patterns of malignant peripheral nerve sheath tumor (MPNST) evolution correlate with clinical outcome and are detectable in cell-free DNA. *Cancer Discov.*, CD-22-0786 (2023).
3. D. Hodzic, C. Kong, M. J. Wainszelbaum, A. J. Charron, X. Su, P. D. Stahl, TBC1D3, a hominoid oncoprotein, is encoded by a cluster of paralogues located on chromosome 17q12. *Genomics* **88**, 731–736 (2006).
4. C. L. Bohrson, A. R. Barton, M. A. Lodato, R. E. Rodin, L. J. Luquette, V. V. Viswanadham, D. C. Gulhan, I. Cortés-Ciriano, M. A. Sherman, M. Kwon, M. E. Coulter, A. Galor, C. A. Walsh, P. J. Park, Linked-read analysis identifies mutations in single-cell DNA-sequencing data. *Nature Genetics* **51**, 749–754 (2019).
5. C. Vermeulen, M. Pagès-Gallego, L. Kester, M. E. G. Kranendonk, P. Wesseling, N. Verburg, P. de Witt Hamer, E. J. Kooi, L. Dankmeijer, J. van der Lugt, K. van Baarsen, E. W. Hoving, B. B. J. Tops, J. de Ridder, Ultra-fast deep-learned CNS tumour classification during surgery. *Nature* **622**, 842–849 (2023).
6. Y. Shiraishi, J. Koya, K. Chiba, A. Okada, Y. Arai, Y. Saito, T. Shibata, K. Kataoka, Precise characterization of somatic complex structural variations from tumor/control paired long-read sequencing data with nanomonsv. *Nucleic Acids Res.* **51**, e74 (2023).
7. A. Keskus, A. Bryant, T. Ahmad, B. Yoo, S. Aganezov, A. Goretsky, A. Donmez, L. A. Lansdon, I. Rodriguez, J. Park, Y. Liu, X. Cui, J. Gardner, B. McNulty, S. Sacco, J. Shetty, Y. Zhao, B. Tran, G. Narzisi, A. Helland, D. E. Cook, P.-C. Chang, A. Kolesnikov, A. Carroll, E. K. Molloy, I. Pushel, E. Guest, T. Pastinen, K. Shafin, K. H. Miga, S. Malikic, C.-P. Day, N. Robine, C. Sahinalp, M. Dean, M. S. Farooqi, B. Paten, M. Kolmogorov, Severus: accurate detection and characterization of somatic structural variation in tumor genomes using long reads. *medRxiv*, doi: 10.1101/2024.03.22.24304756 (2024).
8. D. Heller, M. Vingron, SVIM: structural variant identification using mapped long reads. *Bioinformatics* **35**, 2907–2915 (2019).

9. T. Jiang, S. Liu, S. Cao, Y. Wang, Structural Variant Detection from Long-Read Sequencing Data with cuteSV. *Methods Mol. Biol.* **2493**, 137–151 (2022).
10. T. Rausch, T. Zichner, A. Schlattl, A. M. Stütz, V. Benes, J. O. Korbel, DELLY: structural variant discovery by integrated paired-end and split-read analysis. *Bioinformatics* **28**, i333–i339 (2012).
11. A. Fujimoto, J. H. Wong, Y. Yoshii, S. Akiyama, A. Tanaka, H. Yagi, D. Shigemizu, H. Nakagawa, M. Mizokami, M. Shimada, Whole-genome sequencing with long reads reveals complex structure and origin of structural variation in human genetic variations and somatic mutations in cancer. *Genome Med.* **13**, 1–15 (2021).
12. S. Wang, J. Lin, P. Jia, T. Xu, X. Li, Y. Liu, D. Xu, S. J. Bush, D. Meng, K. Ye, De novo and somatic structural variant discovery with SVision-pro. *Nat. Biotechnol.*, 1–5 (2024).
13. M. Smolka, L. F. Paulin, C. M. Grochowski, D. W. Horner, M. Mahmoud, S. Behera, E. Kalef-Ezra, M. Gandhi, K. Hong, D. Pehlivan, S. W. Scholz, C. M. B. Carvalho, C. Proukakis, F. J. Sedlazeck, Detection of mosaic and population-level structural variants with Sniffles2. *Nat. Biotechnol.*, doi: 10.1038/s41587-023-02024-y (2024).
14. M. Nattestad, S. Goodwin, K. Ng, T. Baslan, F. J. Sedlazeck, P. Rescheneder, T. Garvin, H. Fang, J. Gurtowski, E. Hutton, E. Tseng, C.-S. Chin, T. Beck, Y. Sundaravadanam, M. Kramer, E. Antoniou, J. D. McPherson, J. Hicks, W. Richard McCombie, M. C. Schatz, Complex rearrangements and oncogene amplifications revealed by long-read DNA and RNA sequencing of a breast cancer cell line. *Genome Research* **28**, 1126 (2018).
15. L. G. Martelotto, C. K. Ng, M. R. De Filippo, Y. Zhang, S. Piscuoglio, R. S. Lim, R. Shen, L. Norton, J. S. Reis-Filho, B. Weigelt, Benchmarking mutation effect prediction algorithms using functionally validated cancer-related missense mutations. *Genome Biol* **15**, 484 (2014).
16. D. Aran, M. Sirota, A. J. Butte, Systematic pan-cancer analysis of tumour purity. *Nat. Commun.* **6**, 8971 (2015).
17. P. Priestley, J. Baber, M. P. Lolkema, N. Steeghs, E. de Bruijn, C. Shale, K. Duyvesteyn, S. Haidari, A. van Hoeck, W. Onstenk, P. Roepman, M. Voda, H. J. Bloemendal, V. C. G. Tjan-Heijnen, C. M. L. van Herpen, M. Labots, P. O. Witteveen, E. F. Smit, S. Sleijfer, E. E. Voest, E. Cuppen, Pan-cancer whole-genome analyses of metastatic solid tumours. *Nature* **575**, 210–216 (2019).
18. F. Martínez-Jiménez, A. Movasati, S. R. Brunner, L. Nguyen, P. Priestley, E. Cuppen, A. Van Hoeck, Pan-cancer whole-genome comparison of primary and metastatic solid tumours. *Nature* **618**, 333–341 (2023).
19. R. J. Hause, C. C. Pritchard, J. Shendure, S. J. Salipante, Classification and characterization of microsatellite instability across 18 cancer types. *Nat. Med.* **22**, 1342–1350 (2016).
20. I. Cortes-Ciriano, S. Lee, W.-Y. Park, T.-M. Kim, P. J. Park, A molecular portrait of microsatellite instability across multiple cancers. *Nat. Commun.* **8**, 15180 (2017).
21. Y. Li, N. D. Roberts, J. A. Wala, O. Shapira, S. E. Schumacher, K. Kumar, E. Khurana, S. Waszak, J. O. Korbel, J. E. Haber, M. Imielinski, J. Weischenfeldt, R. Beroukhim, P. J. Campbell, Patterns of somatic structural variation in human cancer genomes. *Nature* **578**, 112–121 (2020).

22. ICGC/TCGA Pan-Cancer Analysis of Whole Genomes Consortium, Pan-cancer analysis of whole genomes. *Nature* **578**, 82–93 (2020).
23. I. Cortés-Ciriano, J. J.-K. Lee, R. Xi, D. Jain, Y. L. Jung, L. Yang, D. Gordenin, L. J. Klimczak, C.-Z. Zhang, D. S. Pellman, PCAWG Structural Variation Working Group, P. J. Park, PCAWG Consortium, Comprehensive analysis of chromothripsis in 2,658 human cancers using whole-genome sequencing. *Nat. Genet.* **52**, 331–341 (2020).
24. J. Espejo Valle-Inclán, I. Cortés-Ciriano, ReConPlot: an R package for the visualization and interpretation of genomic rearrangements. *Bioinformatics* **39** (2023).
25. Y. Li, C. Schwab, S. Ryan, E. Papaemmanuil, H. M. Robinson, P. Jacobs, A. V. Moorman, S. Dyer, J. Borrow, M. Griffiths, N. A. Heerema, A. J. Carroll, P. Talley, N. Bown, N. Telford, F. M. Ross, L. Gaunt, R. J. Q. McNally, B. D. Young, P. Sinclair, V. Rand, M. R. Teixeira, O. Joseph, B. Robinson, M. Maddison, N. Dastugue, P. Vandenberghe, P. J. Stephens, J. Cheng, P. Van Loo, M. R. Stratton, P. J. Campbell, C. J. Harrison, Constitutional and somatic rearrangement of chromosome 21 in acute lymphoblastic leukaemia. *Nature* **508**, 98–102 (2014).
26. M. Martin, P. Ebert, T. Marschall, Read-Based Phasing and Analysis of Phased Variants with WhatsHap. *Methods in molecular biology (Clifton, N.J.)* **2590** (2023).
27. J. O. Korb, P. J. Campbell, Criteria for inference of chromothripsis in cancer genomes. *Cell* **152**, 1226–1236 (2013).
28. A.-K. Afflerbach, A. Albers, A. Appelt, L. Schweizer, W. Paulus, M. Bockmayr, U. Schüller, C. Thomas, Nanopore sequencing from formalin-fixed paraffin-embedded specimens for copy-number profiling and methylation-based CNS tumor classification. *Acta Neuropathol* **147**, 74 (2024).

Response to Reviewers

We thank the Reviewers for their careful reading of our manuscript and their positive comments. We provide below point-by-point responses to the comments raised by the Reviewers.

Point-by-Point Responses

Reviewer #1

I appreciate the authors' dedication in enhancing the quality of the manuscript and reworking the code to incorporate new functionalities. This work represents a valuable contribution to the community, and I would be delighted to see it published in Nature Methods.

One of the most notable improvements is the inclusion of the Tumor-only mode, which is thoroughly discussed on pages 9 to 12 of the rebuttal letter. I recognize the complexity of this undertaking and was pleasantly surprised by the authors' decision to pursue this approach. While it is expected that the Tumor-only mode introduces more noise, as demonstrated in Figures R10 to R13, the reporting of real event remains valuable. Certain SVs and SCNAs may be anticipated and actionable in clinical contexts. However, I do not advocate for clinical diagnostics based solely on the Tumor-only mode if a normal sample is obtainable. I align with the authors' conclusion: "...given all the reasons discussed above about the potentially low specificity of tumour only analysis, we still recommend the use of a matched normal sample as germline control whenever possible".

I certainly noted the efforts to improve speed and accuracy, as highlighted on page 13, and I particularly acknowledge the enhanced tumor purity estimates derived from the matched normal, rather than relying solely on the aligner (phased BAM). Additionally, employing a panel of SNPs as a fallback strategy is an excellent approach. Regarding the point-by-point discussion, I will address it page by page without directly copying the relevant paragraphs.

We thank the Reviewer for these positive comments and appreciating the effort and impact of the new functionalities implemented into SAVANA.

P.15: Reliance on third-party aligners: I agree that incorporating and using SAVANA with commonly used aligners is straightforward. My only concern is that errors from the aligner could potentially limit the performance of downstream software, possibly leading to missed or over-reported events. However, I believe this issue will largely be mitigated with the future use of long reads, which should provide more accurate genomic locus mapping (while alignment may be less critical in the context of SV detection). There is little that can be done to address this issue at present, and it should not impede the publication of the rest of the work. I appreciate the deposition of the 99 T/N reference data at the EGA. COLO829 reference has its limitations, and I am grateful to see that the authors also acknowledge that.

We thank the Reviewer for these encouraging and reasonable comments.

P.16: Germline callers: I agree with the authors that this is a common practice and a 'necessary evil' in benchmarking studies. It is far better to include comparisons to this 'ground truth' than to be left wondering how the somatic SV caller relate to the germline counterparts.

P.17: Thank you for providing additional clarification on the statistical assessment. I am pleased to see such a substantial number of replicates!

P.18: I welcome the tumor purity improvements and the authors' clarification regarding their estimation method. Thank you for clarifying the MSI status protocol. While I was not expecting many SNVs in the MMR genes, it is reassuring that it is now clearly stated that SNV/InDel calling was performed.

We thank the Reviewer for these positive comments and for appreciating the effort we invested in addressing their comments during the revision of our manuscript.

P.19: The application of machine learning in clinical practice is a complex topic that should not hinder the publication of the current work. I recognize the value of SAVANA in scientific research and investigative clinical practice. While the argument that model-state equates to version model is theoretically sound, its practical implementation across multiple clinical sites— especially without data sharing—could pose significant challenges. Ultimately, this is a matter for regulatory authorities to address. Thank you for the clarification. Fig. 19 and related: Thank you for the additional comments and the revised figure caption. Personally, I am not concerned about the RAM consumption, as long as the software is able to fit in the most computer configurations. Speed is always a positive feature—often the first question raised, but typically one of the last priorities in clinical settings. That said, great job!

In conclusion, I commend the authors for their additional efforts, their dedication to improving both the manuscript and the software, and their rational in addressing my concerns. Therefore, I strongly recommend this work for publication.

We thank the Reviewer for these encouraging and positive comments!

Reviewer #1 (Remarks on code availability):

I can verify that the code is easily accessible and available with few mouse clicks. The software installation is possible, and the tool can be set and run without any advanced knowledge through the Conda environment (or can be built from the source with a relatively small effort). The project seems very well managed, and the work looks solid. Although, I have the technical expertise to review all source lines and the logic, going through everything will consume too much of my time, so I haven't performed an extensive review.

We thank the Reviewer for these positive comments on the accessibility and sound management of the code, which are aspects we have worked on carefully over the last few years.

Reviewer #2

I appreciate the extra analysis and method improvement that had been done by the author. I think most of my previous questions are addressed. The revision is well organized but the manuscript lacks description for some important results, especially in Figure 2.

We thank the Reviewer for appreciating the work we have performed to address their comments. We provide responses below to address the remaining questions.

Questions and comments.

1. The green box in Figure R2 and others lack proper legend. It is hard to match the IGV view with its top panel. For example, in Figure R2 IGV view, the DEL's left breakpoint locates inside COL2A1, which seems discordant with the top panel.

We thank the Reviewer for bringing this to our attention. We have now modified Supplementary Figure 23 to improve clarity following the Reviewer's comments.

2. "Tumor-only" mode is definitely an important function for clinical applications. The author says "SAVANA can now detect somatic SVs by using a machine learning model trained for this purpose and using a large panel of normals to filter out likely germline SVs". While I am not able to find other details regarding to this part, what is the difference of this "tumor-only" model. Looking at the examples called from "tumor-only" mode, it seems that this function actually depends on filtering likely germline SVs in large normal panel.

We thank the Reviewer for the opportunity to clarify this point. The only difference between the tumour-only model and the one we use when a matched germline sample is provided is that, in tumour-only mode, we do not include covariates encoding information about normal reads mapping to the locus containing a candidate somatic breakpoint in the tumour sample. We have now clearly indicated in the Supplementary Table 2 which covariates are used in the tumour-only mode. As the Reviewer points out, a panel of normals is still required to filter SV calls detected in tumour-only mode as without normal information, the model will likely classify population-level SVs as somatic. We have added details on the panel of normals used to filter tumour-only somatic calls to the Methods section and in the Github repository of SAVANA. Specifically, SVs satisfying the following criteria are filtered out when using the tumour-only mode: (1) SVs mapping within 100bp of an SV present in the publicly available gnomadSV v4.1 database with a population allele-frequency of at least 0.10, and (2) any SVs in the publicly available Hartwig Medical Foundation SV panel of normals (v5 34). We have also updated the SAVANA repository to advise users to filter population SVs in this way.

3. How is the copy number estimated for centromere? For example, in Figure1, the total and minor allele copy number is 1 and 0. The example in Figure 4 shows total copy number 3 for the middle part of the centromere. Since centromere is one of the most divergent regions among humans, could those just be mapping artifact? I am aware this might out of the scope while it would be great to clarify this.

We thank the Reviewer for raising this very relevant question. We currently do not treat centromeric regions differently from other genomic regions in our copy number analysis. We decided to enable copy number calling in these regions whenever SAVANA is applied to long read data set with long enough reads to be mapped to centromeres. Therefore, we include centromeric regions in our analysis without additional imputation. However, we acknowledge that centromeres are highly divergent among individuals and that, as shown by the T2T Consortium, reads >50Kbp are needed to achieve high mapping quality, all of which could contribute to variability in copy number estimates. Thus, to facilitate excluding centromeric regions, we provide an optional blacklisting functionality within SAVANA to allow users to exclude centromeric regions from the analysis at their discretion.

4. The authors showed important evaluation for somatic SV detection in Figure 2b-d but need further details in the method section:

a. It seems that the author created two replicates. Dose "all replicates" in Figure 2b indicate both replicate?

Yes, to clarify "all replicates" refers to "both replicates". We thank the Reviewer for highlighting this as the confusion likely stems from the inconsistent labelling between this and Figure 2e. We have now updated Figure 2b to read "SV detected in both replicates" to clarify this point and maintain consistency between figures.

b. For the y-axis labels in Figure 2d "Fraction of validated SVs using replicates", dose it indicate SVs that are detected in both replicates?

We thank the Reviewer for the opportunity to clarify. The y-axis label refers to the number of SVs detected in both replicates divided by the total number of SVs detected in either one or both replicates. To clarify this, we have updated the wording to "Fraction of SVs identified in both replicates". We have also updated the figure caption to include this information.

c. Figure 2d decompose the result in 2c by allele frequency (x-axis in 2d) estimated by supporting reads. Looking at the circle size in 2d, it seems that savana detects similar number of SVs at different AF while Severus and NanomonSV decrease. It is also odd that Sniffles2 have fewer SVs of AF smaller than 0.2 and there is significant increase at (0.2,0.25].

We appreciate the opportunity to provide additional information regarding this plot. Due to large differences in the number of SVs called by each algorithm, it is difficult to see changes in the number of SVs detected at different allele frequencies unless the differences are several orders of magnitude, such as in the case noted by the Reviewer of a significant increase in Sniffles2 SVs at (0.2,0.25]. We can indeed confirm that the underlying calls show this significant increase, and have provided an additional figure (Supplementary Figure 8 in the revised version of manuscript, also shown on the right) to illustrate this point. In contrast, SAVANA recapitulates the full spectrum of allele fractions, with a high proportion of SVs present in both replicates, even at low allele fractions.

d. In 2d, the number of validated SVs using replicates can be calculated by multiply the value on y-axis and the circle size (total number of SVs). For example, the savana y-axis value ~ 0.75 at (0.15,0.2] with a circle size of 10000, so the number of validated SV is ~ 7500 . Is this number should be concordant with the number shown in Figure 2b?

We thank the Reviewer for pointing out that the Figure was not fully clear. The SAVANA y-axis value is at 0.776, the total number of SVs at this point is 3989, and the number of validated SVs 3094. We have added an additional bin to the point size to better reflect the underlying values. As for concordance with Figure 2b, Figure 2d shows SVs across samples in the cohort, whereas in 2b each point represents one of the 64 samples in the SV replication experiment. We have updated the Figure captions of both 2b and 2d to clarify this point.

5. I am confused about the phasing based evaluation shown in Figure 2g and 2h.
 a. How those SVs shown in 2h are selected for this evaluation.
 b. It says in line 288-289 "The key idea is that ...". It is true for a heterozygous SV but not for homozygous events.
 c. According to the method section, the author tested if some reads are incorrect assigned to an allele with the null hypothesis, a give SVs is supported by reads from one allele only. In 2g, inconclusive are SVs rejected the null hypothesis but the definition of inconsistent phasing is unclear.
 d. It seems that the test is only applied to SVs supported by reads from single allele, but why the inconclusive legend is used for "SVs with support from both alleles" in 2g.

We thank the Reviewer for indicating that this section was not sufficiently clear. Somatic SVs are supported by only one parental allele. They may appear as homozygous variants if a loss of heterozygosity (LOH) event has occurred in the genomic region to which they map. However, even in this situation, true somatic SVs would be supported by reads phased to only one parental allele. That is, we consider that the probability of the same exact somatic SV occurring in both parental alleles independently is zero.

For this analysis ($n = 82$ samples, Figures 2h and 2g), we selected SVs in regions with no LOH events (i.e. CN minor allele > 0.5) and where the number of copies of both parental alleles was comparable (i.e. CN major allele / CN minor allele < 2). We then classified the SVs based on whether they showed read-support from only one allele, which is the expected pattern for a true somatic SV, or from both alleles. For SVs with support from both alleles, we aimed to account for the potential error introduced by phasing when tagging sequencing reads. That is, we acknowledge that phasing sequencing reads is not perfect and that some reads might not be assigned to the correct parental allele. To address this, we estimated the binomial probability that the frequency of the most common allele among all phased SV-supporting reads was significantly lower than 0.95, assuming a 5% error from phasing. As a result, SVs were classified as "False positive: inconsistent phasing" if the P value was significant, or "Inconclusive" otherwise. While the first category clearly indicates a false positive event, the second category leaves us unable to distinguish whether the issue lies with the phasing or SV calling, hence why we label these cases as inconclusive.

6. In Figure 1, the step "SVs refine segmentation and copy-number change points rescue ..." is not described in the method section.

We thank the Reviewer for noting this issue, which we have now addressed in the Methods section.

7. The method section "Detection of SVs using existing algorithms" miss the desription for SVision-pro.

We have now added a description of how SVision-pro was run to the Methods section.

Reviewer #2 (Remarks on code availability):
The code is well organized with clear instructions.

We thank the Reviewer for this positive comment.

Reviewer #3

The authors answered all my comments.

We thank the Reviewer for this positive assessment and for helping us improve the manuscript and SAVANA.